# Understanding Difficult-to-learn Examples in Contrastive Learning: A Theoretical Framework for Spectral Contrastive Learning

## Abstract

Unsupervised contrastive learning has shown significant performance improvements in recent years, often approaching or even rivaling supervised learning in various tasks. However, its learning mechanism is fundamentally different from that of supervised learning. Previous works have shown that difficult-to-learn examples (well-recognized in supervised learning as examples around the decision boundary), which are essential in supervised learning, contribute minimally in unsupervised settings. In this paper, perhaps surprisingly, we find that the direct removal of difficult-to-learn examples, although reduces the sample size, can boost the downstream classification performance of contrastive learning. To uncover the reasons behind this, we develop a theoretical framework modeling the similarity between different pairs of samples. Guided by this theoretical framework, we conduct a thorough theoretical analysis revealing that the presence of difficult-to-learn examples negatively affects the generalization of contrastive learning. Furthermore, we demonstrate that the removal of these examples, and techniques such as margin tuning and temperature scaling can enhance its generalization bounds, thereby improving performance. Empirically, we propose a simple and efficient mechanism for selecting difficult-to-learn examples and validate the effectiveness of the aforementioned methods, which substantiates the reliability of our proposed theoretical framework.

## 1 Introduction

Contrastive learning has demonstrated exceptional empirical performance in the realm of unsupervised representation learning, effectively learning high-quality representations of high-dimensional data using substantial volumes of unlabeled data by aligning an anchor point with its augmented views in the embedding space (Chen et al., 2020a;b; He et al., 2020; Chen et al., 2021; Caron et al., 2020). Unsupervised contrastive learning may own quite different working mechanisms from supervised learning, as discussed in (Joshi & Mirzasoleiman, 2023). For example, difficult-to-learn examples (a well-recognized concept in supervised learning as examples around the decision boundary), which contribute the most to supervised learning, contribute the least or even negatively to contrastive learning performance. They show that on image datasets such as CIFAR-100 and STL-10, excluding 20%-40% of the examples does not negatively impact downstream task performance. More surprisingly, their results showed, but somehow failed to notice, that excluding these samples on certain datasets like STL-10 can lead to performance improvements in downstream tasks.

Taking a step further beyond their study, we find that this surprising result is not just a specialty of a certain dataset, but a universal phenomenon across multiple datasets. Specifically, we run SimCLR on the original CIFAR-10, CIFAR-100, STL-10, and TinyImagenet datasets, the SAS core subsets (Joshi & Mirzasoleiman, 2023) selected with a deliberately tuned size, and a subset selected by a sample removal mechanism to be proposed in this paper. In Figure 1, we report the gains of linear probing accuracy by using the subsets compared with the original datasets. We see that on all these bench-

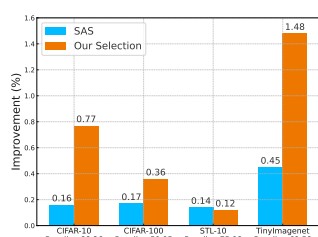

Figure 1: Excluding difficult-to-learn examples improves contrastive learning.

mark datasets, excluding a certain fraction of examples results in
comparable and even better downstream performance. This result is somewhat anti-intuitive because
deep learning models trained with more samples, benefiting from lower sample error, usually perform
better. Yet our observation indicates that difficult-to-learn examples can actually hurt contrastive
learning performances. This observation naturally raises a question:

> *What is the mechanism behind difficult-to-learn examples impacting the learning*
> *process of unsupervised contrastive learning?*

To comprehensively characterize such impact, we first develop a theoretical framework, i.e., the
similarity graph, to describe the similarity between different sample pairs. Specifically, pairs con-
taining difficult-to-learn samples, termed as difficult-to-learn pairs, exhibit higher similarities than
other different-class pairs. Based on this similarity graph, we derive the linear probing error bounds
of contrastive learning models trained with and without difficult-to-learn samples, proving that the
presence of difficult-to-learn examples negatively affects performance. Next, we prove that the most
straightforward idea of directly removing difficult-to-learn examples improves the generalization
bounds. Further, we also theoretically demonstrate that commonly used techniques such as margin
tuning (Zhou et al., 2024) and temperature scaling (Khaertdinov et al., 2022; Zhang et al., 2021;
Kukleva et al., 2023) mitigate the negative effects of difficult-to-learn examples by modifying the
similarity between sample pairs from different perspectives, thereby improving the generalization
bounds. Experimentally, we propose a simple but effective mechanism for selecting difficult-to-learn
samples that does not rely on pre-trained models. The performance improvements achieved by
addressing difficult-to-learn samples through the aforementioned methods align with our theoretical
analysis of the generalization bounds.

The contributions of this paper are summarized as follows:

- We find that removing certain training examples boosts the performance of unsupervised
  contrastive learning is a universal empirical phenomenon on multiple benchmark datasets.
  Through a mixing-image experiment, we conjecture that the removal of difficult-to-learn
  examples is the cause.

- We design a theoretical framework that models the similarity between different pairs of
  samples to characterize how difficult-to-learn samples in contrastive learning affect the
  generalization of downstream tasks. Based on this framework, we theoretically prove that
  the existence of difficult-to-learn samples hurts contrastive learning performances.

- We theoretically analyze how possible solutions, i.e. directly removing difficult-to-learn
  samples, margin tuning, and temperature scaling, can address the issue of difficult-to-learn
  examples by improving the generalization bounds in different ways.

- In experiments, we propose a simple and efficient mechanism for selecting difficult-to-learn
  examples and validate the effectiveness of the aforementioned methods, which substantiates
  the reliability of our proposed theoretical framework.

## 2  Difficult-to-learn Examples Hurt Contrastive Learning: A Mixing Image Experiment

We start this section by revealing that difficult-to-learn examples do hurt contrastive learning perfor-
mances through a proof-of-concept toy experiment.

The concept of difficult-to-learn examples is borrowed from supervised learning, denoting the
examples around the decision boundary. It is somewhat related to hard negative samples, a pure
unsupervised learning concept defined as highly similar negative samples to the anchor point, but is
different in nature. (See Appendix A.1 for more discussions.) However, in real image datasets, as
difficult-to-learn examples rely on the specific classifier trained in the supervised learning manner,
we can not preciously know the ground truth difficult-to-learn examples. Therefore, we in turn add
additional difficult-to-learn examples and observe the effects of these additional examples.

Specifically, we generate a new mixing-image dataset containing more difficult-to-learn samples
by mixing a $\gamma$ fraction of images on the CIFAR-10 dataset at the pixel level (these samples lying
around the class boundary), termed as $\gamma$-Mixed CIFAR-10 datasets. Then, we train the representative

contrastive learning algorithm SimCLR (Chen et al., 2020a) on the original, 10%-, and 20%-Mixed CIFAR-10 datasets using ResNet18 model. We report the linear probing accuracy in Figure 2.

Compared with the model trained on the original dataset, we find that with the mixed difficult-to-learn examples included in the training dataset, the performance of contrastive learning drops. This result indicates that the (mixed) difficult-to-learn samples significantly negatively impact contrastive learning. As the mixing ratio $\gamma$ increases, the performance drops, indicating that more difficult-to-learn examples lead to worse contrastive learning performances.

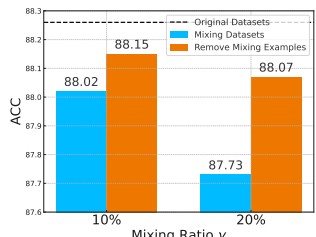

Moreover, we show that removing the (mixed) difficult-to-learn samples can boost performance. Specifically, we compare performance on the Mixed CIFAR-10 datasets with that on the datasets removing the mixed examples. As shown in Figure 2, despite being trained with a smaller sample size, models trained on datasets removing the mixed examples perform better than the ones trained with the mixed examples, which further verifies that difficult-to-learn examples hurt unsupervised contrastive learning, and removal of these difficult-to-learn examples can boost learning performance.

Figure 2: Excluding (mixed) difficult-to-learn examples improves contrastive learning.

## 3 THEORETICAL CHARACTERIZATION OF WHY DIFFICULT-TO-LEARN EXAMPLES HURT CONTRASTIVE LEARNING

In this section, to explain why difficult-to-learn examples negatively impact the performance of contrastive learning, we provide theoretical evidence on generalization bounds. In Section 3.1 we present the necessary preliminaries that lay the foundation for our theoretical analysis. In Section 3.2, we introduce the similarity graph describing difficult-to-learn examples. In Section 3.3, we respectively derive error bounds of contrastive learning with and without difficult-to-learn examples.

### 3.1 PRELIMINARIES

**Notations.** Given a natural data $\bar{x} \in \bar{\mathcal{X}} := \mathbb{R}^d$, we denote the distribution of its augmentations by $\mathcal{A}(\cdot|\bar{x})$ and the set of all augmented data by $\mathcal{X}$, which is assumed to be finite but exponentially large. For mathematical simplicity, we assume class-balanced data with $n$ denoting the number of augmented samples per class and $r + 1$ denoting the number of classes, hence $|\mathcal{X}| = n(r+1)$. Let $n_d$ represent the number of difficult-to-learn examples per class and $\mathbb{D}_d$ the set of difficult-to-learn examples.

**Similarity Graph (Augmentation Graph).** As described in HaoChen et al. (2021), an augmentation graph $\mathcal{G}$ represents the distribution of augmented samples, where the edge weight $w_{xx'}$ signifies the joint probability of generating augmented views $x$ and $x'$ from the same natural data, i.e., $w_{xx'} := \mathbb{E}\bar{x} \sim \bar{\mathcal{P}}[\mathcal{A}(x|\bar{x})\mathcal{A}(x'|\bar{x})]$. The total probability across all pairs of augmented data sums up to 1, i.e., $\sum_{x,x' \in \mathcal{X}} w_{xx'} = 1$. The adjacency matrix of the augmentation graph is denoted as $\boldsymbol{A} = (w_{xx'})_{x,x' \in \mathcal{X}}$, and the normalized adjacency matrix is $\bar{\boldsymbol{A}} = \boldsymbol{D}^{-1/2}\boldsymbol{A}\boldsymbol{D}^{-1/2}$, where $D := \mathrm{diag}(w_x)_{x \in \mathcal{X}}$, and $w_x := \sum_{x' \in \mathcal{X}} w_{xx'}$. The concept of augmentation graph is further extended to describe similarities beyond image augmentation, such as cross-domain images (Shen et al., 2022), multi-modal data (Zhang et al., 2023), and labeled examples (Cui et al., 2023).

**Contrastive losses.** For theoretical analysis, we consider the spectral contrastive loss $\mathcal{L}(f)$ proposed by HaoChen et al. (2021) as a good performance proxy for the widely used InfoNCE loss

$$\mathcal{L}_{\mathrm{Spec}}(\boldsymbol{x}; f) := -2 \cdot \mathbb{E}_{x,x^+}[f(x)^\top f(x^+)] + \mathbb{E}_{x,x'}\left[\left(f(x)^\top f(x')\right)^2\right]. \tag{1}$$

As proved in Johnson et al. (2022), the spectral contrastive loss and the InfoNCE loss share the same population minimum with variant kernel derivations. Further, the spectral contrastive loss is theoretically shown to be equivalent to the matrix factorization loss. For $F \in \mathbb{R}^{n \times k} = (u_x)x \in \mathcal{X}$, where $u_x = w_x^{1/2} f(x)$, the matrix factorization loss is:

$$\mathcal{L}_{\mathrm{mf}}(F) := \|\bar{\boldsymbol{A}} - FF^\top\|_F^2 = \mathcal{L}_{\mathrm{Spec}}(f) + const. \tag{2}$$

## 3.2 MODELING OF DIFFICULT-TO-LEARN EXAMPLES

We start by introducing a similarity graph, to describe the relationships between various samples. In contrastive learning, examples are used in a pairwise manner, so we define difficult-to-learn sample pairs as sample pairs that include at least one difficult-to-learn sample. As difficult-to-learn examples lie around the decision boundary, they should have higher augmentation similarity to examples from different classes. Therefore, it is natural for us to define the difficulty-to-learn pairs as different-class sample pairs with higher similarity. Correspondingly, easy-to-learn pairs are defined as different-class sample pairs containing no difficult-to-learn samples, or different-class sample pairs with lower similarity.

Specifically, we define the augmentation similarity between a sample and itself as $1$. Then we assume the similarity between same-class samples is $\alpha$ (Figure 3(a)), the similarity between a sample (conceptually far away from the class boundary) and all samples from other classes is $\beta$ (Figure 3(b)), and the similarity between different-class difficult-to-learn samples (conceptually close to the class boundary) is $\gamma$ (Figure 3(c)). Naturally, we have $0 \leq \beta < \gamma < \alpha < 1$.

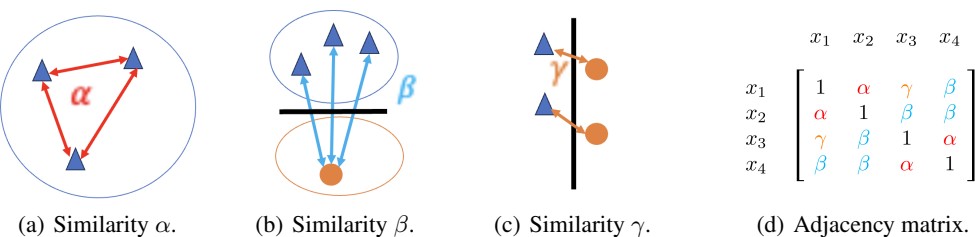

(a) Similarity $\alpha$.     (b) Similarity $\beta$.     (c) Similarity $\gamma$.     (d) Adjacency matrix.

Figure 3: Modeling of difficult-to-learn examples. The similarity between same-class samples is $\alpha$ (a), the similarity between different-class difficult-to-learn samples is $\gamma$ (c), and the similarity between other samples is $\beta$ (b). The adjacency matrix of a 4-sample subset is shown in (d).

In Figure 3(d), we illustrate our modeling of adjacency matrix through a 4-sample subset $\mathbb{D}_4 := x_1, x_2, x_3, x_4$, where $x_1$ and $x_2$ belong to Class 0, and $x_3$ and $x_4$ belong to Class 1. We define $x_1$ and $x_3$ as difficult-to-learn samples (assuming these two samples are distributed around the classification boundary as depicted in Figure 3(c)), i.e. $x_1, x_3 \in \mathbb{D}_d$. Conversely, we define $x_2$ and $x_4$ (assuming these samples are distributed far from the classification boundary) as easy-to-learn samples, i.e. $x_2, x_4 \in \mathbb{D}_4 \setminus \mathbb{D}_d$. The relationship between each pair of samples in $\mathbb{D}_4$ can be mathematically formulated as an adjacency matrix shown in Figure 3(d). In what follows, our theoretical analysis is based on the generalized similarity graph containing $|\mathcal{X}| = n(r + 1)$ samples. The formal definition of the generalized adjacency matrix is shown in Appendix B. In Section B.3, we also discuss that relaxation on the ideal adjacency matrix with randomizing the elements does not affect the core conclusions of this paper.

## 3.3 ERROR BOUNDS WITH AND WITHOUT DIFFICULT-TO-LEARN EXAMPLES

Based on the similarity graph in Section 3.2, we derive the linear probing error bounds for contrastive learning models trained with and without difficult-to-learn examples in Theorems 3.1 and 3.2. We mention that we adopt the label recoverability (with labeling error $\delta$) and realizability assumptions from HaoChen et al. (2021). The formal assumptions and proofs are shown in Appendix B.1.

**Theorem 3.1** (Error Bound without difficult-to-learn Examples). *Denote $\mathcal{E}_{\mathrm{w.o.}}$ as the linear probing error of a contrastive learning model trained on a dataset without difficult-to-learn examples. Then*

$$\mathcal{E}_{\mathrm{w.o.}} \leq \frac{4\delta}{1 - \frac{1-\alpha}{(1-\alpha)+n\alpha+nr\beta}} + 8\delta. \tag{3}$$

**Theorem 3.2** (Error Bound with difficult-to-learn Examples). *Denote $\mathcal{E}_{\mathrm{w.d.}}$ as the linear probing error of a contrastive learning model trained on a dataset with $n_d$ difficult-to-learn examples per*

*class. Then if $n_d \leq k \leq n_d + r + 1$, there holds*

$$\mathcal{E}_{\text{w.d.}} \leq \frac{4\delta}{1 - \frac{(1-\alpha)+r(\gamma-\beta)}{(1-\alpha)+n\alpha+nr\beta+n_d r(\gamma-\beta)}} + 8\delta. \quad (4)$$

**Discussions.** By comparing Theorems 3.1 and 3.2, also considering that $\frac{(1-\alpha)+r(\gamma-\beta)}{(1-\alpha)+n\alpha+nr\beta+n_d r(\gamma-\beta)} > \frac{1-\alpha}{(1-\alpha)+n\alpha+nr\beta}$, we see the presence of difficult-to-learn examples leads to a strictly worse linear probing error bound for a contrastive learning model. Moreover, more challenging difficult-to-learn examples (larger $\gamma - \beta$) result in worse error bounds. Specifically, when $\gamma = \beta$, i.e. no difficult-to-learn examples exist, the bound in Theorem 3.2 reduces to that in Theorem 3.1.

## 4 THEORETICAL CHARACTERIZATION ON HOW TO ELIMINATE EFFECTS OF DIFFICULT-TO-LEARN EXAMPLES

Building on the above unified theoretical framework, we theoretically analyze that directly removing difficult-to-learn samples (Section 4.1), margin tuning (Section 4.2), and temperature scaling (Section 4.3) can handle difficult-to-learn examples by improving the generalization bounds in different ways.

### 4.1 REMOVING DIFFICULT-TO-LEARN SAMPLES

In Figures 1 and 2, empirical experiments demonstrated that removing difficult-to-learn samples can improve learning performance. Corollary 4.1 provides a theoretical explanation for this counter-intuitive phenomenon based on our established framework.

**Corollary 4.1.** *Denote $\mathcal{E}_{\text{R}}$ as the linear probing error of a contrastive learning model trained on a selected subset removing all difficult-to-learn examples $\mathbb{D}_d$. Then there holds*

$$\mathcal{E}_{\text{R}} \leq \frac{4\alpha}{1 - \frac{1-\alpha}{(1-\alpha)+(n-n_d)\alpha+(n-n_d)r\beta}} + 8\delta. \quad (5)$$

Corollary 4.1 shows that when the difficult-to-learn examples are removed, the linear probing error bound has the same form as the case where no difficult-to-learn examples are present (Theorem 3.1), but with $n$ replaced by $n - n_d$. Compared with the case without removing difficult-to-learn examples (Theorem 3.2), the bound in equation 5 is smaller than that in equation 4 when $\gamma - \beta > \frac{n_d(1-\alpha)(\alpha+r\gamma)}{r[(1-\alpha)+(n-n_d)(\alpha+r\beta)]}$. This indicates that removing difficult-to-learn examples enhances the error bound when these samples are significantly harder than the easy ones (i.e., large $\gamma - \beta$) or when the number of difficult-to-learn samples is small (i.e., small $n_d$).

### 4.2 MARGIN TUNING

Margin tuning is useful in contrastive learning as highlighted in (Zhou et al., 2024). Here, we delve into how margin tuning can enhance the generalization in the presence of difficult-to-learn examples.

**Theorem 4.2.** *The margin tuning loss is equivalent to the matrix factorization loss*

$$\mathcal{L}_{\text{mf}-\text{M}}(F) := \|(\bar{A} - \bar{M}) - FF^\top\|_F^2, \quad (6)$$

*where $\bar{A}$ is the normalized adjacency matrix, and $\bar{M}$ is the normalized margin matrix.*

Theorem 4.2 indicates that adjusting margins alters the similarity graph by subtracting a normalized margin matrix $\bar{M}$ from the normalized similarity matrix $\bar{A}$. Intuitively, by subtracting the additional similarity values of difficult-to-learn examples with appropriately chosen margins, the remaining values will match those of easy-to-learn examples. Specifically, in the following Theorem 4.3, we show that properly chosen margins can eliminate the negative impact of difficult-to-learn examples.

**Theorem 4.3.** *Denote $\mathcal{E}_{\text{M}}$ as the linear probing error for the margin tuning loss equation 31 trained on a dataset with difficult-to-learn samples $\mathbb{D}_d$. If we let*

$$m_{x,x'} = c_0/(c_1^2 c_2) \cdot (\gamma - \beta) \quad (7)$$

*for $y(x) \neq y(x'), x, x' \in \mathbb{D}_d$, where $c_0 := (1 - \alpha) + n\alpha + (n - n_d)r\beta$, $c_1 := (1 - \alpha) + n\alpha + nr\beta + n_d r(\gamma - \beta)$ and $c_2 := (1 - \alpha) + n\alpha + nr\beta$, and $m_{x,x'} = 0$ for $x, x' \notin \mathbb{D}_d$, then we have*

$$\mathcal{E}_{\mathrm{M}} = \mathcal{E}_{\mathrm{w.o.}}. \tag{8}$$

Note that when $n$ is large enough, $m_{x,x'}$ for $x$ or $x' \notin \mathbb{D}_d$ are higher-order infinitesimals relative to equation 7, and primarily affect normalization rather than the core problem. Thus, we focus on cases where $x, x' \in \mathbb{D}_d$ and defer specific forms of other $m_{x,x'}$ values to the proofs for brevity.

Theorem 4.3 shows that with appropriately chosen margins, the linear probing error bound for the margin tuning loss in the presence of difficult-to-learn examples becomes equivalent to the standard contrastive loss without such examples, as indicated in Theorem 3.1. Since *equation 7 > 0*, this suggests applying a positive margin to the difficult-to-learn example pairs. Additionally, the more challenging the example pairs are (i.e., the larger $\gamma - \beta$), the greater the margin value should be.

### 4.3 TEMPERATURE SCALING

Temperature scaling is a well-validated technique in various contrastive learning tasks (Khaertdinov et al., 2022; Zhang et al., 2021; Kukleva et al., 2023). Here, we investigate how temperature scaling can enhance generalization, particularly in the presence of difficult-to-learn examples.

**Theorem 4.4.** *The temperature scaling loss is equivalent to the matrix factorization loss*

$$\mathcal{L}_{\mathrm{mf-T}}(F) := \|\boldsymbol{T} \odot \bar{\boldsymbol{A}} - FF^{\top}\|_{wF}^2, \tag{9}$$

*where $\bar{\boldsymbol{A}}$ is the normalized adjacency matrix of similarity graph, $\boldsymbol{T} \odot \bar{\boldsymbol{A}}$ is the element-wise product of matrices $\boldsymbol{T}$ and $\bar{\boldsymbol{A}}$, and $\| \cdot \|_{wF}$ is the weighted Frobenius norm (specified in the proof).*

Theorem 4.4 shows that adjusting temperatures modifies the similarity graph by multiplying the temperature values with the normalized similarity matrix $\bar{\boldsymbol{A}}$. Intuitively, by scaling the similarity values between difficult-to-learn examples, we can match these values to those of easy-to-learn examples, thereby mitigating the negative effects of difficult-to-learn examples. Specifically, the following Theorem 4.5 outlines the appropriate temperature values to be chosen.

**Theorem 4.5.** *Denote $\mathcal{E}_{\mathrm{T}}$ as the linear probing error for the temperature scaling loss equation 40 trained on a dataset with difficult-to-learn samples $\mathbb{D}_d$. If we let*

$$\tau_{x,x'} = (c_1/c_2)(\beta/\gamma) \tag{10}$$

*for $y(x) \neq y(x'), x, x' \in \mathbb{D}_d$, where $c_1 := (1 - \alpha) + n\alpha + nr\beta + n_d r(\gamma - \beta)$ and $c_2 := (1 - \alpha) + n\alpha + nr\beta$, and $\tau_{x,x'} = 1$ for $x, x' \notin \mathbb{D}_d$, then we have*

$$\mathcal{E}_{\mathrm{T}} \leq \frac{4[1 - (n_d/n)^2 + (\gamma/\beta)^2(n_d/n)^2]\delta}{1 - \frac{1-\alpha}{(1-\alpha)+n\alpha+nr\beta}} + 8\delta. \tag{11}$$

Likewise, here we only focus on the temperature values between difficult-to-learn examples, and defer the specific forms of other $\tau_{x,x'}$ values to the proofs for brevity.

Theorem 4.5 shows the linear probing error bound of the temperature scaling loss when trained on data containing difficult-to-learn examples. Specifically, with large $n$ and $n_d/n \to 0$, we have $\mathcal{E}_{\mathrm{T}}/\mathcal{E}_{\mathrm{w.o.}} - 1 \approx O((n_d/n)^2)$ and $\mathcal{E}_{\mathrm{w.d.}}/\mathcal{E}_{\mathrm{w.o.}} - 1 \approx O(1/n)$. This indicates that, when $O(n_d) \lesssim O(n^{1/2})$, $\mathcal{E}_{\mathrm{T}}/\mathcal{E}_{\mathrm{w.o.}} \lesssim \mathcal{E}_{\mathrm{w.d.}}/\mathcal{E}_{\mathrm{w.o.}}$, meaning $\mathcal{E}_{\mathrm{T}}$ converges faster to $\mathcal{E}_{\mathrm{w.o.}}$. Detailed calculations show that when $n_d < \sqrt{\frac{r}{(\alpha+r\beta)(\gamma+\beta)}}\beta \cdot n^{1/2}$, there holds $\mathcal{E}_{\mathrm{T}} < \mathcal{E}_{\mathrm{w.d.}}$, which means that temperature scaling improves the error bound. Note that $(c_1/c_2)(\beta/\gamma) \in (0, 1)$. This inspires us to choose smaller temperature values for the difficult-to-learn example pairs. The more difficult the example pairs (smaller $\beta/\gamma$), the smaller the temperature values that should be chosen.

## 5 VERIFICATION EXPERIMENTS

This paper primarily focuses on theoretical analysis, explaining how different samples in contrastive learning impact generalization. The experiments in this part are mainly designed to validate the

theoretical insights and demonstrate that the proposed directions for improving performance are sound. The experiments are not intended to achieve state-of-the-art results but rather to confirm the correctness of our theoretical findings. We hope that readers will appreciate the theoretical contributions of this work and not focus excessively on the experimental results.

In Section 5.1, we present a straightforward and efficient mechanism for selecting difficult-to-learn samples. We subsequently conduct a comprehensive evaluation of various methods, including the removal of difficult-to-learn samples (Section 5.2), margin tuning (Section 5.3), and temperature scaling (Section 5.4), all of which are theoretically established to mitigate the impact of these difficult-to-learn examples. In Section 5.5, we propose an extended method that combines margin tuning and temperature scaling, and discuss the scalability under different paradigms and the connection between difficult-to-learn samples and long-tail distribution. The specific loss forms and algorithms can be found in Appendix A.2.

### 5.1 DIFFICULT-TO-LEARN EXAMPLES SELECTION

Based on the preceding analysis, we have established that difficult-to-learn samples play a crucial role in enhancing the generalization of contrastive learning. In this section, we aim to develop a straightforward and efficient selection mechanism to validate our theoretical analysis, which avoids additional pretrained models and extra costs (Joshi & Mirzasoleiman, 2023).

To identify difficult-to-learn sample pairs—those from different classes but with high similarity—we compute the cosine similarity of each sample to other samples in the same batch using features before projector mapping. We define $posHigh$ and $posLow$ as percentiles of the similarity sorted in descending order, where $Sim_{posHigh}$ and $Sim_{posLow}$ are the corresponding similarities. Generally, following the characterization in Section 3.2 and Appendix B, we can roughly assume $posHigh$ corresponds to $1/(r+1)$, where $r+1$ is the class number[1]. Sample pairs with cosine similarities above $Sim_{posHigh}$ are considered from the same class. Sample pairs with the similarity between $Sim_{posHigh}$ and $Sim_{posLow}$ are considered as difficult-to-learn examples. Sample pairs with cosine similarities below $Sim_{posLow}$ are considered as easy-to-learn samples from different classes. Here for $posLow$, we note that when optimizing $\gamma$ of difficult-to-learn examples, if some easy-to-learn samples are involved, the process will also optimize $\beta$, which is a good thing for the representation learning to push samples from different classes further apart. Therefore, we can easily find a value close to the bottom of the sorted similarity for $posLow$, even 100%. Experiments in Figure 4(a) and Figure 4(b) show that our method is not sensitive to the exact values of $posHigh$ and $posLow$.

Using this selection mechanism, for an augmented sample pair $(x_i, x_j)$ in the current batch, we define the selecting indicator of difficult-to-learn pairs as

$$p_{i,j} := \mathbf{1}_{[Sim_{posLow} \leq s_{ij} < Sim_{posHigh}]}, \tag{12}$$

where $s_{i,j}$ denotes the cosine similarity between the representations of $x_i$ and $x_j$, and $\mathbf{1}_{[\text{condition}]}$ denotes the indicator function returning 1 if the condition holds and 0 otherwise. For each sample $x_i$, we get a vector $P_i = (p_{i,j})_{j=1}^{2N}$ representing the indicator of difficult-to-learn pairs. After calculating these indicators for all samples in the current batch, we stack the vectors $P_i$ row-wise to create the selection matrix $\boldsymbol{P}$. In practice, $P_i$ can be computed in parallel, making the computation of $\boldsymbol{P}$ efficient. The elements of $\boldsymbol{P}$ are either 0 or 1, indicating whether pairs are difficult-to-learn or not.

We can use the class information to verify the proportions of sample pairs from different classes in $(Sim_{posLow}, Sim_{posHigh})$ on CIFAR-10, which can demonstrate the effectiveness of our selection mechanism. As shown in Figure 4(c), along with the progress of training, the ratio of sample pairs from different classes approaches close to 100% within the range $(Sim_{posLow}, Sim_{posHigh})$.

### 5.2 REMOVING DIFFICULT-TO-LEARN SAMPLES

We here introduce a simple and practical method for removing difficult-to-learn samples based on our proposed selection mechanism. Eliminating the impact of difficult-to-learn samples means preventing sample pairs that include difficult-to-learn samples from interfering with the training process. To achieve this, we use the selection matrix $\boldsymbol{P}$ to identify and remove difficult-to-learn samples.

---

[1]We do not need to know the exact label of each class. A rough class number is enough, which can be easily known by clustering.

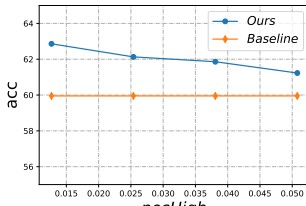 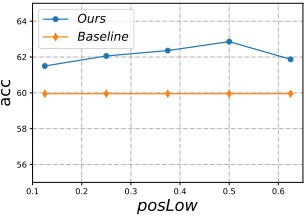 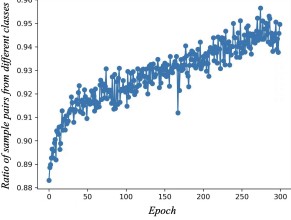

(a) Acc with different $posHigh$     (b) Acc with different $posLow$     (c) Ratio trends in the training

Figure 4: Parameter sensitivity of difficult-to-learn example interval ends $posHigh$ (4(a)) and $posLow$ (4(b)). Parameter analysis on CIFAR-100: the trend of the ratio of sample pairs from different classes in $(Sim_{posLow}, Sim_{posHigh})$ during the training process (4(c)).

Table 1: Classification accuracy with or without removing difficult-to-learn examples on CIFAR-10, CIFAR-100, STL-10 and TinyImagenet dataset using SimCLR. Results are averaged over three runs.

| Method | CIFAR-10 | CIFAR-100 | STL-10 | TinyImagenet |
|---|---|---|---|---|
| SimCLR (Baseline) | 88.26 | 59.95 | 75.98 | 69.58 |
| SimCLR (Removing) | **89.03** | **60.31** | **76.10** | **71.06** |

The results are in Table 1, in the first line we report the results of SimCLR as a baseline method. The second line is the result of removing difficult-to-learn examples. It can be observed that removing difficult-to-learn examples yields a 0.8% performance boost on CIFAR-10, a 0.6% performance boost on CIFAR-100, and a 3.7% performance boost on TinyImagenet compared to the baseline method which does not address difficult-to-learn samples. We reach the same conclusion as in (Joshi & Mirzasoleiman, 2023): By removing difficult-to-learn samples, we can achieve comparable results or even slight improvements over the baseline. However, removing difficult-to-learn samples may not be the most effective method for handling difficult-to-learn samples, because it shrinks sample size. Next, we investigate two techniques that handle difficult-to-learn samples better, margin tuning in Section 5.3 and temperature scaling in Section 5.4.

## 5.3 MARGIN TUNING ON DIFFICLUT-TO-LEARN SAMPLES

To effectively apply margin tuning in line with our theoretical analysis, we adopt a margin tuning factor $\sigma > 0$. For the selected difficult-to-learn sample pairs identified by the selection matrix $\boldsymbol{P}$, we add a margin $\sigma$ to the similarity values, and for the unselected pairs, we use the original InfoNCE.

Table 2: Classification accuracy with or without margin tuning on CIFAR-10, CIFAR-100, STL-10 and TinyImagenet dataset. Results are averaged over three runs.

| Method | CIFAR-10 | CIFAR-100 | STL-10 | TinyImagenet |
|---|---|---|---|---|
| Baseline | 88.26 | 59.95 | 75.98 | 69.58 |
| MT (All Samples) | 88.52 | 60.09 | 76.02 | 70.06 |
| MT (Selected Samples) | **89.16** | **61.28** | **76.83** | **79.14** |

As shown in Table 2, in the first line we report the result of SimCLR as baseline. The second line is the result of using margin tuning (we here use MT as an abbreviation) to all samples, while the third line is the result of using margin tuning to selected difficult-to-learn samples. We can observe that applying margin tuning to all samples directly only achieves comparable results as the baseline, highlighting the importance of the selection mechanism for difficult-to-learn examples. While applying margin tuning to selected samples yields a 1.0% performance boost on CIFAR-10, a 2.2% performance boost on CIFAR-100, and a 13.7% performance boost on TinyImagenet compared to the baseline method which has no operation to difficult-to-learn samples. These results validate both the effectiveness of our selection mechanism and the reliability of our analysis on margin tuning.

## 5.4 TEMPERATURE SCALING ON DIFFICLUT-TO-LEARN SAMPLES

We define the temperature scaling factor $\rho > 0$. Given the base temperature $\tau > 0$, we attach temperature $\rho\tau$ to the selected difficult-to-learn sample pairs identified by the selection matrix $P$, whereas attach base temperature $\tau$ to the unselected pairs.

Table 3: Classification accuracy with or without temperature scaling on CIFAR-10, CIFAR-100, STL-10 and TinyImagenet dataset. Results are averaged over three runs.

| Method | CIFAR-10 | CIFAR-100 | STL-10 | TinyImagenet |
|---|---|---|---|---|
| Baseline | 88.26 | 59.95 | 75.98 | 69.58 |
| TS (All Samples) | 88.38 | 59.20 | 75.76 | 69.36 |
| TS (Selected Samples) | **89.24** | **61.67** | **76.62** | **78.52** |

As shown in Table 3, in the first line we report the result of SimCLR as baseline. The second line is the result of using temperature scaling (we here use TS as an abbreviation) to all samples, while the third line is the result of using temperature scaling to selected difficult-to-learn samples. We observe that applying temperature scaling to all samples directly can even hurt the performance of contrastive learning, highlighting the importance of selecting difficult-to-learn examples. In contrast, applying temperature scaling to selected samples yields a 1.1% performance improvement on CIFAR-10, a 2.9% performance improvement on CIFAR-100, and a 12.8% performance improvement on TinyImagenet compared to the baseline method which has no operation to difficult-to-learn samples. These experimental results validate both the effectiveness of our selection mechanism and the reliability of our analysis on temperature scaling.

## 5.5 EXTENSIONS

**Combined method.** From Sections 4.2 and 4.3, we observe that margin tuning and temperature scaling eliminate the effects of difficult-to-learn examples in different ways. Therefore, it is natural to combine the two methods, and see if the combined method could reach better performances.

Table 4: Classification accuracy with or without combined method on CIFAR-10, CIFAR-100, STL-10 and TinyImagenet dataset. Results are averaged over three runs.

| Method | CIFAR-10 | CIFAR-100 | STL-10 | TinyImagenet |
|---|---|---|---|---|
| Baseline | 88.26 | 59.95 | 75.98 | 69.58 |
| Margin Tuning | 89.16 | 61.28 | 76.83 | 79.14 |
| Temperature Scaling | 89.24 | 61.67 | 76.62 | 78.52 |
| Combined Method | **89.68** | **62.86** | **77.35** | **80.00** |

As shown in Table 4, the first line reports the result of SimCLR. The second line shows the result of using margin tuning . The third line shows the result of using temperature scaling. The fourth line shows the result of using the combined method which yields a 1.6% performance improvement on CIFAR-10, a 4.9% performance improvement on CIFAR-100 and a 15.0% performance improvement on TinyImagenet compared to the baseline method. The improvement surpasses that achieved by using only margin tuning or temperature scaling. The combined method on the Mixed CIFAR-10 datasets also achieves performance improvements consistently as shown in Section A.5.

**Alternative contrastive learning paradigm.** We delve deeper into the scalability of our methods across various self-supervised learning paradigms. Results in Table 5 demonstrate consistent performance enhancements comparable to those achieved by SimCLR on the MoCo on CIFAR-10.

**Complex classification scenarios.** We explore our method by targeting difficult-to-learn samples under the long-tail classification scenario, where boundary samples are even more difficult to learn according to the imbalanced distributions. The findings in Table 6 illustrate that our approach outperforms the benchmark method SimCLR in scenarios involving distributional imbalance, indicating the adaptivity of our approach to complex classification scenarios.

Table 5: The results of incorporating the Combined method with different architectures on CIFAR-10.

| Method | MoCo | SimCLR |
|---|---|---|
| Baseline | 85.84 | 88.26 |
| Combined Method | **86.82** | **89.68** |

Table 6: Classification accuracy by using Combined method on TinyImagenet-LT. We also use SimCLR as the baseline method.

| Method | TinyImagenet-LT |
|---|---|
| Baseline | 43.34 |
| Combined Method | **47.62** |

**Further discussions.** To further illustrate our experimental results, we provide a sensitivity analysis of parameters in section A.4. We also conduct a more detailed analysis of results in Table 5 and Table 6 in Section A.5. Furthermore, discussions about which features are advantageous for selecting difficult-to-learn examples are also presented in Section A.5. In Section A.5, we have also included the experimental results on ImageNet-1K, the trending of the derived bounds with Mixed CIFAR-10 dataset and the significance analysis of $\gamma$ and $\beta$.

## 6 CONCLUSION

In this paper, we construct a theoretical framework to specifically analyze the impact of difficult-to-learn examples on contrastive learning. We prove that difficult-to-learn examples hurt the performance of contrastive learning from the perspective of linear probing error bounds. We further demonstrate how techniques such as margin tuning, temperature scaling, and the removal of these examples from the dataset can improve performance from the perspective of enhancing the generalization bounds. The detailed experimental results demonstrate the reliability of our theoretical analysis.

## 7 ETHICS STATEMENT

This study does not raise any ethical issues. We devise computational algorithms on benchmark datasets and provide theoretical explanations.

## 8 REPRODUCIBILITY STATEMENT

All the datasets used in this paper are open-source and publicly available for download. The proposed method can be found in Algorithm 1. All experimental settings and implementation details can be referred to in Appendix A.3. Detailed mathematical formations, assumptions, and all proofs of the theoretical parts mentioned in this paper are provided in Appendix B.

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

# A APPENDIX

## A.1 RELATED WORKS

**Self-supervised contrastive learning.** Self-supervised contrastive learning (Chen et al., 2020a;b; He et al., 2020; Chen et al., 2021) aims to learn an encoder that maps augmentations (e.g. flips, random crops, etc.) of the same input to proximate features, while ensuring that augmentations of distinct inputs yield divergent features. The encoder, once pre-trained, is later fined-tuned on a specific downstream dataset. The effectiveness of contrastive learning methods are typically evaluated through the performances of the downstream tasks such as linear classification. Depending on the reliance of negative samples, contrastive learning methods can be broadly categorized into two kinds. The first kind (Chen et al., 2020a;b; He et al., 2020) learns the encoder by aligning an anchor point with its augmented versions (positive samples) while at the same time explicitly pushing away the others (negative samples). On the other hand, the second kind do not depend on negative samples. They often necessitate additional components like projectors (Grill et al., 2020), stop-gradient techniques (Chen & He, 2021), or high-dimensional embeddings (Zbontar et al., 2021). Nevertheless, the first kind of methods continue to be the mainstream in self-supervised contrastive learning and have been expanded into numerous other domains (Khaertdinov et al., 2021; Aberdam et al., 2021; Lee et al., 2022). The analysis and discussions of this paper focus mainly on the first kind of contrastive learning methods that relies on both positive and negative samples.

**Contrastive Learning Theory.** The early studies of theoretical aspects of contrastive learning manage to link contrastive learning to the supervised downstream classification. Arora et al. (2019) proves that representations learned by contrastive learning algorithms can achieve small errors in the downstream linear classification task. Nozawa & Sato (2021); Ash et al. (2022); Bao et al. (2022) incorporate the effect of negative samples and further extend surrogate bounds. Later on, HaoChen et al. (2021) focuses on the unsupervised nature of contrastive learning by modeling the feature similarities between augmented samples and provides generalization guarantee for linear evaluation through borrowing mathematical tools from spectral clustering. The idea of modeling similarities is later extended to analyzing contrastive learning for unsupervised domain adaption (Shen et al., 2022) and weakly supervised learning (Cui et al., 2023). In a similar vein, Wang et al. (2021) put forward the idea of *augmentation overlap* to explain the alignment of positive samples. Besides, contrastive learning is also interpreted through various other theoretical frameworks in unsupervised learning, such as nonlinear independent component analysis (Zimmermann et al., 2021), neighborhood component analysis (Ko et al., 2022), stochastic neighbor embedding (Hu et al., 2023), geometric analysis of embedding spaces (Huang et al., 2023), and message passing techniques (Wang et al., 2023). In this paper, our basic assumptions are based on HaoChen et al. (2021) and focus on modeling the similarities between difficult-to-learn example pairs.

**Difference between difficult-to-learn examples and hard negative samples.** Difficult-to-learn examples and hard negative samples both significantly affect the performance of self-supervised learning. However, while difficult-to-learn examples are associated with the classification boundary, hard negative samples (Robinson et al., 2020; Kalantidis et al., 2020) are defined in relation to the anchor point. Previous research on hard negative sampling typically modifies contrastive learning models to emphasize these challenging samples so as to achieve better performance. In contrast, our findings indicate that unmodified contrastive learning models experience performance degradation due to the existence of difficult-to-learn samples. Aside from ad hoc modifications, a straightforward removal of these difficult-to-learn samples can also boost performance. As a systematic explanation of this finding is lacking, we establish a unified theoretical framework that addresses this challenge.

## A.2 LOSS FUNCTIONS OF SAMPLE REMOVAL, MARGIN TUNING, AND TEMPERATURE SCALING

Based on the sample selection matrix $\boldsymbol{P}$ defined in equation 12, we adapt the InfoNCE loss into versions of sample removal, margin tuning, and temperature scaling, respectively.

**Sample Removal.** We define the removal loss as follows:

$$\ell_{\mathrm{R}}(i,j) := -\log \frac{\exp\left((s_{i,j}(1-p_{i,j}))/\tau\right)}{\sum_{k=1}^{2N} \mathbf{1}_{[k\neq i]} \exp\left((s_{i,k}(1-p_{i,k}))/\tau\right)}, \tag{13}$$

where $s_{i,j}$ denotes the similarity between augmented instances $x_i$ and $x_j$. If $p_{i,j} = 0$, the sample pair $x_i$ and $x_j$ does not include difficult-to-learn samples, so $(s_{i,j}(1 - p_{i,j}))/\tau = s_{i,j}/\tau$, retaining the original form of the InfoNCE loss. If $p_{i,j} = 1$, the sample pair $x_i$ and $x_j$ are difficult-to-learn pairs, so $(s_{i,j}(1 - p_{i,j}))/\tau = 0$, effectively removing them.

**Margin Tuning.** We start with the basic form of the widely used InfoNCE loss and define the margin tuning loss for each positive pair. Specifically, within each minibatch of size $N$, we generate $2N$ samples through data augmentation. Given the margin tuning factor $\sigma > 0$, for an anchor sample $x_i$ and its corresponding positive sample $x_j$, we define the margin tuning loss as follows:

$$\ell_{\mathrm{M}}(i, j) := - \log \frac{\exp\left((s_{i,j} + p_{i,j}\sigma)/\tau\right)}{\sum_{k=1}^{2N} \mathbf{1}_{[k \neq i]} \exp\left((s_{i,k} + p_{i,k}\sigma)/\tau\right)}, \tag{14}$$

where $s_{i,j}$ denotes the similarity between augmented instances $x_i$ and $x_j$, and $\tau > 0$ denotes the temperature parameter. After the above operation, we assign the same margin value to all selected difficult-to-learn sample pairs, achieving the goal of margin tuning for specific sample pairs.

**Temperature Scaling.** To apply temperature scaling consistent with our theoretical analysis, we start with the basic form of the InfoNCE loss and define the temperature scaling loss for each positive pair. Specifically, within each minibatch, given the temperature scaling factor $\rho$, for an anchor sample $x_i$ and its corresponding positive sample $x_j$, we define the temperature scaling loss as follows:

$$\ell_{\mathrm{T}}(i, j) := - \log \frac{\exp\left(\frac{s_{i,j}}{[p_{i,j}\rho + (1 - p_{i,j})]\tau}\right)}{\sum_{k=1}^{2N} \mathbf{1}_{[k \neq i]} \exp\left(\frac{s_{i,k}}{[p_{i,k}\rho + (1 - p_{i,k})]\tau}\right)}, \tag{15}$$

where $s_{i,j}$ denotes the similarity between augmented instances $x_i$ and $x_j$.

**Combined Method.** The combined loss function as

$$\ell(i, j) := - \log \frac{\exp\left(\frac{s_{i,j} + p_{i,j}\sigma}{[p_{i,j}\rho + (1 - p_{i,j})]\tau}\right)}{\sum_{k=1}^{2N} \mathbf{1}_{[k \neq i]} \exp\left(\frac{s_{i,k} + p_{i,k}\sigma}{[p_{i,k}\rho + (1 - p_{i,k})]\tau}\right)}, \tag{16}$$

where $s_{i,j}$ denotes the similarity between augmented instances $x_i$ and $x_j$. The whole training procedure of the combined method is shown in Algorithm 1.

---

**Algorithm 1** Training procedure of Combined method

---

**Input:** batch size $N$, base temperature $\tau$, $posHigh$ and $posLow$ for determining the size of the interval, margin tuning factor $\sigma$, temperature scaling factor $\rho$, encoder $f(\cdot)$, projector $g(\cdot)$ and data augmentation $T$.
**Output:** encoder network $f(\cdot)$, and throw away $g(\cdot)$.
 1: **for** sampled minibatch $\{\bar{x}_k\}_{k=1}^N$ **do**
 2:     **for** all $k \in \{1,...,N\}$ **do**
 3:         Draw two augmentation functions $t, t' \sim T$;
 4:         $x_{2k-1} = t(\bar{x}_k)$ and $x_{2k} = t'(\bar{x}_k)$;
 5:         $h_{2k-1} = f(x_{2k-1})$ and $h_{2k} = f(x_{2k})$;
 6:         $z_{2k-1} = g(h_{2k-1})$ and $z_{2k} = g(h_{2k})$.
 7:     **end for**
 8:     **for** all $k \in \{1,...,2N\}$ **do**
 9:         Calculate $P_i = (p_{i,j})_{j=1}^{2N}$ by using $h_j, j \in \{1,...,2N\}$ according to Eq. equation 12;
10:     **end for**
11:     The matrix $\boldsymbol{P}$ is obtained by splicing $P_i, i \in \{1,...,2N\}$ by rows.
12:     **for** all $i \in \{1,...,2N\}$ and all $j \in \{1,...,2N\}$ **do**
13:         $s_{i,j} = z_i^\top z_j / (\|z_i\| \|z_j\|)$.
14:     **end for**
15:     Calculate $\ell(i, j)$ according to Eq. equation 16;
16:     Calculate $\mathcal{L} = \frac{1}{2N} \sum_{k=1}^N [\ell(2k - 1, 2k) + \ell(2k, 2k - 1)]$; Update networks $f$ and $g$ to minimize $\mathcal{L}$.
17: **end for**

---

### A.3 TRAINING DETAILS

We run all experiments on an NVIDIA GeForce RTX 3090 24G GPU and we run experiments with ResNet-18 on the CIFAR-10, CIFAR-100 and STL-10 dataset and ResNet-50 on the TinyImagenet dataset.

For CIFAR-10 we set batch size as 512, learning rate as 0.25 and base temperature as 0.5. We choose 0.15 as the $posHigh$ and 0.22 as the $posLow$. We set $\sigma$ as 0.03 and $\rho$ as 0.6 for CIFAR-10. For both our method and SimCLR, we evaluate the models using linear probing, when evaluating we set batch size as 512 and learning rate as 1. This experimental setup is also applicable to the Mixed CIFAR-10 dataset.

For CIFAR-100 we set batch size as 512, learning rate as 0.5 and base temperature as 0.1. We choose 0.013 as the $posHigh$ and 0.5 as the $posLow$. We set $\sigma$ as 0.1 and $\rho$ as 0.7 for CIFAR-100. For both our method and SimCLR, we evaluate the models using linear probing, when evaluating we set batch size as 512 and learning rate as 0.1.

For STL-10 we set batch size as 256, learning rate as 0.5 and base temperature as 0.1. We choose 0.15 as the $posHigh$ and 0.22 as the $posLow$. We set $\sigma$ as 0.1 and $\rho$ as 0.7 for STL-10. For both our method and SimCLR, we evaluate the models using linear probing, when evaluating we set batch size as 256 and learning rate as 0.1.

For TinyImagenet we set batch size as 512, learning rate as 0.5 and base temperature as 0.5. We choose 0.013 as the $posHigh$ and 0.5 as the $posLow$. We set $\sigma$ as 0.1 and $\rho$ as 0.7 for TinyImagenet. For both our method and SimCLR, we evaluate the models using linear probing, when evaluating we set batch size as 512 and learning rate as 0.1.

For the experimental results presented in Figure 1, we selected 20% SAS coreset for CIFAR-10, 95% SAS coreset for CIFAR-100, 80% SAS coreset for STL-10, and 60% SAS coreset for TinyImagenet, following the filtering method mentioned in (Joshi & Mirzasoleiman, 2023).

### A.4 PARAMETER SENSITIVITY ANALYSIS

**Evaluating different $\sigma$ used in margin tuning part**. The intention of $\sigma$ is to add margins to the similarity terms between difficult-to-learn example pairs. We show the performance with different $\sigma$ in Figure 5(a), and the results show that when $\sigma = 0.1$ the proposal achieves the best performance on CIFAR-100, and the performance does not degrade significantly with $\sigma$ changes. This demonstrates that our proposal is quite robust with the selection of $\sigma$.

**Evaluating different $\rho$ used in temperature scaling part**. $\rho$ is used for scaling downwards the temperatures on the difficult-to-learn example pairs so that we can eliminate the negative effects of difficult-to-learn examples. We show the performance with different $\rho$ in Figure 5(b), and the results show that when $\rho = 0.7$ the proposal achieves the best performance on CIFAR-100, and the performance does not degrade significantly with $\rho$ changes. We figure out that different values of $\rho$ can all result in performance improvements.

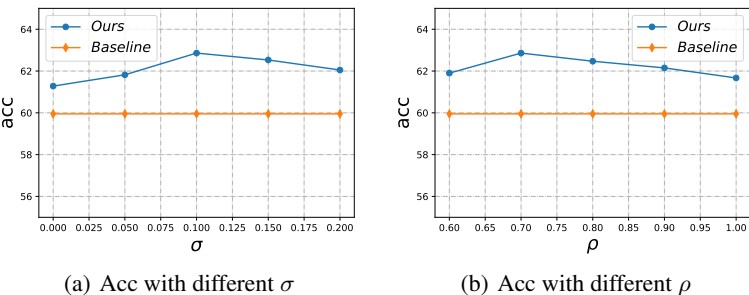

(a) Acc with different $\sigma$          (b) Acc with different $\rho$

Figure 5: (a) Parameter analysis of margin tuning factor $\sigma$,(b) temperature scaling factor $\rho$, all of the above results are implemented on CIFAR-100.

### A.5 FURTHER DISCUSSION

**Which feature is better for difficult-to-learn examples selection?** In SimCLR, the authors found that the proposal of projector $g(\cdot)$ allows the model to learn the auxiliary task better thus having better downstream generalization. However, as mentioned in (Cosentino et al., 2022) they suggest the problem of representation dimensional collapse after using projector, therefore, we here explore

whether it is better to use features before projector $f(x)$ for difficult-to-learn examples selection or $g(f(x))$ after projector.

Table 7: Classification accuracy by using Combined method on CIFAR-10 and CIFAR-100. Features before projector means that we use $f(x)$ for difficult-to-learn examples selection and features after projector means that we use $g(f(x))$ for difficult-to-learn examples selection.

| Features | Baseline | After projector | Before projector |
|---|---|---|---|
| CIFAR-10 | 88.26 | 87.86 | **89.68** |
| CIFAR-100 | 59.95 | 60.63 | **62.86** |

As shown in Table 7, We find that when using $f(x)$ rather than $g(f(x))$ for difficult-to-learn examples selection we can gain a 2.1% performance improvement on CIFAR-10 and a 3.7% performance improvement on CIFAR-100. These results suggest that utilizing features before projector is more beneficial for difficult-to-learn examples selection.

**The combined method is also effective for the Mixed CIFAR-10 datasets.** As we discussed earlier, the Mixed CIFAR-10 datasets contain a large number of mixed difficult-to-learn samples, making the learning difficulty of this dataset significantly greater than that of the original dataset. Based on this fact, this section explores whether our proposed method can achieve performance improvements on the Mixed CIFAR-10 datasets that are consistent with those on CIFAR-100, Tiny ImageNet, and other datasets. We use the 10%- and 20%-Mixed CIFAR-10 datasets as our baselines, while the 0%-Mixed CIFAR-10 datasets serve as our standard CIFAR-10 baseline. The experimental results are shown in Figure 6. We found that

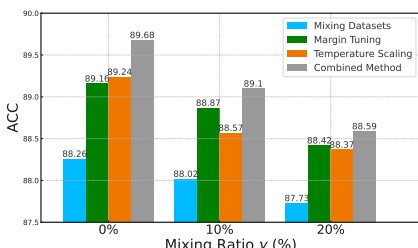

Figure 6: Detailed experimental results on the Mixed CIFAR datasets.

using either margin tuning or temperature scaling alone can improve performance over the original baseline, while the combined method yields better results than using either approach individually. This finding is consistent with the experimental results on other datasets and further validates the effectiveness of our method.

**The proposal is effective for real-world datasets.** We evaluated our method on the Imagenet-1k dataset, which contains 1,000 categories and 1,281,167 training samples. We used ResNet18 as our backbone, set the batch size to 256, and resized each image to 96x96. We set the learning rate to 0.5 and the base temperature to 0.1. We chose 0.01 as the posHigh and 0.5 as the posLow. We set $\sigma$ to 0.1 and $\rho$ to 0.7. We also evaluated the models using linear probing. When evaluating, we set the batch size to 256 and the learning rate to 1. The specific results are shown in Table 8.

Table 8: Classification accuracy on Imagenet-1k.

| Methods | Baseline | Removing | Temperature Scaling | Margin Tuning | Combined |
|---|---|---|---|---|---|
| Accuracy | 34.21 | 34.33 | 35.02 | 35.17 | 35.68 |

From the results on the real-world dataset, Imagenet-1k, which contains more categories, We can see that even after running for only 100 epochs, our method achieves a performance improvement trend consistent with the results mentioned in the paper, compared to the baseline method. These results strengthen the findings and demonstrate broader applicability of this paper.

**Focusing on difficult-to-learn examples and removing them are both effective methods.** We use temperature scaling as an example to illustrate how we should handle difficult-to-learn examples. We note that placing greater emphasis on difficult-to-learn examples (by selecting a smaller temperature) and discarding this sample (which is effectively equivalent to setting the temperature to infinity (we use a large value of 1,000,000,000 to approximate infinity here)) may seem contradictory. However, as shown in Table 9, both approaches are indeed valid. This means that effectively handling difficult-to-learn samples is possible under sufficiently good conditions, while in the absence of such mechanisms, simply discarding them can also be effective.

Table 9: Classification accuracy with various temperature scaling factors on CIFAR-100 datasets. Setting the Temperature Scaling Factor to 0.7 represents using our proposed theoretical framework to specifically address difficult-to-learn samples, while setting the Temperature Scaling Factor to 1e9 means discarding these difficult-to-learn samples. Results are averaged over three runs.

| Temperature Scaling Factor | 0.7 | 1 | 10 | 100 | 1000 | 1e9 |
|---|---|---|---|---|---|---|
| Accuracy | 61.67 | 59.95 | 59.63 | 59.82 | 60.05 | 60.31 |

**The scalability of our proposal under other contrastive learning paradigms.** As mentioned in (Johnson et al., 2022), InfoNCE and Spectral contrastive loss share the same population minimum with variant kernel derivations. By using similar techniques of positive-pair kernel, our conclusions can also be further generalized to other self-supervised learning frameworks. To demonstrate the scalability of the combined method, we supplement the comparative experiments based on the MoCo (Chen et al., 2020b) algorithm. The experimental results demonstrate consistent improvements of our method over both MoCo and SimCLR and show the scalability of our proposal under different contrastive learning paradigms.

**Connection between difficult-to-learn examples and long-tailed distribution.** Under the definition that difficult-to-learn examples contribute least to contrastive learning and that are consequently difficult to distinguish by contrastive learning models, we can easily draw the following conclusion: difficult-to-learn examples can lead to unclear classification boundaries for the classes they belong to.

Due to the significant difference in the number of samples in the head and tail classes, the boundary of tail classes is difficult to be accurately estimated due to the tail classes are prone to collapse when the data is distributed with long-tailed distribution, as mentioned in (Samuel & Chechik, 2021). In other words, tail classes can lead to unclear classification boundaries for the classes they belong to as mentioned in (Fang et al., 2021).

So in this view, tail classes samples can also be seen as difficult-to-learn samples. To better illustrate this point, we will further validate the connection between them through the following experiments. We validate our proposed Combined method on the classical long-tailed distribution dataset tiny-Imagenet-LT to explore whether our proposed algorithm can achieve a performance improvement over the comparison method SimCLR when distributional imbalance as another form of difficult-to-learn samples also exists. The results in Table 6 show that we can achieve better performance when distributional imbalance also exists.

**Analysis of the trending of the derived bounds.** We analyze the trending of the derived bounds on the Mixed CIFAR-10 dataset. Specifically, we vary the mixing ratios from 0% to 30%, where 0% represents the standard CIFAR-10 without mixing. The experimental parameter settings can be referenced to A.3. For each class of samples, we sort them based on the difference between the maximum and second-largest values after applying softmax to the outputs, and select the 8% (the ratio is consistent with what is reported in the paper) smallest differences as the difficult-to-learn examples, as described in the paper. For the calculation of $\alpha$, we take the mean of the similarity between all samples of the same class. For the calculation of $\beta$, we take the mean of the similarity for the sample pairs from different classes that do not contain the difficult-to-learn examples. For the calculation of $\gamma$, we take the mean of the similarity for the sample pairs from different classes that contain the difficult-to-learn examples.

Table 10: The trends of $\alpha$, $\beta$, $\gamma$, and other metrics as the Mixing Ratio changes.

| Mixing Ratio | 0% | 10% | 20% | 30% |
|---|---|---|---|---|
| **acc (%)** | 88.3 | 88.0 | 87.7 | 86.2 |
| $\alpha$ | 47.2 | 44.0 | 41.2 | 38.7 |
| $\beta$ | 19.1 | 19.5 | 20.1 | 20.8 |
| $\gamma$ | 20.9 | 22.1 | 23.1 | 24.1 |
| $\gamma - \beta$ | 1.80 | 2.60 | 3.00 | 3.30 |
| **Eigenvalue** ($\times 10^{-5}$) | 2.93 | 3.36 | 3.58 | 3.72 |

In Table 10, we show that as the mixing ratio increases, the linear probing accuracy drops, and the $(K + 1)$-th eigenvalue increases. Note that the classification error (left hand side of Eq.4) is 1-acc, and the error bound (right hand side of Eq.4) increases with the eigenvalue increasing. This result indicates that as the difficult-to-lean examples increases, the classification error and the bound share the same variation trend, thus validating theorem 3.2 that larger $\gamma - \beta$ results in worse error bound.

**Significance analysis of $\gamma$ and $\beta$.** To verify the significance of $\gamma$ and $\beta$., we tested $\gamma$ and $\beta$, as well as $\gamma - \beta$, on more real datasets. From the first three rows of Table 11, we found that on the CIFAR-100 dataset (which has 10 times more classes than CIFAR-10), the difference between $\gamma$ and $\beta$ remained consistent with that on the CIFAR-10 dataset. On the ImageNet-1k dataset (which has 100 times more classes than CIFAR-10,for specific experimental details and results on ImageNet-1k), the difference between $\gamma$ and $\beta$ was even larger than on CIFAR-10. As a possible intuitive explanation, we conjecture that the higher $\gamma - \beta$ might results from the higher complexity of imagenet images, e.g. different-class images with similar backgrounds can share higher similarity (higher $\gamma$), whereas CIFAR images have relatively simple and consistent backgrounds. These results demonstrate that even on real-world datasets, the difference between $\gamma$ and $\beta$ is significant.

Table 11: Comparison of $\beta$, $\gamma$ $\gamma - \beta$ , t-statistic and P value across different datasets.

| Datasets | CIFAR-10 | CIFAR-100 | Imagenet-1k |
|---|---|---|---|
| $\beta$ | 19.1 | 35.6 | 39.8 |
| $\gamma$ | 20.9 | 37.4 | 42.9 |
| $\gamma - \beta$ | 1.8 | 1.8 | 3.1 |
| $t$-statistic | -502.63 | -539.36 | -3844.21 |
| $P$ value | 0.0 | 0.0 | 0.0 |

To better illustrate the significant difference between $\gamma$ and $\beta$, we conducted an independent samples t-test to support our conclusion. Specifically, we first collected all the $\beta$ and $\gamma$ values, and due to the large sample size, we chose to use Welch's t-test, which does not assume equal variances between the two groups and is suitable for cases where the variances may differ. In the experiment, we focus on two key statistics:

t-statistic: This measures the difference between the means of the two groups relative to the variance within the samples. The t-statistic is a standardized measure used to determine whether the mean difference between the two groups is significant or could be attributed to random fluctuations. The larger the t-statistic, the more significant the difference between the two groups.

P value: The p-value indicates the probability of observing the current difference or more extreme results under the assumption that the null hypothesis (i.e., no significant difference between the two groups) is true. If the p-value is less than 0.05, it suggests that the observed difference is highly unlikely under the null hypothesis, and we can reject the null hypothesis, concluding that there is a significant difference between the two groups.

As shown in the last two rows of Table 11, on all datasets (CIFAR-10, CIFAR-100, Imagenet-1k), the absolute value of the T-statistic is very large, and the P-value is close to zero. This indicates that the mean difference between $\gamma$ and $\beta$ is highly statistically significant.

## B  PROOFS

Recall that in Section 3.2, we introduce the adjacency matrix of the similarity graph based on a 4-sample subset. Here we give the formal definition of the adjacency matrix of a generalized similarity graph containing $|\mathcal{X}| = n(r + 1)$ samples, with $n$ denoting the number of augmented samples per class, and $r + 1$ denoting the number of classes.

Denote $\mathbb{D} = x_1, \ldots, x_{n(r+1)}$ as the dataset, where $x_{n(i-1)+1}, \ldots, x_{ni}$ belong to Class $i$ for $i \in 1, \ldots, r + 1$. Denote $n_d$ as the number of difficult-to-learn examples per class and $\mathbb{D}_d$ as the set of difficult-to-learn examples. Naturally, we denote $n_e := n - n_d$ as the number of easy-to-learn examples per class. Without loss of generality, we assume that the last $n_d$ examples in each class are difficult-to-learn examples. Let $0 \le \beta < \gamma < \alpha < 1$. Then we define the elements of the adjacency

matrix $\boldsymbol{A} = (w_{x,x'})_{x,x' \in \mathcal{X}}$ as $w_{x,x'} := 1$ for $x = x'$; $w_{x,x'} := \alpha$ for $x \neq x'$, $y(x) = y(x')$; $w_{x,x'} := \gamma$ for $x, x' \in \mathbb{D}_d$, $y(x) \neq y(x')$, and $w_{x,x'} := \beta$ otherwise.

Specifically, we have the adjacency matrix of a similarity graph without difficult-to-learn examples as

$$\boldsymbol{A}_{\text{w.o.}} = \begin{bmatrix} \boldsymbol{A}_{\text{same-class}} & \boldsymbol{A}_{\text{different-class}} & \cdots & \boldsymbol{A}_{\text{different-class}} \\ \boldsymbol{A}_{\text{different-class}} & \boldsymbol{A}_{\text{same-class}} & \cdots & \boldsymbol{A}_{\text{different-class}} \\ \vdots & \vdots & & \vdots \\ \boldsymbol{A}_{\text{different-class}} & \boldsymbol{A}_{\text{different-class}} & \cdots & \boldsymbol{A}_{\text{same-class}} \end{bmatrix}_{(r+1) \times (r+1)} \tag{17}$$

and the adjacency matrix of a similarity graph with difficult-to-learn examples as

$$\boldsymbol{A}_{\text{w.d.}} = \begin{bmatrix} \boldsymbol{A}_{\text{same-class}} & \boldsymbol{A}'_{\text{different-class}} & \cdots & \boldsymbol{A}'_{\text{different-class}} \\ \boldsymbol{A}'_{\text{different-class}} & \boldsymbol{A}_{\text{same-class}} & \cdots & \boldsymbol{A}'_{\text{different-class}} \\ \vdots & \vdots & & \vdots \\ \boldsymbol{A}'_{\text{different-class}} & \boldsymbol{A}'_{\text{different-class}} & \cdots & \boldsymbol{A}_{\text{same-class}} \end{bmatrix}_{(r+1) \times (r+1)} \tag{18}$$

where

$$\boldsymbol{A}_{\text{same-class}} = \begin{bmatrix} 1 & \alpha & \cdots & \alpha \\ \alpha & 1 & \cdots & \alpha \\ \cdots & & & \\ \alpha & \alpha & \cdots & 1 \end{bmatrix}_{n \times n}, \tag{19}$$

$$\boldsymbol{A}_{\text{different-class}} = \begin{bmatrix} \beta & \cdots & \beta \\ \vdots & & \vdots \\ \beta & \cdots & \beta \end{bmatrix}_{n \times n}, \tag{20}$$

and

$$\boldsymbol{A}'_{\text{different-class}} = \begin{bmatrix} \beta & \cdots & \beta & \beta & \cdots & \beta \\ \vdots & & \vdots & \vdots & & \vdots \\ \beta & \cdots & \beta & \beta & \cdots & \beta \\ \beta & \cdots & \beta & \gamma & \cdots & \gamma \\ \vdots & & \vdots & \vdots & & \vdots \\ \beta & \cdots & \beta & \gamma & \cdots & \gamma \end{bmatrix}_{(n_e + n_d) \times (n_e + n_d)}. \tag{21}$$

## B.1 PROOFS RELATED TO SECTION 3.3

Before proceeding, we introduce some basic assumptions adapted from HaoChen et al. (2021) to derive the error bounds.

**Assumption B.1** (Labels are recoverable from augmentations). Let $\bar{x} \sim \mathcal{P}_{\bar{\mathcal{X}}}$ and $y(\bar{x})$ be its label. Let the augmentation $x \sim \mathcal{A}(\cdot|\bar{x})$. We assume that there exists a classifier $g$ that can predict $y(\bar{x})$ given $x$ with error at most $\delta$, i.e. $g(x) = y(\bar{x})$ with probability at least $1 - \delta$.

**Assumption B.2** (Realizability). Let $\mathcal{F}$ be a hypothesis class containing functions from $\mathcal{X}$ to $\mathbb{R}^k$. We assume that at least one of the global minima of $\mathcal{L}_{\text{Spec}}$ belongs to $\mathcal{F}$.

Assumption B.1 indicates that labels are recoverable from the augmentations, and Assumption B.2 indicates that the universal minimizer of the population spectral contrastive loss can be realized by the hypothesis class.

*Proof of Theorem 3.1.* For a dataset without difficult-to-learn examples, the similarity between a sample and itself is 1, the similarity between same-class samples is $\alpha$, and the similarity between different-class samples is $\beta$. Then the adjacent matrix of the similarity graph can be decomposed into the sum of several matrix Kronecker products:

$$\boldsymbol{A} = (1 - \alpha)\boldsymbol{I}_{r+1} \otimes \boldsymbol{I}_n + (\alpha - \beta)\boldsymbol{I}_{r+1} \otimes (\mathbf{1}_n \cdot \mathbf{1}_n^\top) + \beta(\mathbf{1}_{r+1} \cdot \mathbf{1}_{r+1}^\top) \otimes (\mathbf{1}_n \cdot \mathbf{1}_n^\top), \tag{22}$$

where $\boldsymbol{I}_{r+1}$ and $\boldsymbol{I}_n$ denote the $(r+1) \times (r+1)$ and $n \times n$ identity matrices respectively, and $\mathbf{1}_{r+1} := (1, \ldots, 1)^\top \in \mathbb{R}^{r+1}$ and $\mathbf{1}_n := (1, \ldots, 1)^\top \in \mathbb{R}^n$ denote the all-one vectors.

First, we calculate the eigenvalues and eigenvectors of $\boldsymbol{A}$. Note that $\boldsymbol{I}_{r+1}$ and $\boldsymbol{I}_n$ have eigenvalues 1 with arbitrary eigenvectors, $\mathbf{1}_n \cdot \mathbf{1}_n^\top$ has eigenvalue $n$ with eigenvector $\bar{\mathbf{1}}_n := \frac{1}{\sqrt{n}} \mathbf{1}_n$ and eigenvalues 0 with eigenvectors $\{\mu : \mu^\top \mathbf{1}_n = 0\}$, and $\mathbf{1}_{r+1} \cdot \mathbf{1}_{r+1}^\top$ has eigenvalue $r+1$ with eigenvector $\bar{\mathbf{1}}_{r+1} := \frac{1}{\sqrt{r+1}} \mathbf{1}_{r+1}$ and eigenvalues 0 with eigenvectors $\{\nu : \nu^\top \mathbf{1}_{r+1} = 0\}$. Therefore, $\boldsymbol{A}$ has the following sets of eigenvalues and eigenvectors:

$$
\begin{aligned}
\lambda_1 &= (1 - \alpha) + n(\alpha - \beta) + n(r+1)\beta, && \text{with eigenvector } \bar{\mathbf{1}}_{r+1} \otimes \bar{\mathbf{1}}_n; \\
\lambda_2 = \ldots = \lambda_{r+1} &= (1 - \alpha) + n(\alpha - \beta), && \text{with eigenvectors } \nu \otimes \bar{\mathbf{1}}_n; \\
\lambda_{r+2} = \ldots = \lambda_{n+r} &= 1 - \alpha, && \text{with eigenvectors } \bar{\mathbf{1}}_{r+1} \otimes u; \\
\lambda_{n+r+1} = \ldots = \lambda_{n(r+1)} &= 1 - \alpha, && \text{with eigenvectors } u \otimes v.
\end{aligned}
$$

Next, we calculate the eigenvalues of $\bar{\boldsymbol{A}} := \boldsymbol{D}^{-1/2} \boldsymbol{A} \boldsymbol{D}^{-1/2}$. By definition, we have $\boldsymbol{D} = \text{diag}(w_1, \ldots, w_{n(r+1)}) = [(1 - \alpha) + n\alpha + nr\beta] \boldsymbol{I}_{n(r+1)}$. Therefore, we have the eigenvalues of $\boldsymbol{A}$ as

$$
\begin{aligned}
\lambda_1 &= 1, \\
\lambda_2 = \ldots = \lambda_{r+1} &= \frac{(1 - \alpha) + n(\alpha - \beta)}{(1 - \alpha) + n\alpha + nr\beta}, \\
\lambda_{r+2} = \ldots = \lambda_{n(r+1)} &= \frac{1 - \alpha}{(1 - \alpha) + n\alpha + nr\beta}.
\end{aligned}
$$

Then according to Theorem B.3 in HaoChen et al. (2021), when $k > r$, there holds

$$
\mathcal{E}_{\text{w.o.}} \le \frac{4\delta}{1 - \lambda_{k+1}} + 8\delta = \frac{4\delta}{1 - \frac{1-\alpha}{(1-\alpha)+n\alpha+nr\beta}} + 8\delta. \tag{23}
$$

$\square$

*Proof of Theorem 3.2.* For a dataset with $n_d$ difficult-to-learn examples per class, the similarity between a sample and itself is 1, the similarity between same-class samples is $\alpha$, the similarity between different-class easy-to-learn samples is $\beta$, and the similarity between different-class hard-to-learn samples is $\gamma$. Without loss of generality, we assume that $n$ is an integral multiple of $n_d$, i.e. there exist a $\kappa \in \mathbb{Z}^+$ such that $n = \kappa n_d$. Then the adjacent matrix of the similarity graph can be decomposed into the sum of several matrix Kronecker products:

$$
\begin{aligned}
\boldsymbol{A} = {}&(1 - \alpha)\boldsymbol{I}_{r+1} \otimes \boldsymbol{I}_n + (\alpha - \beta)\boldsymbol{I}_{r+1} \otimes (\mathbf{1}_n \cdot \mathbf{1}_n^\top) + \beta(\mathbf{1}_{r+1} \cdot \mathbf{1}_{r+1}^\top) \otimes (\mathbf{1}_n \cdot \mathbf{1}_n^\top) \\
&+ (\gamma - \beta)(\mathbf{1}_{r+1} \cdot \mathbf{1}_{r+1}^\top) \otimes (\boldsymbol{e}_\kappa \cdot \boldsymbol{e}_\kappa^\top) \otimes \boldsymbol{I}_{n_d} - (\gamma - \beta)\boldsymbol{I}_{r+1} \otimes (\boldsymbol{e}_\kappa \cdot \boldsymbol{e}_\kappa^\top) \otimes \boldsymbol{I}_{n_d},
\end{aligned} \tag{24}
$$

where $\boldsymbol{I}_{r+1}$, $\boldsymbol{I}_n$, and $\boldsymbol{I}_{n_d}$ denote the $(r+1) \times (r+1)$, $n \times n$, and $n_d \times n_d$ identity matrices respectively, $\mathbf{1}_{r+1} := (1, \ldots, 1)^\top \in \mathbb{R}^{r+1}$ and $\mathbf{1}_n := (1, \ldots, 1)^\top \in \mathbb{R}^n$ denote the all-one vectors, and $\boldsymbol{e}_\kappa := (0, \ldots, 0, 1)^\top \in \mathbb{R}^\kappa$.

Similarly, we can decompose $\boldsymbol{D}$ into

$$
\boldsymbol{D} = \boldsymbol{I}_{r+1} \otimes \Big[ [(1 - \alpha) + n\alpha + nr\beta]\boldsymbol{I}_n + n_d r(\gamma - \beta)(\boldsymbol{e}_\kappa \cdot \boldsymbol{e}_\kappa^\top) \otimes \boldsymbol{I}_{n_d} \Big], \tag{25}
$$

and therefore we have

$$
\boldsymbol{D}^{-1} = \boldsymbol{I}_{r+1} \otimes \Big[ \frac{1}{c_2} [\boldsymbol{I}_\kappa - (\boldsymbol{e}_\kappa \cdot \boldsymbol{e}_\kappa^\top)] + \frac{1}{c_1}(\boldsymbol{e}_\kappa \cdot \boldsymbol{e}_\kappa^\top) \Big] \otimes \boldsymbol{I}_{n_d}, \tag{26}
$$

where we denote $c_1 := (1 - \alpha) + n\alpha + nr\beta + n_d r(\gamma - \beta)$ and $c_2 := (1 - \alpha) + n\alpha + nr\beta$.

Then we have the decomposition of the normalized similarity matrix as

$$\bar{A} = D^{-1/2}AD{-1/2}$$

$$= (1-\alpha)I_{r+1} \otimes \left[\frac{1}{c_2}[I_\kappa - (e_\kappa \cdot e_\kappa^\top)] + \frac{1}{c_1}(e_\kappa \cdot e_\kappa^\top)\right] \otimes I_{n_d}$$

$$+ (\gamma-\beta)(1_{r+1} \cdot 1_{r+1}^\top) \otimes \frac{1}{c_1}(e_\kappa \cdot e_\kappa^\top) \otimes I_{n_d}$$

$$- (\gamma-\beta)I_{r+1} \otimes \frac{1}{c_1}(e_\kappa \cdot e_\kappa^\top) \otimes I_{n_d}.$$

$$+ (\alpha-\beta)I_{r+1} \otimes \left[[\frac{1}{\sqrt{c_2}}(1_\kappa - e_\kappa) + \frac{1}{\sqrt{c_1}}e_\kappa] \cdot [\frac{1}{\sqrt{c_2}}(1_\kappa - e_\kappa) + \frac{1}{\sqrt{c_1}}e_\kappa]^\top\right] \otimes (1_{n_d} \cdot 1_{n_d}^\top)$$

$$+ \beta(1_{r+1} \cdot 1_{r+1}^\top) \otimes \left[[\frac{1}{\sqrt{c_2}}(1_\kappa - e_\kappa) + \frac{1}{\sqrt{c_1}}e_\kappa] \cdot [\frac{1}{\sqrt{c_2}}(1_\kappa - e_\kappa) + \frac{1}{\sqrt{c_1}}e_\kappa]^\top\right] \otimes (1_{n_d} \cdot 1_{n_d}^\top)$$

$$= \frac{1}{c_2}(1-\alpha)I_{r+1} \otimes I_\kappa \otimes I_{n_d}$$

$$+ \frac{1}{c_1}(\gamma-\beta)(1_{r+1} \cdot 1_{r+1}^\top) \otimes (e_\kappa \cdot e_\kappa^\top) \otimes I_{n_d}$$

$$- \left[\frac{1}{c_1}(\gamma-\beta) + (\frac{1}{c_2} - \frac{1}{c_1})(1-\alpha)\right]I_{r+1} \otimes (e_\kappa \cdot e_\kappa^\top) \otimes I_{n_d}.$$

$$+ (\alpha-\beta)I_{r+1} \otimes \left[[\frac{1}{\sqrt{c_2}}(1_\kappa - e_\kappa) + \frac{1}{\sqrt{c_1}}e_\kappa] \cdot [\frac{1}{\sqrt{c_2}}(1_\kappa - e_\kappa) + \frac{1}{\sqrt{c_1}}e_\kappa]^\top\right] \otimes (1_{n_d} \cdot 1_{n_d}^\top)$$

$$+ \beta(1_{r+1} \cdot 1_{r+1}^\top) \otimes \left[[\frac{1}{\sqrt{c_2}}(1_\kappa - e_\kappa) + \frac{1}{\sqrt{c_1}}e_\kappa] \cdot [\frac{1}{\sqrt{c_2}}(1_\kappa - e_\kappa) + \frac{1}{\sqrt{c_1}}e_\kappa]^\top\right] \otimes (1_{n_d} \cdot 1_{n_d}^\top).$$

$$(27)$$

Now we calculate the eigenvalues and eigenvectors of $A$. For notational simplicity, we denote the first three terms of equation 27 as $\bar{A}_1$ and the last two terms as $\bar{A}_2$. Note that $I_{r+1}$, $I_\kappa$, and $I_{n_d}$ have eigenvalues 1 with arbitrary eigenvectors, $1_{r+1} \cdot 1_{r+1}^\top$ has eigenvalue $r+1$ with eigenvector $\bar{1}_{r+1} := \frac{1}{\sqrt{r+1}}1_{r+1}$ and eigenvalues 0 with eigenvectors $\{\nu : \nu^\top 1_{r+1} = 0\}$, and $e_\kappa \cdot e_\kappa^\top$ has eigenvalue 1 with eigenvector $e_1 = (1, 0, \ldots, 0)^\top \in \mathbb{R}^\kappa$ and eigenvalues 0 with eigenvectors $\{e_2, \ldots, e_\kappa\}$. Let $\xi \in \mathbb{R}^{n_d}$ denote an arbitrary vector. Then $\bar{A}_1$ has the following sets of eigenvalues and eigenvectors:

$$\lambda_{1,1} = \ldots = \lambda_{1,n_d} = \frac{1}{c_2}(1-\alpha) + \frac{1}{c_1}(\gamma-\beta)(r+1) - \left[\frac{1}{c_1}(\gamma-\beta) + (\frac{1}{c_2} - \frac{1}{c_1})(1-\alpha)\right],$$

$$= \frac{1}{c_1}(1-\alpha) + \frac{r}{c_1}(\gamma-\beta), \qquad \text{with eigenvectors } \bar{1}_{r+1} \otimes e_1 \otimes \xi;$$

$$\lambda_{1,n_d+1} = \ldots = \lambda_{1,n} = \frac{1}{c_2}(1-\alpha), \qquad \text{with eigenvectors } \bar{1}_{r+1} \otimes e_i \otimes \xi, i = 2, \ldots, \kappa;$$

$$\lambda_{1,n+1} = \ldots = \lambda_{1,(r+1)n-rn_d} = \frac{1}{c_2}(1-\alpha), \qquad \text{with eigenvectors } \nu \otimes e_i \otimes \xi, i = 2, \ldots, \kappa;$$

$$\lambda_{1,(r+1)n-rn_d+1} = \ldots = \lambda_{1,(r+1)n} = \frac{1}{c_2}(1-\alpha) - \left[\frac{1}{c_1}(\gamma-\beta) + (\frac{1}{c_2} - \frac{1}{c_1})(1-\alpha)\right],$$

$$= \frac{1}{c_1}(1-\alpha) - \frac{1}{c_1}(\gamma-\beta), \qquad \text{with eigenvectors } \nu \otimes e_1 \otimes \xi.$$

On the other hand, note that $1_{n_d} \cdot 1_{n_d}^\top$ has eigenvalue $n_d$ with eigenvector $\bar{1}_{n_d} := \frac{1}{\sqrt{n_d}}1_{n_d}$ and eigenvalues 0 with eigenvectors $\{\eta : \eta^\top 1_{n_d} = 0\}$, and that by calculations, $[\frac{1}{\sqrt{c_2}}(1_\kappa - e_\kappa) + \frac{1}{\sqrt{c_1}}e_\kappa] \cdot [\frac{1}{\sqrt{c_2}}(1_\kappa - e_\kappa) + \frac{1}{\sqrt{c_1}}e_\kappa]^\top$ has eigenvalue $\frac{\kappa-1}{c_2} + \frac{1}{c_1}$ with eigenvector $\{\eta : \sum_{i=1}^{\kappa-1}\eta_i = 0, \eta_\kappa = (\kappa-1)\sqrt{c_1/c_2}\}$ and eigenvalues 0 with eigenvectors $\{\theta : \sum_{i=1}^{\kappa-1}\theta_i = 0, \eta_\kappa = 0\}$. Then $\bar{A}_2$ has the

following sets of eigenvalues and eigenvectors:

$$\lambda_{2,1} = (\alpha - \beta)\Big[\frac{\kappa - 1}{c_2} + \frac{1}{c_1}\Big]n_d + \beta(r+1)\Big[\frac{\kappa - 1}{c_2} + \frac{1}{c_1}\Big]n_d,$$

$$= (\alpha + r\beta)\Big[\frac{\kappa - 1}{c_2} + \frac{1}{c_1}\Big]n_d, \qquad \text{with eigenvectors } \bar{\mathbf{1}}_{r+1} \otimes \eta \otimes \bar{\mathbf{1}}_{n_d};$$

$$\lambda_{2,2} = \ldots = \lambda_{2,r+1} = (\alpha - \beta)\Big[\frac{\kappa - 1}{c_2} + \frac{1}{c_1}\Big]n_d, \qquad \text{with eigenvectors } \nu \otimes \eta \otimes \bar{\mathbf{1}}_{n_d};$$

$$\lambda_{2,r+2} = \ldots = \lambda_{2,(r+1)n} = 0, \qquad \text{with other combinations of eigenvectors.}$$

By Equation 13 in Fulton (2000), for two real symmetric $n(r+1) \times n(r+1)$ matrices $\bar{A}_1$ and $\bar{A}_2$, we have the $k+1$-th largest eigenvalue of $\bar{A} := \bar{A}_1 + \bar{A}_2$ satisfies

$$\lambda_{k+1} \leq \min_{i+j=k+2} \lambda_{1,i} + \lambda_{2,j}$$

$$= \begin{cases} \frac{1}{c_1}(1-\alpha) + \frac{r}{c_1}(\gamma - \beta) + (\alpha - \beta)\Big[\frac{\kappa - 1}{c_2} + \frac{1}{c_1}\Big]n_d, & \text{for } k < r+1, \\[2mm] \min\Big\{\frac{1}{c_1}(1-\alpha) + \frac{r}{c_1}(\gamma - \beta), \frac{1}{c_2}(1-\alpha) + (\alpha - \beta)\Big[\frac{\kappa - 1}{c_2} + \frac{1}{c_1}\Big]n_d\Big\} \\[2mm] = \frac{1}{c_1}(1-\alpha) + \frac{r}{c_1}(\gamma - \beta), & \text{for } r+1 \leq k < n_d + r + 1. \end{cases}$$

Then according to Theorem B.3 in HaoChen et al. (2021), when $r+1 \leq k < n_d + r + 1$, there holds

$$\mathcal{E}_{\text{w.d.}} \leq \frac{4\delta}{1 - \lambda_{k+1}} + 8\delta = \frac{4\delta}{1 - \frac{1}{c_1}(1-\alpha) - \frac{r}{c_1}(\gamma - \beta)} + 8\delta = \frac{4\delta}{1 - \frac{(1-\alpha)+r(\gamma-\beta)}{(1-\alpha)+n\alpha+nr\beta+n_dr(\gamma-\beta)}} + 8\delta. \tag{28}$$

$\square$

## B.2 Proofs Related to Section 4

*Proof of Corollary 4.1.* By removing the difficult-to-learn examples, we have the adjacency matrix as

$$A = \begin{bmatrix} A_{\text{same-class}} & A_{\text{different-class}} & \cdots & A_{\text{different-class}} \\ A_{\text{different-class}} & A_{\text{same-class}} & \cdots & A_{\text{different-class}} \\ \vdots & \vdots & & \vdots \\ A_{\text{different-class}} & A_{\text{different-class}} & \cdots & A_{\text{same-class}} \end{bmatrix}_{(r+1)\times(r+1)}, \tag{29}$$

where

$$A_{\text{different-class}} = \begin{bmatrix} \beta & \cdots & \beta \\ \vdots & & \vdots \\ \beta & \cdots & \beta \end{bmatrix}_{n_e \times n_e}. \tag{30}$$

Then the matrix $A$ reduces to $A_{\text{w.o.}}$ and the error bound reduces to that in Theorem 3.1 with $n$ replaced with $n_e = n - n_d$. $\square$

The spectral contrastive loss with a margin $M = (m_{x,x'})$ is defined as

$$\mathcal{L}_{\text{M}}(\boldsymbol{x}; f) = -2\mathbb{E}_{x,x^+} f(x)^\top f(x^+) + \mathbb{E}_{x,x'}\Big[f(x)^\top f(x') + m_{x,x'}\Big]^2. \tag{31}$$

*Proof of Theorem 4.2.*

$$\mathcal{L}_{\mathrm{M}} = -2\mathbb{E}_{x,x^+} f(x)^\top f(x^+) + \mathbb{E}_{x,x'}\Big[f(x)^\top f(x') + m_{x,x'}\Big]^2$$

$$= -2\sum_{x,x^+} w_{x,x'} f(x)^\top f(x^+) + \sum_{x,x'} w_x w_{x'}\Big[f(x)^\top f(x') + m_{x,x'}\Big]^2$$

$$= \sum_{x,x'}\Big\{ -2w_{x,x'} f(x)^\top f(x') + w_x w_{x'}\Big[f(x)^\top f(x')\Big]^2 + 2w_x w_{x'} m_{x,x'} f(x)^\top f(x') + w_x w_{x'} m_{x,x'}^2 \Big\}$$

$$= \sum_{x,x'}\Big\{ w_x w_{x'}\Big[f(x)^\top f(x')\Big]^2 - 2[w_{x,x'} - w_x w_{x'} m_{x,x'}]f(x)^\top f(x') + w_x w_{x'} m_{x,x'}^2 \Big\}$$

$$= \sum_{x,x'}\Big\{ \Big[[\sqrt{w_x}f(x)]^\top[\sqrt{w_{x'}}f(x')]\Big]^2 - 2\Big[\frac{w_{x,x'}}{\sqrt{w_x}\sqrt{w_{x'}}} - \sqrt{w_x}\sqrt{w_{x'}}m_{x,x'}\Big][\sqrt{w_x}f(x)]^\top[\sqrt{w_{x'}}f(x')]$$

$$\qquad + \Big[\frac{w_{x,x'}}{\sqrt{w_x}\sqrt{w_{x'}}} - \sqrt{w_x}\sqrt{w_{x'}}m_{x,x'}\Big]^2 + 2w_{x,x'}m_{x,x'} - \frac{w_{x,x'}^2}{w_x w_{x'}}\Big\}$$

$$= \sum_{x,x'}\Big[\frac{w_{x,x'}}{\sqrt{w_x}\sqrt{w_{x'}}} - \sqrt{w_x}\sqrt{w_{x'}}m_{x,x'} - [\sqrt{w_x}f(x)]^\top[\sqrt{w_{x'}}f(x')]\Big]^2 + \sum_{x,x'}\Big(2w_{x,x'}m_{x,x'} - \frac{w_{x,x'}^2}{w_x w_{x'}}\Big)$$

$$:= \|(\bar{\boldsymbol{A}} - \bar{\boldsymbol{M}}) - FF^\top\|_F^2 + \sum_{x,x'}\Big(2w_{x,x'}m_{x,x'} - \frac{w_{x,x'}^2}{w_x w_{x'}}\Big), \tag{32}$$

where we denote $\bar{\boldsymbol{A}} := \boldsymbol{D}^{-1/2}\boldsymbol{A}\boldsymbol{D}^{-1/2}$, $\bar{\boldsymbol{M}} := \boldsymbol{D}^{1/2}\boldsymbol{M}\boldsymbol{D}^{1/2}$, $\boldsymbol{A} := (w_{x,x'})_{x,x'\in\{x_i\}_{i=1}^{n(r+1)}}$, $\boldsymbol{M} := (m_{x,x'})_{x,x'\in\{x_i\}_{i=1}^{n(r+1)}}$, $\boldsymbol{D} := \mathrm{diag}(w_1,\ldots,w_{n(r+1)})$, and $F = (\sqrt{w_x}f(x))_{x\in\{x_i\}_{i=1}^{n(r+1)}}$.

Note that given the adjacency matrix of the similarity graph $\boldsymbol{A}$ and the margin matrix $\boldsymbol{M}$, the second term in equation 32 is a constant. Therefore, minimizing the margin tuning loss $\mathcal{L}_{\mathrm{M}}$ over $f(x)$ is equivalent to minimizing the matrix factorization loss $\mathcal{L}_{\mathrm{mf-M}} := \|(\bar{\boldsymbol{A}} - \bar{\boldsymbol{M}}) - FF^\top\|_F^2$ over $F$. □

*Proof of Theorem 4.3.* Recall that when difficult-to-learn examples exist, we assume that

$$w_{x,x'} := \begin{cases} 1 & \text{for } x = x', \\ \alpha & \text{for } x \neq x', y(x) = y(x'), \\ \gamma & \text{for } x, x' \in \mathbb{D}_d, y(x) \neq y(x'), \\ \beta & \text{otherwise.} \end{cases} \tag{33}$$

Then by definition we have

$$w_x = \sum_{x'} w_{x,x'} = \begin{cases} (1-\alpha) + n\alpha + nr\beta + n_d r(\gamma - \beta), & \text{for } x \in \mathbb{D}_d, \\ (1-\alpha) + n\alpha + nr\beta, & \text{for } x \notin \mathbb{D}_d, \end{cases} \tag{34}$$

and correspondingly

$$\frac{w_{x,x'}}{w_x w_{x'}} = \begin{cases} \dfrac{1}{(1-\alpha) + n\alpha + nr\beta + n_d r(\gamma - \beta)}, & \text{for } x = x', x \in \mathbb{D}_d, \\[2mm] \dfrac{1}{(1-\alpha) + n\alpha + nr\beta}, & \text{for } x = x', x \notin \mathbb{D}_d, \\[2mm] \dfrac{\alpha}{(1-\alpha) + n\alpha + nr\beta + n_d r(\gamma - \beta)}, & \text{for } x \neq x', y(x) = y(x'), x, x' \in \mathbb{D}_d, \\[2mm] \dfrac{\alpha}{\sqrt{(1-\alpha) + n\alpha + nr\beta + n_d r(\gamma - \beta)}\sqrt{(1-\alpha) + n\alpha + nr\beta}}, & \text{for } x \neq x', y(x) = y(x'), x \in \mathbb{D}_d \text{ or } x' \in \mathbb{D}_d, \\[2mm] \dfrac{\alpha}{(1-\alpha) + n\alpha + nr\beta}, & \text{for } x \neq x', y(x) = y(x'), x, x' \notin \mathbb{D}_d, \\[2mm] \dfrac{\gamma}{(1-\alpha) + n\alpha + nr\beta + n_d r(\gamma - \beta)}, & \text{for } y(x) \neq y(x'), x, x' \in \mathbb{D}_d, \\[2mm] \dfrac{\beta}{\sqrt{(1-\alpha) + n\alpha + nr\beta + n_d r(\gamma - \beta)}\sqrt{(1-\alpha) + n\alpha + nr\beta}}, & \text{for } y(x) \neq y(x'), x \in \mathbb{D}_d \text{ or } x' \in \mathbb{D}_d, \\[2mm] \dfrac{\beta}{(1-\alpha) + n\alpha + nr\beta}, & \text{for } y(x) \neq y(x'), x, x' \notin \mathbb{D}_d, \end{cases} \tag{35}$$

If we let

$$m_{x,x'} = \begin{cases} -\dfrac{n_d r(\gamma - \beta)}{[(1-\alpha) + n\alpha + nr\beta + n_d r(\gamma - \beta)]^2[(1-\alpha) + n\alpha + nr\beta]}, & \text{for } x = x', x \in \mathbb{D}_d, \\[3mm] -\dfrac{n_d r(\gamma - \beta)}{[(1-\alpha) + n\alpha + nr\beta + n_d r(\gamma - \beta)]^2[(1-\alpha) + n\alpha + nr\beta]}\alpha, & \text{for } x \neq x', y(x) = y(x'), x, x' \in \mathbb{D}_d, \\[3mm] -\dfrac{\frac{\sqrt{(1-\alpha)+n\alpha+nr\beta+n_d r(\gamma-\beta)}}{\sqrt{(1-\alpha)+n\alpha+nr\beta}} - 1}{[(1-\alpha) + n\alpha + nr\beta + n_d r(\gamma - \beta)][(1-\alpha) + n\alpha + nr\beta]}\alpha, & \text{for } x \neq x', y(x) = y(x'), x \in \mathbb{D}_d \text{ or } x' \in \mathbb{D}_d, \\[3mm] \dfrac{[(1-\alpha) + n\alpha + (n - n_d)r\beta](\gamma - \beta)}{[(1-\alpha) + n\alpha + nr\beta + n_d(\gamma - \beta)]^2[(1-\alpha) + n\alpha + nr\beta]}, & \text{for } y(x) \neq y(x'), x, x' \in \mathbb{D}_d, \\[3mm] -\dfrac{\frac{\sqrt{(1-\alpha)+n\alpha+nr\beta+n_d r(\gamma-\beta)}}{\sqrt{(1-\alpha)+n\alpha+nr\beta}} - 1}{[(1-\alpha) + n\alpha + nr\beta + n_d r(\gamma - \beta)][(1-\alpha) + n\alpha + nr\beta]}\beta, & \text{for } y(x) \neq y(x'), x \in \mathbb{D}_d \text{ or } x' \in \mathbb{D}_d, \\[3mm] 0 & \text{otherwise}, \end{cases} \tag{36}$$

then we have

$$\sqrt{w_x}\sqrt{w_{x'}}m_{x,x'}$$
$$= \begin{cases} -\dfrac{n_d r(\gamma - \beta)}{[(1-\alpha) + n\alpha + nr\beta + n_d r(\gamma - \beta)][(1-\alpha) + n\alpha + nr\beta]}, & \text{for } x = x', x \in \mathbb{D}_d, \\[3mm] -\dfrac{n_d r(\gamma - \beta)}{[(1-\alpha) + n\alpha + nr\beta + n_d r(\gamma - \beta)][(1-\alpha) + n\alpha + nr\beta]}\alpha, & \text{for } x \neq x', y(x) = y(x'), x, x' \in \mathbb{D}_d, \\[3mm] -\dfrac{\sqrt{(1-\alpha) + n\alpha + nr\beta + n_d r(\gamma - \beta)} - \sqrt{(1-\alpha) + n\alpha + nr\beta}}{\sqrt{(1-\alpha) + n\alpha + nr\beta + n_d r(\gamma - \beta)}[(1-\alpha) + n\alpha + nr\beta]}\alpha, & \text{for } x \neq x', y(x) = y(x'), x \in \mathbb{D}_d \text{ or } x' \in \mathbb{D}_d, \\[3mm] \dfrac{[(1-\alpha) + n\alpha + (n - n_d)r\beta](\gamma - \beta)}{[(1-\alpha) + n\alpha + nr\beta + n_d(\gamma - \beta)][(1-\alpha) + n\alpha + nr\beta]}, & \text{for } y(x) \neq y(x'), x, x' \in \mathbb{D}_d, \\[3mm] -\dfrac{\sqrt{(1-\alpha) + n\alpha + nr\beta + n_d r(\gamma - \beta)} - \sqrt{(1-\alpha) + n\alpha + nr\beta}}{\sqrt{(1-\alpha) + n\alpha + nr\beta + n_d r(\gamma - \beta)}[(1-\alpha) + n\alpha + nr\beta]}\beta, & \text{for } y(x) \neq y(x'), x \in \mathbb{D}_d \text{ or } x' \in \mathbb{D}_d, \\[3mm] 0 & \text{otherwise}, \end{cases} \tag{37}$$

and accordingly

$$
\frac{w_{x,x'}}{w_x w_{x'}} - \sqrt{w_x}\sqrt{w_{x'}} m_{x,x'} = \begin{cases} \dfrac{1}{(1-\alpha) + n\alpha + nr\beta} & \text{for } x = x', \\[2mm] \dfrac{\alpha}{(1-\alpha) + n\alpha + nr\beta} & \text{for } x \neq x', y(x) = y(x'), \\[2mm] \dfrac{\beta}{(1-\alpha) + n\alpha + nr\beta} & \text{otherwise.} \end{cases} \tag{38}
$$

In this case, $\bar{A} - \bar{M}$ is equivalent to the normalized similarity matrix of data without difficult-to-learn examples. That is, we have

$$
\mathcal{E}_{\mathrm{M}} = \mathcal{E}_{\mathrm{w.o.}}. \tag{39}
$$

$\square$

The spectral contrastive loss with temperature $T = (\tau_{x,x'})$ is defined as

$$
\mathcal{L}_{\mathrm{T}}(\boldsymbol{x}; f) = -2\mathbb{E}_{x,x^+} \frac{f(x)^\top f(x^+)}{\tau_{x,x^+}} + \mathbb{E}_{x,x'}\left[\frac{f(x)^\top f(x')}{\tau_{x,x'}}\right]^2. \tag{40}
$$

*Proof of Theorem 4.4.*

$$
\mathcal{L}_{\mathrm{T}} = \mathbb{E}_{x,x^+} f(x)^\top f(x^+)/\tau_{x,x^+} + \mathbb{E}_{x,x'}\left[f(x)^\top f(x')/\tau_{x,x'}\right]^2
$$

$$
= -2\sum_{x,x^+} w_{x,x'} f(x)^\top f(x^+)/\tau_{x,x^+} + \sum_{x,x'} w_x w_{x'}\left[f(x)^\top f(x')/\tau_{x,x'}\right]^2
$$

$$
= \sum_{x,x'}\left\{-2 w_{x,x'}/\tau_{x,x'} f(x)^\top f(x^+) + w_x w_{x'}/\tau_{x,x'}^2\left[f(x)^\top f(x')/\tau_{x,x'}\right]^2\right\}
$$

$$
= \sum_{x,x'}\left\{-2\frac{1}{\tau_{x,x'}}\frac{w_{x,x'}}{\sqrt{w_x}\sqrt{w_{x'}}}[\sqrt{w_x}f(x)]^\top[\sqrt{w_{x'}}f(x')] + \frac{1}{\tau_{x,x'}^2}\left[[\sqrt{w_x}f(x)]^\top[\sqrt{w_{x'}}f(x')]\right]^2\right\}
$$

$$
= \sum_{x,x'}\frac{1}{\tau_{x,x'}^2}\left\{\left[[\sqrt{w_x}f(x)]^\top[\sqrt{w_{x'}}f(x')]\right]^2 - 2\frac{\tau_{x,x'} w_{x,x'}}{\sqrt{w_x}\sqrt{w_{x'}}}[\sqrt{w_x}f(x)]^\top[\sqrt{w_{x'}}f(x')] + \frac{\tau_{x,x'}^2 w_{x,x'}^2}{w_x w_{x'}} - \frac{\tau_{x,x'}^2 w_{x,x'}^2}{w_x w_{x'}}\right\}
$$

$$
= \sum_{x,x'}\frac{1}{\tau_{x,x'}^2}\left[\tau_{x,x'}\frac{w_{x,x'}}{\sqrt{w_x}\sqrt{w_{x'}}} - [\sqrt{w_x}f(x)]^\top[\sqrt{w_{x'}}f(x')]\right]^2 - \frac{1}{\tau_{x,x'}^2}\sum_{x,x'}\frac{\tau_{x,x'}^2 w_{x,x'}^2}{w_x w_{x'}}
$$

$$
:= \|\boldsymbol{T} \odot \bar{A} - FF^\top\|_{wF}^2 - \frac{1}{\tau_{x,x'}^2}\sum_{x,x'}\frac{\tau_{x,x'}^2 w_{x,x'}^2}{w_x w_{x'}}, \tag{41}
$$

where we denote $\boldsymbol{T} := (\tau_{x,x'})_{x,x' \in \{x_i\}_{i=1}^{n(r+1)}}$, $\bar{A} := \boldsymbol{D}^{-1/2} \boldsymbol{A} \boldsymbol{D}^{-1/2}$, $\boldsymbol{A} := (w_{x,x'})_{x,x' \in \{x_i\}_{i=1}^{n(r+1)}}$, $\boldsymbol{D} := \mathrm{diag}(w_1, \ldots, w_{n(r+1)})$, $F = (\sqrt{w_x}f(x))_{x \in \{x_i\}_{i=1}^{n(r+1)}}$, $\boldsymbol{T} \odot \bar{A}$ as the element-wise product of matrices $\boldsymbol{T}$ and $\bar{A}$, and $\|\cdot\|_{wF}$ as the weighted Frobenius norm where $\|\boldsymbol{B}\|_{wF}^2 := \sum_{x,x'}\frac{1}{\tau_{x,x'}^2} b_{x,x'}^2$ for arbitrary matrix $\boldsymbol{B} = (b_{x,x'}) \in \mathbb{R}^{n(r+1) \times n(r+1)}$.

Note that given the adjacency matrix of the similarity graph $\boldsymbol{A}$ and the temperature matrix $\boldsymbol{T}$, the second term in equation 41 is a constant. Therefore, minimizing the temperature scaling loss $\mathcal{L}_{\mathrm{T}}$ over $f(x)$ is equivalent to minimizing the matrix factorization loss $\mathcal{L}_{\mathrm{mf-T}} := \|\boldsymbol{T} \odot \bar{A} - FF^\top\|_{wF}^2$ over $F$. $\square$

Before we proceed to the proof of Theorem 4.5, we first extend Theorem B.3 in HaoChen et al. (2021) to the temperature scaling loss by deriving the matrix factorization error bound under the weighted Frobenius norm.

**Lemma B.3.** *Let $f_{\text{pop}}^* \in \arg\min_{f:\mathcal{X}\to\mathbb{R}^k} \mathcal{L}_{\text{T}}(f)$ be a minimizer of the population temperature-scaling loss $\mathcal{L}_{\text{T}}(f)$. Then for any labeling function $\hat{y} : \mathcal{X} \to [r]$, there exists a linear probe $B^* \in \mathbb{R}^{r\times k}$ with norm $\|B^*\|_F \leq 1/(1-\lambda_k)$ such that*

$$\mathbb{E}_{\bar{x}\sim\mathcal{P}_{\bar{X}},x\sim\mathcal{A}(\cdot|\bar{x})}\Big[\|\vec{y} - B^* f_{\text{pop}}^*(x)\|_2^2\Big] \leq \frac{\tilde{\phi}^{\hat{y}}}{1-\lambda_{k+1}} + 4\Delta(y,\hat{y}), \tag{42}$$

*where $\vec{y}(\bar{x})$ is the one-hot embedding of $y(\bar{x})$, and*

$$\tilde{\phi}^{\hat{y}} = \sum_{x,x'\sim\mathcal{X}} \frac{w_{x,x'}}{\tau_{x,x'}^2}\mathbf{1}[\hat{y}(x) \neq \hat{y}(x')]. \tag{43}$$

*Furthermore, the error can be bounded by*

$$\mathcal{E}_{\text{T}} = \Pr_{\bar{x}\sim\mathcal{P}_{\bar{X}},x\sim\mathcal{A}(\cdot|\bar{x})}\Big(g_{f_{\text{pop}^*},B^*}(x) \neq y(\bar{x})\Big) \leq \frac{2\tilde{\phi}^{\hat{y}}}{1-\lambda_{k+1}} + 8\Delta(y,\hat{y}). \tag{44}$$

We also need the following two supporting lemmas to prove Lemma B.3.

**Lemma B.4.** *Let $L$ be the normalized Laplacian matrix of some graph $G$, $v_i$ be the $i$-th smallest unit-norm eigenvector of $\boldsymbol{L}$ with eigenvalue $1 - \lambda_i$, and $\tilde{R}(u) := \frac{\tilde{u}^\top \boldsymbol{L}\tilde{u}}{u^\top u}$ for a vector $u \in \mathbb{R}^N$, where $\tilde{u} = (u_i/\tau_i)_{i=1}^N$. Then for any $k \in \mathbb{Z}^+$ such that $k < N$ and $1 - \lambda_{k+1} > 0$, there exists a vector $b \in \mathbb{R}^k$ with norm $\|b\|_2 \leq \|u\|_2$ such that*

$$\Big\|u - \sum_{i=1}^k b_i v_i\Big\|_w^2 \leq \frac{\tilde{R}(u)}{1-\lambda_{k+1}}\|u\|_2^2, \tag{45}$$

*where $\|\cdot\|$ denotes the weighted $l^2$-norm with weights $\tau^{-2} = (1/\tau_i^2)_{i=1}^N$.*

*Proof of Lemma B.4.* We can decompose the vector $u$ in the eigenvector basis as

$$u = \sum_{i=1}^N \zeta_i v_i. \tag{46}$$

Let $b \in \mathbb{R}^k$ be the vector such that $b_i = \zeta_i$. Then we have $\|b\|_2^2 \leq \|u\|_2^2$ and

$$\begin{aligned}
\Big\|u - \sum_{i=1}^k b_i v_i\Big\|_w^2 &= \|\sum_{i=k+1}^N \zeta_i v_i\|_w^2 \\
&= \sum_{i=k+1}^N \zeta_i^2/\tau_i^2 \\
&\leq \frac{1}{1-\lambda_{k+1}}\sum_{i=k+1}^N (1-\lambda_i)\zeta_i^2/\tau_i^2 \\
&= \frac{1}{1-\lambda_{k+1}}\sum_{i=k+1}^N \zeta_i^2/\tau_i^2 v_i^\top (1-\lambda_i)v_i \\
&= \frac{1}{1-\lambda_{k+1}}\sum_{i=k+1}^N \zeta_i^2/\tau_i^2 v_i^\top \boldsymbol{L}v_i \\
&= \frac{1}{1-\lambda_{k+1}}\sum_{i=k+1}^N (\zeta_i/\tau_i \cdot v_i)^\top \boldsymbol{L}(\zeta_i/\tau_i \cdot v_i). 
\end{aligned} \tag{47}$$

Denote $\tilde{u} = \sum_{i=1}\zeta_i/\tau_i \cdot v_i$ and $\tilde{R}(u) := \frac{\tilde{u}^\top \boldsymbol{L}\tilde{u}}{u^\top u}$. Then we have

$$\Big\|u - \sum_{i=1}^k b_i v_i\Big\|_w^2 \leq \frac{\tilde{R}(u)}{1-\lambda_{k+1}}\|u\|_2^2. \tag{48}$$

$\square$

**Lemma B.5.** *In the setting of Lemma B.4, let $\hat{y}$ be an extended labeling function. Fix $i \in [r]$. Define function $u_i^{\hat{y}}(x) := \sqrt{w_x} \cdot \mathbf{1}[\hat{y}(x) = i]$ and $u_i^{\hat{y}}$ is the corresponding vector in $\mathbb{R}^N$. Also define the following quantity*

$$\tilde{\phi}_i^{\hat{y}} := \frac{\sum_{x,x' \in \mathcal{X}} w_{x,x'}/\tau_{x,x'}^2 \cdot \mathbf{1}[(\hat{y}(x) = i \wedge \hat{y}(x') \neq i) \, or \, (\hat{y}(x) \neq i \wedge \hat{y}(x') = i)]}{\sum_{x \in \mathcal{X}} w_x \cdot \mathbf{1}[\hat{y}(x) = i]}. \tag{49}$$

*Then we have*

$$\tilde{R}(u_i^{\hat{y}}) = \frac{1}{2}\tilde{\phi}_i^{\hat{y}}. \tag{50}$$

*Proof of Lemma B.5.* Let $f$ be any function $\mathcal{X} \to \mathbb{R}$, define function $u(x) := \sqrt{w_x} \cdot f(x)$. Let $u \in \mathbb{R}^N$ be the vector corresponding to $u$. Then by definition of Laplacian matrix, we have

$$\tilde{u}^\top \boldsymbol{L} \tilde{u} = \|\tilde{u}\|_2^2 - \tilde{u}\boldsymbol{D}^{-1/2}\boldsymbol{A}\boldsymbol{D}^{-1/2}\tilde{u}$$

$$= \sum_{x \in \mathcal{X}} w_x/\tau_x^2 f(x)^2 - \sum_{x,x' \in \mathcal{X}} w_{x,x'}/\tau_{x,x'}^2 f(x)f(x')$$

$$= \frac{1}{2} \sum_{x,x' \in \mathcal{X}} w_{x,x'}/\tau_{x,x'}^2 [f(x) - f(x')]^2. \tag{51}$$

Therefore we have

$$\tilde{R}(u_i^{\hat{y}}) = \frac{1}{2} \frac{\sum_{x,x' \in \mathcal{X}} w_{x,x'}/\tau_{x,x'}^2 [f(x) - f(x')]^2}{\sum_{x \in \mathcal{X}} w_x f(x)^2}. \tag{52}$$

Setting $f(x) = \mathbf{1}[\hat{y}(x) = i]$ finishes the proof. $\square$

*Proof of Lemma B.3.* Let $F_{\text{sc}} = [v_1, v_2, \ldots, v_k]$ be the matrix that contains the smallest $k$ eigenvectors of $\boldsymbol{L} = \boldsymbol{I} - \bar{\boldsymbol{A}}$ as columns, and $f_{\text{sc}}$ is the corresponding feature extractor. By Lemma B.4, there exists a vector $b_i \in \mathbb{R}^k$ with norm bound $\|b_i\|_2 \leq \|u_i^{\hat{y}}\|_2$ such that

$$\|u_i^{\hat{y}} - F_{\text{sc}}b_i\|_w^2 \leq \frac{\tilde{R}(u_i^{\hat{y}})}{1 - \lambda_{k+1}}\|u_i^{\hat{y}}\|_2^2. \tag{53}$$

Combined with Lemma B.5, we have

$$\|u_i^{\hat{y}} - F_{\text{sc}}b_i\|_w^2 \leq \frac{\tilde{\phi}_i^{\hat{y}}}{2(1 - \lambda_{k+1})} \cdot \sum_{x \in \mathcal{X} \cdot \mathbf{1}[\hat{y}(x) = i]}$$

$$= \frac{1}{2(1 - \lambda_{k+1})} \sum_{x,x' \in \mathcal{X}} w_{x,x'}/\tau_{x,x'}^2 \cdot \mathbf{1}[(\hat{y}(x) = i \wedge \hat{y}(x') \neq i) \, or \, (\hat{y}(x) \neq i \wedge \hat{y}(x') = i)]. \tag{54}$$

Let matrix $U := (u_i^{\hat{y}})_{i=1}^k$, and let $u : \mathcal{X} \to \mathbb{R}^k$ be the corresponding feature extractor. Define matrix $B \in \mathbb{R}^{N \times k}$ such that $B^\top = (b_1, \ldots, b_k)$. Summing equation 54 over all $i \in [k]$ and by definition of $\tilde{\phi}^{\hat{y}}$ we have

$$\|U - F_{\text{sc}}B^\top\|_{wF}^2 \leq \frac{1}{2(1 - \lambda_{k+1})} \sum_{x,x' \in \mathcal{X}} w_{x,x'}/\tau_{x,x'}^2 \cdot \mathbf{1}[\hat{y}(x) \neq \hat{y}(x')] = \frac{\tilde{\phi}^{\hat{y}}}{2(1 - \lambda_{k+1})}. \tag{55}$$

By Theorem 4.4, for a feature extractor $f_{\text{pop}}^*$ that minimizes the temperature scaling loss $\mathcal{L}_{\hat{T}}$, the function $f_{\text{mf}}^*(x) := \sqrt{w_x} \cdot f_{\text{pop}}^*$ is a minimizer of the matrix factorization loss $\mathcal{L}_{\text{mf}-\text{T}}$. Then we have

$$\mathbb{E}_{\bar{x} \sim \mathcal{P}_{\bar{X}}, x \sim \mathcal{A}(\cdot|\bar{x})} \|\vec{y}(x) - B^* f_{\text{pop}}^*(x)\|_2^2 \leq 2\mathbb{E}_{\bar{x} \sim \mathcal{P}_{\bar{X}}, x \sim \mathcal{A}(\cdot|\bar{x})} \|\vec{\hat{y}}(x) - B^* f_{\text{pop}}^*(x)\|_2^2 + 2\mathbb{E}_{\bar{x} \sim \mathcal{P}_{\bar{X}}, x \sim \mathcal{A}(\cdot|\bar{x})} \|\vec{\hat{y}}(x) - \vec{y}(x)\|_2^2$$

$$= 2 \sum_{x \in \mathcal{X}} w_x \cdot \|\vec{\hat{y}}(x) - B^* f_{\text{pop}}^*(x)\|_2^2 + 4\Delta(y, \hat{y})$$

$$= 2\|U - F_{\text{sc}}B^\top\|_{wF}^2 + 4\Delta(y, \hat{y})$$

$$\leq \frac{\tilde{\phi}^{\hat{y}}}{1 - \lambda_{k+1}} + 4\Delta(y, \hat{y}). \tag{56}$$

$\square$

Then we move on to the formal proof of Theorem 4.5.

*Proof of Theorem 4.5.* According to equation 35 the proof of Theorem 4.3, if we let

$$
\tau_{x,x'} = \begin{cases}
\dfrac{(1-\alpha) + n\alpha + nr\beta + n_d r(\gamma - \beta)}{(1-\alpha) + n\alpha + nr\beta}, & \text{for } y(x) = y(x'), x, x' \in \mathbb{D}_d, \\[3mm]
\dfrac{\sqrt{(1-\alpha) + n\alpha + nr\beta + n_d r(\gamma - \beta)}}{\sqrt{(1-\alpha) + n\alpha + nr\beta}}, & \text{for } x \in \mathbb{D}_d \text{ or } x' \in \mathbb{D}_d, \\[3mm]
\dfrac{[(1-\alpha) + n\alpha + nr\beta + n_d r(\gamma - \beta)]\beta}{[(1-\alpha) + n\alpha + nr\beta]\gamma} & \text{for } y(x) \neq y(x'), x, x' \in \mathbb{D}_d, \\[3mm]
1, & \text{otherwise,}
\end{cases}
\tag{57}
$$

then we have

$$
\tau_{x,x'} \cdot \frac{w_{x,x'}}{w_x w_{x'}} = \begin{cases}
\dfrac{1}{(1-\alpha) + n\alpha + nr\beta} & \text{for } x = x', \\[3mm]
\dfrac{\alpha}{(1-\alpha) + n\alpha + nr\beta} & \text{for } x \neq x', y(x) = y(x'), \\[3mm]
\dfrac{\beta}{(1-\alpha) + n\alpha + nr\beta} & \text{otherwise.}
\end{cases}
\tag{58}
$$

In this case, $\boldsymbol{T} \odot \bar{\boldsymbol{A}}$ is equivalent to the normalized similarity matrix of data without difficult-to-learn examples.

By Lemma B.3, we have

$$
\mathcal{E}_{\mathrm{T}} \leq \frac{2\tilde{\phi}^{\hat{y}}}{1 - \lambda_{k+1}} + 8\Delta(y, \hat{y}).
\tag{59}
$$

By Assumption B.1, we have $\Delta(y, \hat{y}) \leq \delta$. Besides, since $\tau_{x,x'} \leq 1$ for $y(x) \neq y(x'), x, x' \in \mathbb{D}_c$, and otherwise $\tau_{x,x'} \geq 1$, we have

$$
\tilde{\phi}^{\hat{y}} = \sum_{x,x' \in \mathcal{X}} w_{x,x'}/\tau_{x,x'}^2 \mathbf{1}[\hat{y}(x) \neq \hat{y}(x')]
$$

$$
\leq \sum_{x,x' \in \mathcal{X} \setminus \{x,x' : x,x' \in \mathbb{D}_c\}} w_{x,x'} \mathbf{1}[\hat{y}(x) \neq \hat{y}(x')] + \sum_{y(x) \neq y(x'), x, x' \in \mathbb{D}_c} (\gamma/\beta)^2 w_{x,x'} \mathbf{1}[\hat{y}(x) \neq \hat{y}(x')]
$$

$$
= \sum_{x,x' \in \mathcal{X} \setminus \{x,x' : x,x' \in \mathbb{D}_c\}} \mathbb{E}_{\bar{x} \sim \mathcal{P}_{\bar{\mathcal{X}}}}[\mathcal{A}(x|\bar{x})\mathcal{A}(x'|\bar{x}) \cdot \mathbf{1}[\hat{y}(x) \neq \hat{y}(x')]]
$$

$$
+ (\gamma/\beta)^2 \sum_{x,x' \in \mathbb{D}_c} \mathbb{E}_{\bar{x} \sim \mathcal{P}_{\bar{\mathcal{X}}}}[\mathcal{A}(x|\bar{x})\mathcal{A}(x'|\bar{x}) \cdot \mathbf{1}[\hat{y}(x) \neq \hat{y}(x')]]
$$

$$
\leq \sum_{x,x' \in \mathcal{X} \setminus \{x,x' : x,x' \in \mathbb{D}_c\}} \mathbb{E}_{\bar{x} \sim \mathcal{P}_{\bar{\mathcal{X}}}}[\mathcal{A}(x|\bar{x})\mathcal{A}(x'|\bar{x}) \cdot (\mathbf{1}[\hat{y}(x) \neq \hat{y}(\bar{x})] + \mathbf{1}[\hat{y}(x') \neq \hat{y}(\bar{x})])]
$$

$$
+ (\gamma/\beta)^2 \sum_{x,x' \in \mathbb{D}_c} \mathbb{E}_{\bar{x} \sim \mathcal{P}_{\bar{\mathcal{X}}}}[\mathcal{A}(x|\bar{x})\mathcal{A}(x'|\bar{x}) \cdot (\mathbf{1}[\hat{y}(x) \neq \hat{y}(\bar{x})] + \mathbf{1}[\hat{y}(x') \neq \hat{y}(\bar{x})])]
$$

$$
= 2[1 - (n_d/n)^2]\mathbb{E}_{\bar{x} \sim \mathcal{P}_{\bar{\mathcal{X}}}}[\mathcal{A}(x|\bar{x}) \cdot \mathbf{1}[\hat{y}(x) \neq \hat{y}(\bar{x})]] + 2(\gamma/\beta)^2 (n_d/n)^2 \mathbb{E}_{\bar{x} \sim \mathcal{P}_{\bar{\mathcal{X}}}}[\mathcal{A}(x|\bar{x}) \cdot \mathbf{1}[\hat{y}(x) \neq \hat{y}(\bar{x})]]
$$

$$
= 2[1 - (n_d/n)^2 + (\gamma/\beta)^2 (n_d/n)^2]\delta.
\tag{60}
$$

Therefore we have

$$
\mathcal{E}_{\mathrm{T}} \leq \frac{2\tilde{\phi}^{\hat{y}}}{1 - \lambda_{k+1}} + 8\Delta(y, \hat{y}) \leq [1 - (n_d/n)^2 + (\gamma/\beta)^2 (n_d/n)^2] \cdot \frac{4\delta}{1 - \frac{1-\alpha}{(1-\alpha) + n\alpha + nr\beta}} + 8\delta.
\tag{61}
$$

$\square$

### B.3 RELAXATION ON THE IDEAL ADJACENCY MATRIX

To enhance the connection of the theoretical modeling of difficult-to-learn examples (Section 3.2) to real-world scenarios, we hereby discuss a possible relaxation on the ideal adjacency matrix of the similarity graph.

The adjacency matrix could be relaxed by adding random terms to the similarity values. Specifically, we replace $A$ with $\tilde{A} = (\tilde{a}_{ij})$, where $\tilde{a}_{ii} = 1$, and $\tilde{a}_{ij} = \tilde{a}_{ij} + \epsilon \cdot \varepsilon_{ij}$ for $i \neq j$, $a_{ij}$ takes values in $\{\alpha, \beta, \gamma\}$, $\varepsilon_{ij} = \varepsilon_{ji}$ are i.i.d. Gaussian random variables with mean 0 and variance 1, $\epsilon > 0$ is a small constant. Then $\tilde{A}$ can be decomposed into

$$\tilde{A} = A + \epsilon \cdot W - \epsilon \cdot \mathrm{diag}(\varepsilon_{ii}), \tag{62}$$

where $W$ turns out to be a real Wigner matrix or more specifically a Gaussian Orthogonal Ensemble (GOE). Note that as $\varepsilon_{ij} \sim \mathcal{N}(0, 1)$, the normalization matrix $\tilde{D} \to \mathbb{E}\tilde{D} = D$, as $n(r+1) \to \infty$, and therefore we have $\tilde{\bar{A}} = \tilde{D}^{-1/2}\tilde{A}\tilde{D}^{-1/2} \approx D^{-1/2}\tilde{A}D^{-1/2}$.

For mathematical convenience, in the following analysis, we instead perform the relaxation on the normalized adjacency matrix $\bar{A}$, and investigate

$$\tilde{\bar{A}} = \bar{A} + \epsilon' \cdot W' - \epsilon' \cdot \mathrm{diag}(\varepsilon_{ii}), \tag{63}$$

where $\epsilon > 0$ and $W$ is a GOE.

By Equation 13 in Fulton (2000), we have the $k+1$-th largest eigenvalue of $\tilde{\bar{A}}$ satisfies

$$\tilde{\lambda}_{k+1} \leq \min_{i+j=k+1} \lambda_i + \epsilon' \cdot \nu_j - \epsilon' \cdot \varepsilon_{ii}, \tag{64}$$

where $\lambda_i$ denotes the $i$-th largest eigenvalue of $A$, and $\nu_j$ denotes the $j$-th largest eigenvalue of $W$. And in expectation we have

$$\mathbb{E}\tilde{\lambda}_{k+1} \leq \min_{i+j=k+1} \lambda_i + \epsilon' \cdot \mathbb{E}\nu_j, \tag{65}$$

where the values of $\mathbb{E}\nu_j$ could be deduced according to Wigner's semicircle law. Specifically, denoting $\nu_1, \ldots, \nu_{n(r+1)}$ as the eigenvalues of $W$, we define the empirical spectral measure as $\nu = \frac{1}{n(r+1)} \sum_{i=1}^{n(r+1)} \delta_{\nu_i}$. Then $\nu$ converges weakly almost surely to the quarter-circle distribution on $[0, 2]$, with density

$$f(\nu) = \frac{1}{2\pi}\sqrt{4 - x^2}\mathbf{1}[|x| \leq 2]. \tag{66}$$

Note that equation 65 indicates that the effect of the randomized similarity $\epsilon \cdot \varepsilon_{ij}$ is to add an additional term to the upper bound of the eigenvalue, and the effect is the same regardless of $A$ (e.g. with and without the existence of difficult-to-learn examples). As the linear probing error bound is determined by $\tilde{\lambda}_{k+1}$ (given the labeling error $\delta$), our theoretical results that difficult-to-learn examples hurt unsupervised contrastive learning ($equation\ 4 \geq equation\ 3$) still hold under this relaxation. Moreover, Theorems 4.3 and 4.5 also hold because the relaxation has the same effect to $\mathcal{E}_{\mathrm{M}}$, $\mathcal{E}_{\mathrm{T}}$, and $\mathcal{E}_{\mathrm{w.o.}}$.

