# OpenReview forum: "A Unified Theoretical Framework for Understanding Difficult-to-learn Examples in Contrastive Learning"
_ICLR.cc/2025/Conference — Submitted to ICLR 2025_

### Official Review · Reviewer_yxey · 2024-10-31

**Soundness:** 3
**Presentation:** 3
**Contribution:** 3
**Rating:** 5
**Confidence:** 4

**Summary:**

This paper explores the impact of removing difficult-to-learn examples on unsupervised contrastive learning, finding that their elimination can boost classification performance. Through a theoretical framework modeling sample similarity, the study reveals that difficult examples hinder generalization in contrastive learning. Techniques such as example removal, margin tuning, and temperature scaling are shown to enhance generalization bounds and improve performance, supported by empirical validation.

**Strengths:**

>A theoretical framework is provided for Difficult-to-learn Examples in Contrastive Learning. Any advancement in theory is of interest to me.

>The empirical results show improvement on classification tasks.

**Weaknesses:**

>The tightness of the derived bounds should be validated through experiments.


>The inferior performance of hard samples seems too natural to me from the similarity graph view, as it organizes objects through their similarity and may have a bad influence naturally due to the misconnection in the neighborhood. This phenomenon is very likely to happen when the class numbers are small or the sampled batch size is large. The experiments in Figure 1 should include datasets lie ImageNet as it has many label classes and also should discuss the impact of batch size (buffer size in MoCo).


>Some experiments on larger datasets like ImageNet are strongly encouraged to better show the effectiveness.

**Questions:**

>Can your analysis be generalized to InfoNCE loss? As paper [1] shows through augmentation graph that InfoNCE is also closely related to the decomposition of similarities graphs.


[1]: Contrastive Learning Is Spectral Clustering On Similarity Graph, ICLR 2024

---

> ### Author Response · Authors · 2024-11-21
>
> We express our sincere gratitude to Reviewer yxey for appreciating our proposed theoretical framework based on difficult-to-learn examples in contrastive learning, as well as for recognizing the positive results of our empirical experiments. We address your concerns below.
>
> ---
>
> **Q1:** The tightness of the derived bounds should be validated through experiments.
>
> **A1:** Unfortunately, according to the definition in HaoChen et al., it is scarcely possible to reveal the ground truth augmentation graph, making it impossible to get the real values of $\alpha$, $\beta$, $\gamma$, and $\delta$ in the error bounds. On the one hand, in the population augmentation graph, the vertex set is all augmentation data across the distribution, which is exponentially large. On the other hand, it is hard to measure the ground truth similarity between these augmentations. Directly measuring the pixel-level similarity does not typically makes sense, and human labeling on such data can be subjective and impractical. (This difficulty has also been discussed in [1].) Nonetheless, here we use the cosine similarity matrix of the learned representations with two augmented views per sample as a proxy of the augmentation graph, and hopefully we could cast a glance at the relative values (but of course not the ground truth values) of $\alpha$, $\beta$, and $\gamma$. Based on this, we validate the trending of the derived bounds (the bound getting worse with difficult-to-learn examples increasing) instead of the specific bound values.
>
> Specifically, we use the mixed CIFAR-10 datasets with mixing ratios varying from 0\% to 30\%, where 0\% represents the standard CIFAR-10 without mixing. The experimental parameter settings can be referenced to Appendix A.3 of the original paper. For each class of samples, we sort them based on the difference between the maximum and second-largest values after applying softmax to the outputs, and select the 8% （the ratio is consistent with what is reported in the paper）smallest differences as the difficult-to-learn examples, as described in the paper. For the calculation of $\alpha$, we take the mean of the similarity between all samples of the same class. For the calculation of  $\beta$, we take the mean of the similarity for the sample pairs from different classes that do not contain the difficult-to-learn examples. For the calculation of $\gamma$, we take the mean of the similarity for the sample pairs from different classes that contain the difficult-to-learn examples.
>
> | Mixing Ratio | 0\%   | 10\% | 20\% | 30\% |
> |-------|-----|-----|-----|-----|
> |   acc (\%)    |   88.3  |   88.0|   87.7|  86.2    |
> |   $\alpha$ (\%)    |   47.2  |  44.0   |   41.2  |  38.7  |
> |   $\beta$  (\%)    |   19.1  |  19.5   |   20.1  |  20.8  |
> |   $\gamma$ (\%)    |   20.9  |  22.1   |   23.1  |  24.1  |
> |$\gamma-\beta$ (\%) |   1.80  |	 2.60	|   3.00  |	 3.30  |
> | Eigenvalue ($\times10^{-5}$) |  2.93  |	3.36  |	 3.58  |  3.72  |
>
> In the above table, we show that as the mixing ratio increases, the linear probing accuracy drops, and the $(K+1)$-th eigenvalue increases. Note that the classification error (left hand side of Eq.(4)) is 1-acc, and the error bound (right hand side of Eq.(4)) increases with the eigenvalue increasing. This result indicates that as the difficult-to-lean examples increases, the classification error and the bound share the same variation trend, thus validating Theorem 3.2 that larger $\gamma-\beta$ results in worse error bound.
>
> ---

---

> > ### Author Response · Authors · 2024-11-21
> >
> > **Q2:** The inferior performance of hard samples seems too natural to me from the similarity graph view, as it organizes objects through their similarity and may have a bad influence naturally due to the misconnection in the neighborhood. This phenomenon is very likely to happen when the class numbers are small or the sampled batch size is large. The experiments in Figure 1 should include datasets lie ImageNet as it has many label classes and also should discuss the impact of batch size (buffer size in MoCo).
> >
> > **A2:** In response to the questions you raised, we conducted two sets of experiments to address your concerns. On one hand, regarding the batch size, we performed experiments with different batch sizes on the CIFAR-100 dataset. The results are shown in the table below. We found that for both batch size 512 and 1024, our method both led to performance improvements in sample removal.
> >
> >
> > | Acc (cifar100 )   | baseline | removal   |
> > |-------|----------|-----|
> > | 1024 |  60.17  | 60.50 |
> > | 512  | 59.95     | 60.31|
> >
> > On the other hand, we conducted experiments on a larger dataset, ImageNet-1k. This dataset contains 1,000 categories and more training samples. For specific experimental settings and results, please refer to our response in **A3**.
> >
> > | Acc  (Imagenet-1k)  | Baseline | Removing   |
> > |-------|----------|-----|
> > | 50 epoches |  28.63	  | 28.91  |
> >
> > These results show that our method consistently improves performance, even on a larger and more complex dataset containing more categories.
> >
> > We believe these experiments strengthen our claim that our removal method can lead to performance gains on datasets with varying complexities and sizes.
> >
> > ---
> >
> > **Q3:** Some experiments on larger datasets like ImageNet are strongly encouraged to better show the effectiveness.
> >
> > **A3:** Following your suggestion, we evaluated our method on the Imagenet-1k dataset, which contains 1,000 categories and 1,281,167 training samples. Due to limited computing resources and time, we used ResNet18 as our backbone, set the batch size to 256, and resized each image to 96x96. We set the learning rate to 0.5 and the base temperature to 0.1. We chose 0.01 as the posHigh and 0.5 as the posLow. We set $\sigma$ to 0.1 and $\rho$ to 0.7. We also evaluated the models using linear probing. When evaluating, we set the batch size to 256 and the learning rate to 1. The specific results are shown in the table below.
> >
> > | Imagenet-1k | Baseline |Removing   | Temperature Scaling   | Margin Tuning   | Combined Method   |
> > |-------------|----------|-----|-----|-----|-----|
> > | 50 Epoch Accuracy  |  28.63 | 28.91| 29.38| 29.67| 30.06|
> >
> > From the results on the real-world dataset, Imagenet-1k, which contains more categories, We can see that even after running for 50 epochs, our method achieves a performance improvement trend consistent with the results mentioned in the paper, compared to the baseline method. We will update as soon as we have new results and we hope that this set of experiments will strengthen the findings and demonstrate broader applicability of this paper.
> >
> > ---

---

> ### Author Response · Authors · 2024-11-21
>
> **Q4:** Can your analysis be generalized to InfoNCE loss? As paper [1] shows through augmentation graph that InfoNCE is also closely related to the decomposition of similarities graphs. [1]: Contrastive Learning Is Spectral Clustering On Similarity Graph, ICLR 2024
>
> **A4:** Yes, the analysis of difficult-to-learn examples could potentially be generalized to the similarity graph defined in [1] through the stochastic block model (Holland et al., 1983), which is a widely used random graph model $G=(\mathcal{X},E)$. The graph node is the same as that of augmentation graph, and the edge (undirected and unweighted) is in the edge set with probability $w\_{xx'}$. The similarity graph defined in [1] has edge weight $\pi\_{ij}=1/(\\#\mathrm{aug})$,where $\\#\mathrm{aug}$ denotes the number of augmentations per sample. Therefore, its similarity matrix can be viewed as the normalized ajacency matrix of the random graph $G$, i.e., denoting $\boldsymbol{A}$ as the adjacency matrix of augmentation graph defined in our paper, and $\boldsymbol{\pi}$ as the similarity matrix defined in [1], then we have $\boldsymbol{\pi} = \bar{\boldsymbol{A}'}$, where $\mathbb{E}\boldsymbol{A}'=\boldsymbol{A}$. Note that according to [1], on each of the randomly sampled $\boldsymbol{\pi}$, InfoNCE is equivalent to running spectral clustering on $\boldsymbol{\pi}$, i.e. its error bound is related to the top eigenvalues of $\boldsymbol{\pi}$. Then by Lemmas 3 and 4 in Shen et al.(2022), we could use matrix concentration bounds to concentrate the top eigenvectors of $\boldsymbol{A}$ to those of $\boldsymbol{A}=\mathbb{E}\boldsymbol{A}'$ and use matrix perturbation analysis to show that the predictor learned using $\boldsymbol{A}'$ is close to the “ideal” one learned using $\boldsymbol{A}$. By this, we can show that the error bound of InfoNCE is in expectation related to the top eigenvalues of $\boldsymbol{A}$. Finally, according to the analysis in our paper on how difficult-to-learn samples affect the eigenvalues of $\boldsymbol{A}$, we can get the influence of difficult-to-learn samples on the error bound of InfoNCE.
>
> Please also note that "as proved in Johnson et al. (2022), the spectral contrastive loss and the InfoNCE loss share the same population minimum with variant kernel derivations" (line 157). This result could be a simpler way of showing that our theoretical results on the negative impact of difficult-to-learn examples also applies to InfoNCE.
>
> - Holland et al. Stochastic blockmodels: First steps. Social Networks, 1983.
> - Shen et al. Connect, not collapse: Explaining contrastive learning for unsupervised domain adaptation. in ICML, 2022.
> - Johnson et al. Contrastive learning can find an optimal basis for approximately view-invariant functions. In ICLR, 2022.
>
> ---
>
> Thanks for your insightful and constructive comments. Hope our explanations, analysis, and additional experiments can address your concerns.

---

> ### Author Response · Authors · 2024-11-25
>
> Dear Reviewer yxey,
>
> We would like to express our sincere gratitude for the time and effort you have dedicated to reviewing our manuscript and providing thoughtful feedback. We understand that this is a particularly busy period, and we greatly appreciate the demands on your time.
>
> We have made every effort to carefully address your comments and to conduct additional experiments as per your suggestions. As the Author-Reviewer discussion period is drawing to a close, we would be truly grateful if you could kindly let us know whether our responses have adequately addressed your concerns.
>
> Once again, thank you for your invaluable contributions to our work.
>
> Best regards,
> Authors

---

> > ### Author Response · Authors · 2024-11-26
> >
> > Dear Reviewer yxey,
> >
> > We really appreciate your efforts to help improve this paper. We have carefully addressed all the mentioned concerns. We understand that you might be busy at this time, and we would deeply appreciate it if you could spare a little time to take a look at our rebuttal.
> >
> > Having further discussions really helps to achieve consensus and clarify misunderstandings, so we are eager to know if are there any remaining questions. We are more than happy to try our best to address them. Meanwhile, your further opinions are very important for evaluating our revised paper and we are hoping to hear from you soon. Thank you again for your effort and constructive opinions.
> >
> > Best regards,
> >
> > The Authors.

---

> ### Author Response · Authors · 2024-11-27
> **New updates of the ImageNet-1k experiment**
>
> Dear Reviewer yxey,
>
> We have made new progress regarding the ImageNet-1k experiment that we responded to a week ago. We have now obtained the results after running for 100 epochs, and the experimental results are as follows:
>
> | Imagenet-1k | Baseline |Removing   | Temperature Scaling   | Margin Tuning   | Combined Method   |
> |-------------|----------|-----|-----|-----|-----|
> |100 Epoch Accuracy  |  34.21 | 34.33| 35.02| 35.17| 35.68|
>
> From the experimental results, it is clear that our proposed method can achieve performance improvements on the real-world dataset, ImageNet-1k, similar to those observed on other datasets. Our experiments are still ongoing, and we will provide timely updates with new results as they become available.
>
> In addition, we have addressed each of your other concerns one by one. We eagerly look forward to your feedback on the rebuttal and further discussion, as we believe this is crucial for enhancing the quality of this paper.

---

> ### Author Response · Authors · 2024-11-30
> **Further discussion**
>
> Dear Reviewer yxey,
>
> We have carefully addressed your concerns and we are still looking forward to your reply.
>
> Best regards,
>
> The Authors

---

> > ### Author Response · Authors · 2024-12-02
> >
> > Dear Reviewer yxey,
> >
> > We sincerely thank you for the time and effort you have put into reviewing our manuscript. We understand that this is a busy time, and we deeply appreciate the effort you have made in reviewing our work.
> >
> > We have carefully considered your comments and have undertaken additional experiments as per your suggestions. As the Author-Reviewer discussion period is nearing its end, we would be very grateful if you could kindly let us know if our responses have effectively addressed your concerns.
> >
> > Best regards,
> > Authors

---

> > > ### Author Response · Authors · 2024-12-03
> > >
> > > Dear Reviewer yxey,
> > >
> > > We have addressed your comments and conducted additional experiments. As the Author-Reviewer discussion period is within **the final 12 hours**, we would appreciate it if you could confirm whether our responses have resolved your concerns.
> > >
> > > Best regards,
> > >
> > > Authors

---

### Official Review · Reviewer_KG9m · 2024-11-03

**Soundness:** 2
**Presentation:** 3
**Contribution:** 3
**Rating:** 6
**Confidence:** 4

**Summary:**

The manuscript identifies a universal empirical phenomenon: removing certain training examples improves the performance of unsupervised contrastive learning. The theoretical results further show that difficult-to-learn samples can negatively impact contrastive learning performance and propose several potential solutions to address this by enhancing generalization bounds in spectral contrastive learning. Experimental results support these findings, providing empirical validation for the proposed approach.

**Strengths:**

- The paper offers novel insights into the impact of difficult-to-learn examples in contrastive learning.
- The theoretical analysis is robust, and the proposed solutions are straightforward to implement.

**Weaknesses:**

- Although the authors find that removing certain training examples consistently improves unsupervised contrastive learning performance, the theoretical results are limited to spectral contrastive learning. It may be more accurate to reflect this focus in the title.
- The experiments are relatively small in scale, with the largest dataset being Tiny ImageNet. Evaluating the approach on larger datasets could strengthen the findings and demonstrate broader applicability.

**Questions:**

Please see the comment.

---

> ### Author Response · Authors · 2024-11-21
>
> We express our sincere gratitude to Reviewer KG9m for appreciating both the novel insights our paper offers into the impact of difficult-to-learn examples in contrastive learning and the robustness of the theoretical analysis, as well as the straightforward implementation of the proposed solutions. We address your concerns below.
>
> ---
>
> **Q1:** Although the authors find that removing certain training examples consistently improves unsupervised contrastive learning performance, the theoretical results are limited to spectral contrastive learning. It may be more accurate to reflect this focus in the title.
>
> **A1.** Thank you for the advice. We replace the title with "Understanding Difficult-to-learn Examples in Contrastive Learning: A Theoretical Framework for Spectral Contrastive Learning" in our revised manuscript. It is also worthwhile mentioning that "as proved in Johnson et al. (2022), the spectral contrastive loss and the InfoNCE loss share the same population minimum with variant kernel derivations" (line 157). This indicates that our theoretical results could potentially be extended to other contrastive learning paradigms, e.g. methods using the InfoNCE loss.
>
> ---
>
> **Q2:** The experiments are relatively small in scale, with the largest dataset being Tiny ImageNet. Evaluating the approach on larger datasets could strengthen the findings and demonstrate broader applicability.
>
> **A2:** Following your suggestion, we evaluated our method on the Imagenet-1k dataset, which contains 1,000 categories and 1,281,167 training samples. Due to limited computing resources and time, we used ResNet18 as our backbone, set the batch size to 256, and resized each image to 96x96. We set the learning rate to 0.5 and the base temperature to 0.1. We chose 0.01 as the posHigh and 0.5 as the posLow. We set $\sigma$ to 0.1 and $\rho$ to 0.7. We also evaluated the models using linear probing. When evaluating, we set the batch size to 256 and the learning rate to 1. The specific results are shown in the table below.
>
> | Imagenet-1k | Baseline |Removing   | Temperature Scaling   | Margin Tuning   | Combined Method   |
> |-------------|----------|-----|-----|-----|-----|
> | 50 Epoch Accuracy  |  28.63 | 28.91| 29.38| 29.67| 30.06|
>
> From the results on the real-world dataset, Imagenet-1k, which contains more categories, we can see that even after running for only 50 epochs, our method achieves a performance improvement trend consistent with the results mentioned in the paper, compared to the baseline method. We will update as soon as we have new results and we hope that this set of experiments will strengthen the findings and demonstrate broader applicability of this paper.
>
> ---
>
> Thanks for your insightful and constructive comments. Hope our explanations and additional experiments can address your concerns.

---

> ### Author Response · Authors · 2024-11-25
>
> Dear Reviewer KG9m,
>
> We would like to express our sincere gratitude for the time and effort you have dedicated to reviewing our manuscript and providing thoughtful feedback. We understand that this is a particularly busy period, and we greatly appreciate the demands on your time.
>
> We have made every effort to carefully address your comments and to conduct additional experiments as per your suggestions. As the Author-Reviewer discussion period is drawing to a close, we would be truly grateful if you could kindly let us know whether our responses have adequately addressed your concerns.
>
> Once again, thank you for your invaluable contributions to our work.
>
> Best regards,
> Authors

---

> > ### Author Response · Authors · 2024-11-26
> >
> > Dear Reviewer KG9m,
> >
> > We want to express our appreciation for your valuable suggestions, which greatly helped us improve the quality of this paper. We are also glad that you think our work is well-written and insightful. We have made our effort to address all your concerns.
> >
> > Your further opinions are very important for evaluating our revised paper and we are hoping to hear from you soon. Thank you again for your effort and constructive opinions.
> >
> > Best regards,
> >
> > The Authors.

---

> ### Author Response · Authors · 2024-11-27
> **New updates of the ImageNet-1k experiment**
>
> Dear Reviewer KG9m,
>
> We have made new progress regarding the ImageNet-1k experiment that we responded to a week ago. We have now obtained the results after running for 100 epochs, and the experimental results are as follows:
>
> | Imagenet-1k | Baseline |Removing   | Temperature Scaling   | Margin Tuning   | Combined Method   |
> |-------------|----------|-----|-----|-----|-----|
> |100 Epoch Accuracy  |  34.21 | 34.33| 35.02| 35.17| 35.68|
>
> From the experimental results, it is clear that our proposed method can achieve performance improvements on the real-world dataset, ImageNet-1k, similar to those observed on other datasets. Our experiments are still ongoing, and we will provide timely updates with new results as they become available.
>
> In addition, we have addressed each of your other concerns one by one. We eagerly look forward to your feedback on the rebuttal and further discussion, as we believe this is crucial for enhancing the quality of this paper.

---

> ### Author Response · Authors · 2024-11-30
> **Further discussion**
>
> Dear Reviewer KG9m,
>
> We have carefully addressed your concerns and we are still looking forward to your reply.
>
> Best regards,
>
> The Authors

---

> > ### Author Response · Authors · 2024-12-02
> >
> > Dear Reviewer KG9m,
> >
> > We sincerely thank you for the time and effort you have put into reviewing our manuscript. We understand that this is a busy time, and we deeply appreciate the effort you have made in reviewing our work.
> >
> > We have carefully considered your comments and have undertaken additional experiments as per your suggestions. As the Author-Reviewer discussion period is nearing its end, we would be very grateful if you could kindly let us know if our responses have effectively addressed your concerns.
> >
> > Best regards,
> > Authors

---

> > > ### Author Response · Authors · 2024-12-03
> > >
> > > Dear Reviewer KG9m,
> > >
> > > We have addressed your comments and conducted additional experiments. As the Author-Reviewer discussion period is within **the final 12 hours**, we would appreciate it if you could confirm whether our responses have resolved your concerns.
> > >
> > > Best regards,
> > >
> > > Authors

---

> > > > ### Comment · Reviewer_KG9m · 2024-12-03
> > > >
> > > > Thank you very much for the response. I have no further questions and will maintain my rating

---

### Official Review · Reviewer_aLaR · 2024-11-03

**Soundness:** 2
**Presentation:** 3
**Contribution:** 3
**Rating:** 6
**Confidence:** 3

**Summary:**

In this paper, the authors carefully investigated the effect of difficult-to-learn examples in unsupervised contrastive learning.
A preliminary experiment is conducted to demonstrate the negative effect of such examples, while in supervised learning it is well known that such difficult-to-learn example has positive effects.

Inspired by this preliminary experiment, the authors established a theoretical result, characterizing the effects of difficult-to-learn examples using linear probing error bounds. Their result clearly illustrated the effect of the persistence of such examples in the training set.

Further, the authors analyzed several methods for removing/neglecting the effect of difficult-to-learn examples within the established theoretical frameworks. Finally, several experiments are conducted to validate the theories, while a combined method utilizing the aforementioned methods is proposed to better handle the effect of difficult-to-learn examples.

**Strengths:**

- This work provided a simplified theoretical framework (i.e., to my understanding, the simplified augmentation similarity graph as illustrated in Figure 3: a graph with all entries in its adjacency matrix being one of $1$, $\alpha$, $\beta$ or $\gamma$), which allows easier (tractable) analysis with the spectral contrastive loss.
- Based on the established framework, this work further provides valuable analysis for Margin Tuning and Temperature Scaling. It provides a clear explanation of why and how those methods work for unsupervised contrastive learning.

**Weaknesses:**

## Theories
Despite the clarity of the established framework, its connection to real-world scenarios is weak. In the paper, the idealized adjacency matrix $A$ only has 4 possible entries: $1$, $\alpha$, $\beta$, or $\gamma$, used through out the entire paper (esp. Section 3 & 4). Where in real-world scenarios, the adjacency entries can be quite arbitrary and continuous.

The simplification is made such that the direct computation of eigenvalues of $A$ is possible, yet I feel that there lacks an intuitive, or preferably, empirical justification for the connection between the idealized theoretical framework and the practical case. The only direct explanation I could find is around Sec 3.2, "As difficult-to-learn examples lie around the decision boundary, they should have higher augmentation similarity to examples from different classes." (lines 164-166), which is a rather ambiguous intuition or belief. I'd be glad if the authors could emphasize corresponding parts that I might have overlooked.

For example, this work may potentially be improved via:
- A relaxation on the ideal adjacency matrix, e.g., allowing entries to take values such as $(\alpha - \epsilon, \alpha + \epsilon) \dots$ ; or being stochastic with some distribution.
- A visualization and/or quantitative assessment of the argumentation similarity graph in small-scale datasets (perhaps not feasible, though)

After all, I think it is important to answer questions like 1) To what extent do $\beta$ and $\gamma$ differ in real-world datasets (under this work's intuitive assumption)? 2) In what kind of dataset $\beta$ and $\gamma$ are further separated, compared to other datasets?, etc.

Nevertheless, this simplified viewpoint presented in this work still provides clear and intuitive results that are valuable for future research (besides the questions I listed below), and generalization or empirical validation might be difficult. Therefore, the points above are more suggestions rather than must-do's.

## Experiments
The experimental descriptions are a bit blurry in this work, and I think it should be written clearly. For example, the generating process of $\gamma$-Mixed CIFAR-10 is missing, where the authors only claim that it is "mixing at pixel level" (lines 106-107). Please refer to the Questions section for more details.

Furthermore, in results like Table 1, it is perhaps beneficial to show more statistics beyond classification accuracy, e.g., the number of removed datapoints (overall and in each class), etc.

**Questions:**

## Theories

- The inequality around line 212, $\frac{(1-\alpha)+r(\gamma-\beta)}{(1-\alpha)+n \alpha+n r \beta+n_d r(\gamma-\beta)} > \frac{1-\alpha}{(1-\alpha)+n \alpha+n r \beta}$: When does this hold? Is it practical for it to hold in most datasets?
- In line 1019, "when $n_d < k \leq n_d + r + 1$" - This seems rather a strict condition, are there further explanations on this? (or perhaps it is explained but I have overlooked?).

## Experiments

- What is the "pixel-mixing" method in Section 2 line 107?
- In line 344-345, what features are being used? Is the network being pre-trained on some dataset before the unsupervised contrastive learning process?
- In Figure4 & 5, what are the other hyper-parameters when tuning the plotted parameter?
- Is Section A.4 talking about the combined method?

---

> ### Author Response · Authors · 2024-11-21
>
> We express our sincere gratitude to Reviewer aLaR for appreciating the theoretical framework we propose and the theoretical analysis we made on Margin Tuning and Temperature Scaling. We address your concerns below.
>
> ---
>
> **Q1:** The simplification is made such that the direct computation of eigenvalues of $A$ is possible, yet I feel that there lacks an intuitive, or preferably, empirical justification for the connection between the idealized theoretical framework and the practical case. The only direct explanation I could find is around Sec 3.2, "As difficult-to-learn examples lie around the decision boundary, they should have higher augmentation similarity to examples from different classes." (lines 164-166), which is a rather ambiguous intuition or belief.
>
> **A1:** Firstly, through a simple example of binary classification, we intuitively show that sample pairs with higher augmentation similarity are guaranteed to have at least one sample located near the boundary. Let $x_1$ and $x_2$ be two examples belonging to Class 0 and Class 1 with confidence $p$ and $q$, respectively, i.e., $\mathrm{P}(y=0|x_1)=p$ and $\mathrm{P}(y=1|x_2)=q$, where $p, q \in (1/2,1]$. Then the similarity originated from label posterior probability (concept referring to https://arxiv.org/abs/2306.04160) is $s=(p, 1-p)\cdot(q,1-q)^\top=p+q-2pq$. If $x_1$ lies around the boundary, i.e. $p=1/2+\varepsilon$, $\varepsilon\to 0$, then $s=1/2+(1-2q)\varepsilon$, which is greater than the average different-class similarity 3/8 (Assuming $p,q \sim \mathrm{Unif}(1/2,1]$, then $\mathbb{E}_{p,q}s=3/8$). Therefore, if we select different class pairs with similarity higher than 1/2, the selected difficult-to-learn pairs are guaranteed to have at least one difficult-to-learn sample.
>
> Secondy, we discuss that  a relaxation on the ideal adjacency matrix (randomizing the similarity values) does not affect our main theoretical results (refer to **A2**). Thirdly, using a proxy of augmentation graph, we empirically assess the relative values of $\alpha$, $\beta$, and $\gamma$.
>
> ---
>
> **Q2:** A relaxation on the ideal adjacency matrix, e.g., allowing entries to take values such as $(\alpha - \epsilon, \alpha + \epsilon) \dots$; or being stochastic with some distribution.
>
> **A2:** Thank you for the valuable advice. The adjacency matrix could be relaxed by adding random terms to the similarity values. Specifically, we replace $\boldsymbol{A}$ with $\boldsymbol{A}'=(a'\_{ij})$, where $a'\_{ii}=1$, and $a'\_{ij}=a_{ij}+\epsilon \cdot \varepsilon\_{ij}$ for $i\neq j$, $a\_{ij}$ takes values in $\{\alpha, \beta, \gamma\}$, $\varepsilon\_{ij}=\varepsilon\_{ji}$ are i.i.d. Gaussian random variables with mean 0 and variance 1, $\epsilon>0$ is a small constant. Then $\boldsymbol{A}'$ can be decomposed into $\boldsymbol{A}'=\boldsymbol{A} + \epsilon\cdot\boldsymbol{W} - \epsilon \cdot \mathrm{diag}(\varepsilon_{ii})$, where $\boldsymbol{W}$ turn out to be a real Wigner matrix. According to Wigner’s semicircle law, we have the eigenvalues of $\boldsymbol{W}$ with the probability density function $f(\nu)=\frac{1}{2\pi}\sqrt{4-x^2}\boldsymbol{1}[|x|\leq 2]$. Them following the similar approach in the proof Theorem 3.2, we are able to calculate the upper bound of (the expected value of) the $K+1$ largest eigenvalue of $\boldsymbol{A}'$, and accordingly reach a downstream error bound. Note that the effect of this random similarity $\epsilon \cdot \varepsilon_{ij}$ is to add an additional term to the upper bound of the eigenvalue, and the effect is the same regardless of $\boldsymbol{A}$ (e.g. with and without the existence of difficult-to-learn examples). Therefore, our theoretical results still hold under this relaxation.

---

> ### Author Response · Authors · 2024-11-21
>
> **Q3:** A visualization and/or quantitative assessment of the augmentation similarity graph in small-scale datasets (perhaps not feasible, though). 1) To what extent do $\beta$ and $\gamma$ differ in real-world datasets (under this work's intuitive assumption)? 2) In what kind of dataset $\beta$ and $\gamma$ are further separated, compared to other datasets?, etc.
>
> **A3:** Unfortunately, according to the definition in HaoChen et al., it is scarcely possible to reveal the ground truth augmentation graph. Firstly, in the population augmentation graph, the vertex set is all augmentation data across the distribution, which is exponentially large. Secondly, it is hard to measure the ground truth similarity between these augmentations. (Directly measuring the pixel-level similarity does not typically makes sense, and human labeling on such data can be subjective and impractical.) Nonetheless, we use the cosine similarity matrix of the learned representations with two augmented views per sample as a proxy of the augmentation graph, and hopefully we could cast a glance at the relative values (but of course not the ground truth values) of $\alpha$, $\beta$, and $\gamma$.
>
> Specifically, we use the mixed CIFAR-10 datasets with mixing ratios varying from 0\% to 30\%, where 0\% represents the standard CIFAR-10 without mixing. The experimental parameter settings can be referenced to Appendix A.3 of the original paper. For each class of samples, we sort them based on the difference between the maximum and second-largest values after applying softmax to the outputs, and select the 8% (the ratio is consistent with what is reported in the paper) smallest differences as the difficult-to-learn examples, as described in the paper. For the calculation of $\alpha$, we take the mean of the similarity between all samples of the same class. For the calculation of $\beta$, we take the mean of the similarity for the sample pairs from different classes that do not contain the difficult-to-learn examples. For the calculation of $\gamma$, we take the mean of the similarity for the sample pairs from different classes that contain the difficult-to-learn examples.
>
> | ratio | 0\%   | 10\% | 20\% | 30\% |
> |-------|-----|-----|-----|-----|
> |   $\alpha$ (\%)    |   47.2  |  44.0   |   41.2  |  38.7  |
> |   $\beta$ (\%)     |   19.1  |  19.5   |   20.1  |  20.8  |
> |   $\gamma$  (\%)   |   20.9  |  22.1   |   23.1  |  24.1  |
> |$\gamma-\beta$ (\%) |   1.80  |	 2.60	|   3.00  |	 3.30  |
>
> We show that in this case, 1) the differences between $\beta$ and $\gamma$ lie in the range $[1.8\%,3.3\%]$, and 2) datasets with less separable inter-class samples (the mixed-CIFAR datasets with higher mixing ratio), have higher $\gamma-\beta$ values.
>
> ---
> **Q4:** In results like Table 1, it is perhaps beneficial to show more statistics beyond classification accuracy, e.g., the number of removed datapoints (overall and in each class), etc.
>
> **A4:** To better understand the results in Table 1, we present the distribution of the removed sample pairs from different classes in the CIFAR-10 dataset using our method. The results are shown in the table below.
>
>
> | removing pairs | 0     | 1     | 2     | 3     | 4     | 5     | 6     | 7     | 8     | 9     |
> |----------------|-------|-------|-------|-------|-------|-------|-------|-------|-------|-------|
> | 0              | /| 119196| 118195| 105113| 122238| 107847| 118977| 123684| 188477| 256489|
> | 1              | 119196| /| 122935| 101823| 101801| 104581| 117245| 134237| 182918| 273113|
> | 2              | 118195| 122935|/ | 153348| 200188| 173305| 205533| 149932| 67464 | 48044 |
> | 3              | 105113| 101823| 153348| /| 201703| 241832| 230805| 128494| 52336 | 42202 |
> | 4              | 122238| 101801| 200188| 201703| /| 200898| 198208| 266404| 44424 | 28024 |
> | 5              | 107847| 104581| 173305| 241832| 200898|/ | 196646| 173656| 54626 | 40764 |
> | 6              | 118977| 117245| 205533| 230805| 198208| 196646| /| 127022| 86742 | 42924 |
> | 7              | 123684| 134237| 149932| 128494| 266404| 173656| 127022|/ | 60817 | 81156 |
> | 8              | 188477| 182918| 67464 | 52336 | 44424 | 54626 | 86742 | 60817 |  /| 197129|
> | 9              | 256489| 273113| 48044 | 42202 | 28024 | 40764 | 42924 | 81156 | 197129| /|
> | **Column Sum** | 1194966| 1189388| 1045880| 1054853| 1393933| 1428681| 1369300| 1224981| 990492| 1538524| 13795629 |
>
> We can observe that there are more difficult-to-learn example pairs removed between categories that are easily confused with each other (for example, the 3-5 sample pair represents two similar and easily confused animals, cats and dogs, with a count of 241,832, while the 0-9 pair, representing two types of vehicles—car and truck—has a count of 256,489). This observation aligns with our intuition. We hope that this data presentation helps you better understand Table 1.

---

> ### Author Response · Authors · 2024-11-21
>
> **Q5:** The inequality around line 212, $\frac{(1-\alpha) + r(\gamma - \beta )}{(1-\alpha) + n\alpha + nr\beta + n_dr (\gamma - \beta )} > \frac{1-\alpha}{(1-\alpha) + n\alpha + nr\beta}$
>   When does this hold? Is it practical for it to hold in most datasets?
>
> **A5:** This inequality holds as long as $\frac{1}{n_d} > \frac{1-\alpha}{(1-\alpha) + n\alpha + nr\beta}$ or equivalently $\frac{n_d-1}{n} < \frac{\alpha+r\beta}{1-\alpha}$. That is, given $\alpha$, $\beta$, and $r$, as long as $n_d/n$ is small, this inequality always holds. For CIFAR-10, $n_d/n \approx 0.08$ in our experiments, whereas according to **A3**, $\frac{1-\alpha}{(1-\alpha) + n\alpha + nr\beta} = 4.51$, significantly greater than $\frac{n_d-1}{n}$.
>
> ---
>
> **Q6:** In line 1019, "when $n_d < k \leq n_d + r + 1$" - This seems rather a strict condition, are there further explanations on this? (or perhaps it is explained but I have overlooked?)
>
> **A6:** We apologize for the typo here. It should be $r+1\leq k<n_d+r+1$. On the one side, it is natual to assume that in most datasets, the dimension $k$ is larger than the number of classes $r+1$, since the number of classes typically cannot be too large. On the other side, the number of difficult-to-learn samples $n_d$ is usually larger, as the overall augmented sample size is usually large. Specifically, we have $n_d = n_{aug} \cdot \rho_d \cdot n$, where we denote $n_{aug}$ as the number of augmented views of each sample and $\rho_d$ as the fraction of difficult-to-learn samples in the dataset. Theoretically, $n_{aug}$ should be sufficiently large since the augmentation graph describes the similarity between all kinds of augmentations (despite that it is usually chosen to be 2 in empirical settings). According to our experiments, $\rho_d$ is around $8\\%$. Therefore, it is easy for $n_d + r + 1$ to exceed $k$.
>
> ---
>
> **Q7:** What is the "pixel-mixing" method in Section 2 line 107?
>
> **A7:** The "pixel-mixing" method involves selecting a random image from a different class and then stitching the two images together by splitting each image into halves. Specifically, we take the first half of the current image and the second half of a randomly selected image from another class and combine them to form a new mixed image. This process is used to augment the data, creating a dataset with more difficlut-to-learn examples.
>
> ---
>
> **Q8:** In line 344-345, what features are being used? Is the network being pre-trained on some dataset before the unsupervised contrastive learning process?
>
> **A8:** We do not use any other pre-trained models, but instead, we train the model from scratch based on the optimization objective detailedly described in Appendix A.2, specifically addressing the presence of difficult-to-learn examples.
>
> ---
> **Q9:** In Figure 4 & 5, what are the other hyper-parameters when using the tuned parameter?
>
> **A9:** Figures 4 & 5 present ablation studies conducted on the CIFAR-100 dataset. As described in Section A.3, we set the batch size to 512, the learning rate to 0.5, and the base temperature to 0.1. We chose 0.013 as the posHigh and 0.5 as the posLow. We set $\sigma$ to 0.1 and $\rho$ to 0.7. We also evaluated the models using linear probing. During evaluation, we set the batch size to 256 and the learning rate to 1. Specifically, in Fig 4(a), we only changed posHigh, in Fig 4(b), we only changed posLow, in Fig 5(a), we only changed $\sigma$, and in Fig 5(b), we only changed $\rho$, with all other parameters remaining consistent with the above settings.
>
> ---
>
> **Q10:** Is Section A.4 talking about the combined method?
>
> **A10:** Yes, Section A.4 is talking about the combined method.
>
> ---
>
> Thanks for your insightful and constructive comments. Hope our explanations, analysis, and additional experiments can address your concerns.

---

> ### Author Response · Authors · 2024-11-25
>
> Dear Reviewer aLaR,
>
> We would like to express our sincere gratitude for the time and effort you have dedicated to reviewing our manuscript and providing thoughtful feedback. We understand that this is a particularly busy period, and we greatly appreciate the demands on your time.
>
> We have made every effort to carefully address your comments and to conduct additional experiments as per your suggestions. As the Author-Reviewer discussion period is drawing to a close, we would be truly grateful if you could kindly let us know whether our responses have adequately addressed your concerns.
>
> Once again, thank you for your invaluable contributions to our work.
>
> Best regards,
> Authors

---

> > ### Author Response · Authors · 2024-11-26
> >
> > Dear Reviewer aLaR,
> >
> > We want to express our appreciation for your valuable suggestions, which greatly helped us improve the quality of this paper. We are also glad that you think our work is well-written and insightful. We have made our effort to address all your concerns.
> >
> > Your further opinions are very important for evaluating our revised paper and we are hoping to hear from you soon. Thank you again for your effort and constructive opinions.
> >
> > Best regards,
> >
> > The Authors.

---

> > > ### Comment · Reviewer_aLaR · 2024-11-26
> > >
> > > Thank you for addressing my concerns.
> > >
> > > I am happy that the authors provide an explanation on the possible extension of the existing framework, as well as clarifying some parts of the paper, especially for experiment setups and technical details. It might be beneficial to include such remark (i.e. the relaxed graph) in the paper (please correct me if it already included in the revision) as it will be more convincing for the readers.
> > >
> > > That being said, the experiment provided in A3 is still a bit unconvincing for me. The dataset, mixed CIFAR-10, is particularly artificial and perhaps could be distant from real-life datasets. Since you kept a large proportion of the image in either class in the mixed image, the increase in the mean of “$\gamma - \beta$” is not surprising. I doubt if the difference between $\gamma$ and $\beta$ will be significant in real datasets. Nevertheless, the mixed CIFAR dataset is a well-illustrated example that fits as a preliminary experiment, demonstrating the overall idea. Also, as the authors have said, it is nearly impossible to know much about the augmentation graph in general, and perhaps this is the best we can get. A3 for sure uncovered some underlying properties of the augmentation graph.
> > >
> > > Due to above reasons and limitations, I still think the original score is appropriate. I deeply appreciate the detailed response of the authors that has addressed my concerns!

---

> > > > ### Author Response · Authors · 2024-11-26
> > > > **Response to your latest concerns**
> > > >
> > > > We thank you again for your timely replies. We would like to follow up on your comments and further discuss the points you raised.
> > > >
> > > > ---
> > > >
> > > > **Q1:** It might be beneficial to include such remark (i.e. the relaxed graph) in the paper (please correct me if it already included in the revision) as it will be more convincing for the readers.
> > > >
> > > > **A1:** Thank you for your suggestion. We have the related discussions in Section B.3 of the revised manuscript.
> > > >
> > > > ---
> > > >
> > > > **Q2:** I doubt if the difference between $\gamma$ and $\beta$ will be significant in real datasets.
> > > >
> > > > **A2:** To address your concerns, we tested $\gamma$ and $\beta$, as well as $\gamma - \beta$, on more real datasets. We found that on the CIFAR-100 dataset (which has 10 times more classes than CIFAR-10), the difference between $\gamma$ and $\beta$ remained consistent with that on the CIFAR-10 dataset. On the ImageNet-1k dataset (which has 100 times more classes than CIFAR-10，for specific experimental details and results on ImageNet-1k, we present them below and also in Section A.5 of the revised manuscript), the difference between $\gamma$ and $\beta$ was even larger than on CIFAR-10. As a possible intuitive explanation, we conjecture that the higher $\gamma-\beta$ might results from the higher complexity of imagenet images, e.g. different-class images with similar backgrounds can share higher similarity (higher $\gamma$), whereas CIFAR images have relatively simple and consistent backgrounds. These results demonstrate that even on real-world datasets, the difference between $\gamma$ and $\beta$ is significant.
> > > >
> > > > | Datasets | CIFAR-10 (10 classes)  | CIFAR-100 (100 classes) | Imagenet-1k (1,000 classes) |
> > > > |-------|-----|-----|-----|
> > > > |   $\alpha$    |   47.2  |  48.1  |  54.3   |
> > > > |   $\beta$     |   19.1  |   35.6  |  39.8   |
> > > > |   $\gamma$    |   20.9  |   37.4  |   42.9  |
> > > > |$\gamma-\beta$ |   1.8  |	  1.8 |  3.1   |
> > > >
> > > > To better illustrate the significant difference between $\gamma$ and $\beta$, we conducted an independent samples t-test to support our conclusion. Specifically, we first collected all the $\beta$ and $\gamma$ values, and due to the large sample size, we chose to use Welch's t-test, which does not assume equal variances between the two groups and is suitable for cases where the variances may differ. We report the t-statistics and P values in the following table.
> > > >
> > > >
> > > > | Datasets  | CIFAR-10 (10 classes)  | CIFAR-100 (100 classes) | Imagenet-1k (1,000 classes) |
> > > > |-------|-----|-----|-----|
> > > > |   t-statistic   |   -502.63  |  -539.36  |   -3844.21  |
> > > > |   P value    |   0.0  |   0.0  |   0.0  |
> > > > |   statistically significant   |   Yes |   Yes  |  Yes   |
> > > >
> > > >
> > > > Based on the calculated results, on all datasets (CIFAR-10, CIFAR-100, Imagenet-1k), the absolute value of the t-statistic is very large, and the P value is close to zero( if the p-value is less than 0.05, it typically means that the difference between the two groups is statistically significant). This indicates that the mean difference between $\gamma$ and $\beta$ is highly statistically significant.
> > > >
> > > > **We hope that these two sets of experiments will alleviate your concerns regarding whether the difference between $\gamma$ and $\beta$ will be significant in real-world datasets. We have the related discussions in Section A.5 of the revised manuscript.**
> > > >
> > > >
> > > > (Additional experiment on ImageNet-1k.) We evaluated our method on the Imagenet-1k dataset, which contains 1,000 categories and 1,281,167 training samples. Due to limited computing resources and time, we used ResNet18 as our backbone, set the batch size to 256, and resized each image to 96x96. We set the learning rate to 0.5 and the base temperature to 0.1. We chose 0.01 as the posHigh and 0.5 as the posLow. We set $\sigma$ to 0.1 and $\rho$ to 0.7. We also evaluated the models using linear probing. When evaluating, we set the batch size to 256 and the learning rate to 1. The specific results are shown in the table below.
> > > >
> > > > | Imagenet-1k | Baseline |Removing   | Temperature Scaling   | Margin Tuning   | Combined Method   |
> > > > |-------------|----------|-----|-----|-----|-----|
> > > > |100 Epoch Accuracy  |  34.21 | 34.33| 35.02| 35.17| 35.68|
> > > >
> > > > ---
> > > >
> > > > We sincerely appreciate your constructive comments. We hope that our explanations and additional experiments will help address your concerns, and we look forward to your response.

---

> > > > ### Author Response · Authors · 2024-11-28
> > > > **Further discussion**
> > > >
> > > > Dear Reviewer aLaR,
> > > >
> > > > Thank you for your positive feedback on our efforts. Regarding your remaining concern about whether $\gamma$ and $\beta$ will be significant in real datasets, we have conducted additional experiments to address this. We hope that our response has addressed your concerns.
> > > >
> > > > Could you take some time to review our response? We look forward to hearing from you soon.
> > > >
> > > > Thank you again for your effort and constructive feedback.
> > > >
> > > > Best regards,
> > > >
> > > > The Authors

---

> > > > > ### Author Response · Authors · 2024-11-30
> > > > >
> > > > > Dear Reviewer aLaR,
> > > > >
> > > > > We have carefully addressed your concerns and we are still looking forward to your reply.
> > > > >
> > > > > Best regards,
> > > > >
> > > > > The Authors

---

> > > > > > ### Author Response · Authors · 2024-12-02
> > > > > >
> > > > > > Dear Reviewer aLaR,
> > > > > >
> > > > > > We sincerely thank you for the time and effort you have put into reviewing our manuscript. We understand that this is a busy time, and we deeply appreciate the effort you have made in reviewing our work.
> > > > > >
> > > > > > Thank you for your recognition of our previous responses. **Regarding your final concern, whether the difference between $\beta$ and $\gamma$ will be significant in real-world datasets, we have conducted extensive experiments to demonstrate that $\beta$ and $\gamma$ are indeed significant in real-world datasets.**
> > > > > >
> > > > > > As the Author-Reviewer discussion period is nearing its end, we would be very grateful if you could kindly let us know if our responses have effectively addressed your concerns.
> > > > > >
> > > > > > Best regards,
> > > > > > Authors

---

> > > > > > > ### Author Response · Authors · 2024-12-03
> > > > > > >
> > > > > > > Dear Reviewer aLaR,
> > > > > > >
> > > > > > > We have addressed your comments and conducted additional experiments. As the Author-Reviewer discussion period is within **the final 12 hours**, we would appreciate it if you could confirm whether our responses have resolved your concerns.
> > > > > > >
> > > > > > > Best regards,
> > > > > > >
> > > > > > > Authors

---

### Official Review · Reviewer_eRke · 2024-11-06

**Soundness:** 2
**Presentation:** 2
**Contribution:** 2
**Rating:** 3
**Confidence:** 5

**Summary:**

This paper proposes a theoretical framework for understanding difficulty-to-learn examples in contrastive learning. This further suggests a heuristic approach to determine difficulty-to-learn examples, removes them, do margin tuning, and perform temperature scaling to improve performance.

**Strengths:**

- The idea of determining difficulty-to-learn examples is intuitive and interesting.

**Weaknesses:**

- The terminology difficulty-to-learn examples seems to be confusing because indeed, the paper focuses on difficulty-to-learn pairs (different classes and stay closely to the boundary).
- The definitions of difficulty-to-learn or easy-to-learn pairs are not rigorous.
- In the theory development in Theorems 3.1, 3.2, and so on, it is unclear how to train the feature extractor f for obtaining feature vectors. If it is learned using Eq. (1) similar to HaoChen et al., it is unclear how $\alpha, \beta, \gamma$ are relevant.
- The difficult-to-learn example selection is very heuristic and connects losingly to the theory development.
- The paper seems to be a combination of HaoChen et al. and Zhou et al.
- The experiments are humble and the paper does not have strong practical implication.

**Questions:**

Please address my questions in Weaknesses.

---

> ### Author Response · Authors · 2024-11-21
>
> We express our sincere gratitude to Reviewer eRke for appreciating the intuitive and interesting idea of determining difficult-to-learn examples. We address your concerns below.
>
> ---
>
> **Q1:** The terminology difficulty-to-learn examples seems to be confusing because indeed, the paper focuses on difficulty-to-learn pairs (different classes and stay closely to the boundary).
>
> **A1:** The concept difficulty-to-learn examples is important in the motivation of our work, i.e., difficulty-to-learn examples (boundary examples), which are vitally important in supervised learning, can actually hurt the performance of unsupervised contrastive learning. As contrastive learning leverages examples in a pairwise manner, we require the concept of difficult-to-learn sample pairs, i.e., sample pairs that include at least one difficult-to-learn sample. To avoid possible confusion, we clearly emphasize the relationship between "difficult-to-learn samples" and "difficult-to-learn pairs" at the beginning of Section 3.2 in our revised manuscript.
>
> ---
>
> **Q2:** The definitions of difficulty-to-learn or easy-to-learn pairs are not rigorous.
>
> **A2:** In this paper, we define difficult-to-learn sample pairs as sample pairs that include at least one difficult-to-learn sample. "In real image datasets, as difficult-to-learn examples rely on the specific classifier trained in the supervised learning manner, we can not preciously know the ground truth difficult-to-learn examples" (line 102-104). Therefore, we instead define the difficulty-to-learn pairs as different-class sample pairs with higher similarity, because "pairs containing difficult-to-learn samples exhibit higher similarities than other different-class pairs" (line 63). Correspondingly, easy-to-learn pairs are defined as different-class sample pairs containing no difficult-to-learn samples, or different-class sample pairs with lower similarity. We add these definitions at the beginning of Section 3.2 in our revised manuscript.
>
> ---
>
> **Q3:** In the theory development in Theorems 3.1, 3.2, and so on, it is unclear how to train the feature extractor f for obtaining feature vectors. If it is learned using Eq. (1) similar to HaoChen et al., it is unclear how alpha beta gamma are relevant.
>
> **A3:** First, we clarify that $\alpha$, $\beta$, and $\gamma$ are not parts of the algorithm. Instead, they are part of the theoretical modeling, representing the similarities between different kinds of sample pairs, based on which we derive our main theoretical results that difficult-to-learn samples hurt unsupervised contrastive learning, and this negative effect can be fixed through methods such as sample removal, margin tuning, and temperature scaling. In Theorems 3.1 and 3.2, $f$ is trained under standard (spectral) contrastive learning, but trained with different dataset (whether containing difficult-to-learn example or not). In Corollary 4.1, Theorem 4.3, and Theorem 4.5, $f$ is trained under contrastive learning with sample removal, margin tuning, and temperature scaling, repectively. In Theorems 4.3 and 4.5, we point out that the corresponding contrastive losses are Eq.(31) and Eq.(40) (lines 263 and 295). Besides, "the specific loss forms and algorithms can be found in Appendix A.2" (line 343).
>
> ---

---

> > ### Author Response · Authors · 2024-11-21
> >
> > **Q4:** The difficult-to-learn example selection is very heuristic and connects losingly to the theory development.
> >
> > **A4:** Please note that the primary focus of our theoretical work is not on how to identify difficult-to-learn samples but revealing the impact of their presence on the error bound. Nonetheless, we argue that our selection mechanism is highly intuitive and well-aligned with our theoretical modeling. In fact, it guarantees the selected pair to have at least one sample located near the boundary. Taking binary classification as an example. Let $x_1$ and $x_2$ be two examples belonging to Class 0 and Class 1 with confidence $p$ and $q$, respectively, i.e., $\mathrm{P}(y=0|x_1)=p$ and $\mathrm{P}(y=1|x_2)=q$, where $p, q \in (1/2,1]$. Then the similarity originated from label posterior probability (concept referring to https://arxiv.org/abs/2306.04160) is $s=(p, 1-p)\cdot(q,1-q)^\top=p+q-2pq$. If $x_1$ lies around the boundary, i.e. $p=1/2+\varepsilon$, $\varepsilon\to 0$, then $s=1/2+(1-2q)\varepsilon$, which is greater than the average different-class similarity 3/8 (Assuming $p,q \sim \mathrm{Unif}(1/2,1]$, then $\mathbb{E}_{p,q}s=3/8$). Therefore, if we select different class pairs with similarity higher than 1/2, the selected difficult-to-learn pairs are guaranteed to have at least one difficult-to-learn sample. Moreover, Figure 4(C) demonstrates that our proposed selection mechanism is highly accurate in identifying difficult-to-learn samples, collectively highlighting the effectiveness of our approach.
> >
> > ---
> >
> > **Q5:** The paper seems to be a combination of HaoChen et al. and Zhou et al.
> >
> > **A5:** We respectfully disagree. Our work provides unique contributions, particularly in demonstrating how difficult-to-learn samples negatively impact unsupervised contrastive learning performance, and how sample removal, margin tuning, and temperature scaling can improve generalization bounds in different ways. These aspects have not been explored in previous works, neither in HaoChen et al. nor in Zhou et al..
> >
> > While HaoChen et al. first proposed the concepts of augmentation graph and spectral contrastive learning, they did not address the effects of difficult-to-learn examples, which is the central focus of our study.
> >
> > Seemingly similar to our paper, Zhou et al. introduced the margin tuning technique to rectify incorrect connections. However, when taking a closer look, its motivaition, technical details, and explanations are entirely different from ours. Firstly, the incorrect connections in Zhou et al. originate from cross-domain images or label noise, instead of boundary examples. That is, we are still the first to investigate the effect of difficult-to-learn examples to unsupervised contrastive learning. Secondly, the application details of margin tuning and the working mechanisms are different. In Zhou et al., margin tuning is applied to all examples, and it works because subtracting a suitable positive constant and re-normalization lowers the unwanted edge weights. On the contrary, we apply margin tuning only to the selected difficult-to-learn example pairs, and the mechanism is to subtract a normalized margin matrix from the normalized similarity matrix (Theorem 4.2). Thirdly, aside from margin tuning, we also investigate the effects of sample removing and temperature scaling. Lastly, we conjecture that the reviewer relate our paper to Zhou et al. also because these two papers use the same set of notations ($\alpha$, $\beta$, $\gamma$), but it is important to emphasize that these are merely notational similarities, and the meaning of these notations, the related modeling, and analysis are entirely different in our work.
> >
> > ---

---

> ### Author Response · Authors · 2024-11-21
>
> **Q6:** The experiments are humble and the paper does not have strong practical implication.
>
> **A6:** We respectfully acknowledge the reviewer’s concern about the practical implications of our work. However, as we emphasized at the beginning of Section 5, our primary focus is on the theoretical analysis, where we explore how different types of samples in contrastive learning impact generalization. The experiments are designed to validate these theoretical insights and demonstrate that the proposed approaches for improving performance are sound and applicable.
>
> In summary, our experimental results demonstrate that the proposed method consistently achieves robust performance improvements on widely recognized contrastive learning datasets such as CIFAR-10, CIFAR-100, STL-10, and TinyImageNet (refer to Table 4). We also provide ablation studies (refer to Table 4) and hyperparameter analyses (refer to Figure 5 and Table 8).
>
> Furthermore, during this discussion period, we conduct additional experiments validating our theoretical insights.
>
> 1) We conducted experiments on the ImageNet-1K dataset, which contains 1,000 categories and 1,281,167 training images. Due to limited computing resources and time, we used ResNet18 as our backbone, set the batch size to 256, and resized each image to 96x96. We set the learning rate to 0.5 and the base temperature to 0.1. We chose 0.01 as the posHigh and 0.5 as the posLow. We set $\sigma$ to 0.1 and $\rho$ to 0.7. We also evaluated the models using linear probing. When evaluating, we set the batch size to 256 and the learning rate to 1. The experimental results show that our method achieves consistent performance improvements on real-world datasets with a larger number of categories when running 50 epoches, we will update as soon as we have new results.
>
> | Imagenet-1k | Baseline |Removing   | Temperature Scaling   | Margin Tuning   | Combined Method   |
> |-------------|----------|-----|-----|-----|-----|
> | 50 Epoch Accuracy  |  28.63 | 28.91| 29.38| 29.67| 30.06|
>
>
> 2) We use the cosine similarity matrix of the learned representations with two augmented views per sample as a proxy of the augmentation graph, and validate the trending of our derived bounds. Specifically, we use the mixed CIFAR-10 datasets with mixing ratios varying from 0\% to 30\%, where 0\% represents the standard CIFAR-10 without mixing. We show that 1) datasets with less separable inter-class samples (higher mixing ratio), have higher $\gamma-\beta$ values, and 2) the classification error (1-acc, left hand side of Eq.(4)) and the bound (positively correlated with the $K+1$-th largest eigenvalue, right hand side of Eq.(4)) share the same variation trend.
>
> | Mixing Ratio | 0\%   | 10\% | 20\% | 30\% |
> |-------|-----|-----|-----|-----|
> |   acc (\%)         |    88.3  |   88.0|   87.7|  86.2    |
> |   $\alpha$ (\%)    |   47.2  |  44.0   |   41.2  |  38.7  |
> |   $\beta$  (\%)    |   19.1  |  19.5   |   20.1  |  20.8  |
> |   $\gamma$  (\%)   |   20.9  |  22.1   |   23.1  |  24.1  |
> |$\gamma-\beta$ (\%) |   1.80  |	 2.60	|   3.00  |	 3.30  |
> | Eigenvalue ($\times10^{-5}$) |  2.93  |	3.36  |	 3.58  |  3.72  |
>
> These results highlight the practical relevance of our method, as it demonstrates effectiveness across datasets with varying complexity. We believe this set of experiments reinforces our theoretical framework and demonstrates the broader applicability of our work.
>
> ---
>
> Thanks for your comments. Hope our explanations and additional experiments can address your concerns.

---

> > ### Comment · Reviewer_eRke · 2024-12-01
> > **Response to the rebuttal from the authors**
> >
> > Thanks for answering my questions and conducting more experiments. Sorry for quite late response due to my busy week.
> >
> > My biggest concern is the meaning of the adjacency matrix in Figure 3d. Is it the one used to approximate the true adjacency matrix for further developing the theories? If so, the current approximation using the hard-to-learn examples is not appropriate to me
> > -  It does not take into account the class similarities. For example the classes dog and cat are highly similar, hence their similarities are always high no matter of hard or easy-to-learn examples.
> > -  You define hard-to-learn examples as the ones located nearby the decision boundary and say that *As difficult-to-learn examples lie around the decision boundary, they should have higher augmentation similarity to examples from different classes*. I do not think it is an adequate comment because decision boundary should be relevant to two specific classes, for example, a cat example looking similar to dogs, hence lying nearby the decision boundary of cat/dog does not have high similarity to cars.
> > -  Finally, I do not think that discretizing the true adjacency matrices using only three values $\alpha, \beta$, and $\gamma$ provides sufficient precision to the task.
> >
> > In terms of novelty, I still maintain of point of limited novelty based on previous works. Therefore, I decide to maintain my current scores.

---

> > > ### Author Response · Authors · 2024-12-01
> > > **Responses to further concerns of Reviewer eRke (1/2)**
> > >
> > > We appreciate the feedback from reviewer eRke，but we must assert that there are still significant misunderstandings of the contributions of our paper. We understand your busyness, but we strongly suggest the reviewer spend more time on reading our paper.
> > >
> > > ---
> > >
> > > **Q1.** My biggest concern is the meaning of the adjacency matrix in Figure 3d. Is it the one used to approximate the true adjacency matrix for further developing the theories?
> > >
> > > **A1.** Yes, but Figure 3d is just a 4-sample toy example of our theoretical modeling. The specific adjacency matrix is shown in Appendix B (Eq.(18)).
> > >
> > > ---
> > >
> > > **Q2.** It does not take into account the class similarities. For example the classes dog and cat are highly similar, hence their similarities are always high no matter of hard or easy-to-learn examples.
> > >
> > > **A2.** Our modeling does not take into account the class similarities because the unsupervised contrastive learning model is designed to treat all classes equally. **Artificially defining difficult-to-learn examples as a class-wise concept obeys this design and may provide false insights about the effect of difficult-to-learn examples.** The unsupervised model will not treat the examples of dog or cat different because they belong to the so-called hard classes.
> > >
> > > It is true that the representations of the classes dog and cat usually have higher similarity compared with other classes, whereas **it is exactly because these two classes have more difficult-to-learn examples rather than that "their similarities are always high"**. On the contrary, **most of the dog-cat pairs are in fact easy-to-learn**. In the following table, we count the empirically selected difficult-to-learn examples per class to verify this point.
> > >
> > > | selected pairs | 0     | 1     | 2     | 3     | 4     | 5     | 6     | 7     | 8     | 9     |
> > > |----------------|-------|-------|-------|-------|-------|-------|-------|-------|-------|-------|
> > > | 0              | /| 119196| 118195| 105113| 122238| 107847| 118977| 123684| 188477| 256489|
> > > | 1              | 119196| /| 122935| 101823| 101801| 104581| 117245| 134237| 182918| 273113|
> > > | 2              | 118195| 122935|/ | 153348| 200188| 173305| 205533| 149932| 67464 | 48044 |
> > > | 3              | 105113| 101823| 153348| /| 201703| 241832| 230805| 128494| 52336 | 42202 |
> > > | 4              | 122238| 101801| 200188| 201703| /| 200898| 198208| 266404| 44424 | 28024 |
> > > | 5              | 107847| 104581| 173305| 241832| 200898|/ | 196646| 173656| 54626 | 40764 |
> > > | 6              | 118977| 117245| 205533| 230805| 198208| 196646| /| 127022| 86742 | 42924 |
> > > | 7              | 123684| 134237| 149932| 128494| 266404| 173656| 127022|/ | 60817 | 81156 |
> > > | 8              | 188477| 182918| 67464 | 52336 | 44424 | 54626 | 86742 | 60817 |  /| 197129|
> > > | 9              | 256489| 273113| 48044 | 42202 | 28024 | 40764 | 42924 | 81156 | 197129| /|
> > > | **Column Sum** | 1194966| 1189388| 1045880| 1054853| 1393933| 1428681| 1369300| 1224981| 990492| 1538524| 13795629 |
> > >
> > > Class 3 in this table represents the category "cat", and Class 5 represents "dog".
> > >
> > > On one hand, looking at the column for "cat" (Class 3), the total number of sample pairs that contain cat is 1,054,853, and the cat-dog sample pairs account for 241,832 pairs. This suggests that difficult-to-learn example pairs related to cats are not, as you may have feared, exclusively cat-dog pairs. On the other hand, there are a total of 1,000,000 cat-dog sample pairs in the dataset, which means that 76% of these cat-dog pairs are actually easy-to-learn example pairs. Only a small portion of them are defined as difficult-to-learn example pairs.
> > >
> > > ---
> > >
> > > **Q3.** You define hard-to-learn examples as the ones located nearby the decision boundary and say that As difficult-to-learn examples lie around the decision boundary, they should have higher augmentation similarity to examples from different classes. I do not think it is an adequate comment because decision boundary should be relevant to two specific classes, for example, a cat example looking similar to dogs, hence lying nearby the decision boundary of cat/dog does not have high similarity to cars.
> > >
> > > **A3.** In fact, as shown in the detailed theoretical modeling of the adjacency matrix (Eq.(18), Appendix B), we do not assign high similarity to the cat-car pair just because the cat lies near the cat-dog boundary. **The difficult-to-learn sample pairs are defined within each two specific classes (within each $\boldsymbol{A}'_{\mathrm{different-class}}$ submatrix).** For the sake of preciseness, we will replace the expression with "As difficult-to-learn examples lie around the decision boundary, they should have higher augmentation similarity to examples *sharing this decision boundary* but from the different class", and accordingly revise all corresponding expressions in our next version (as the pdf revision deadline is past now).

---

> > > > ### Author Response · Authors · 2024-12-01
> > > > **Responses to further concerns of Reviewer eRke (2/2)**
> > > >
> > > > **Q4.** I do not think that discretizing the true adjacency matrices using only three values, $\alpha$, $\beta$, and $\gamma$ provides sufficient precision to the task.
> > > >
> > > > **A4.** Instead of using only three values, **our modeling of the adjacency matrix could be relaxed by adding random terms to the similarity values.** Specifically, we replace $\boldsymbol{A}$ with $\boldsymbol{A}'=(a'\_{ij})$, where $a'\_{ii}=1$, and $a'\_{ij}=a\_{ij}+\epsilon \cdot \varepsilon\_{ij}$ for $i\neq j$, $a\_{ij}$ takes values in $\{\alpha, \beta, \gamma\}$, $\varepsilon_{ij}=\varepsilon_{ji}$ are i.i.d. Gaussian random variables with mean 0 and variance 1, $\epsilon>0$ is a small constant. Then $\boldsymbol{A}'$ can be decomposed into $\boldsymbol{A}'=\boldsymbol{A} + \epsilon\cdot\boldsymbol{W} - \epsilon \cdot \mathrm{diag}(\varepsilon_{ii})$, where $\boldsymbol{W}$ turn out to be a real Wigner matrix. According to Wigner’s semicircle law, we have the eigenvalues of $\boldsymbol{W}$ with the probability density function $f(\nu)=\frac{1}{2\pi}\sqrt{4-x^2}\boldsymbol{1}[|x|\leq 2]$. Them following the similar approach in the proof Theorem 3.2, we are able to calculate the upper bound of (the expected value of) the $K+1$ largest eigenvalue of $\boldsymbol{A}'$, and accordingly reach a downstream error bound. Note that the effect of this random similarity $\epsilon \cdot \varepsilon_{ij}$ is to add an additional term to the upper bound of the eigenvalue, and the effect is the same regardless of $\boldsymbol{A}$ (e.g. with and without the existence of difficult-to-learn examples). Therefore, **our theoretical results still hold under this relaxation.** Please refer to Appendix B.3 for more detailed discussions.
> > > >
> > > > ---
> > > >
> > > > **Q5.** In terms of novelty, I still maintain of point of limited novelty based on previous works.
> > > >
> > > > **A5.** **As commented by Reviewer KG9m, "The paper offers novel insights into the impact of difficult-to-learn examples in contrastive learning".** Also, as mentioned in the contributions, we are the first (both empirically and theoretically) to demonstrate that removing difficult-to-learn examples boosts the performance of unsupervised contrastive learning. **It is impossible for this work to be a combination of previous works, because WE ARE THE FIRST TO INVESTIGATE THIS TOPIC.** Please specify if you have further novelty concerns.
> > > >
> > > > ---
> > > >
> > > > Therefore, we confidently uphold our perspective and respectfully disagree with reviewer eRke's evaluation of our work. At the same time, we hope that our response can clarify reviewer eRke's misunderstandings of the paper and lead to a re-evaluation of the work. We are always welcoming further discussions.

---

> > > > > ### Author Response · Authors · 2024-12-02
> > > > >
> > > > > Dear Reviewer eRke,
> > > > >
> > > > > We sincerely thank you for the time and effort you have put into reviewing our manuscript. We understand that this is a busy time, and we deeply appreciate the effort you have made in reviewing our work.
> > > > >
> > > > > **We have addressed each of the concerns you raised yesterday, and we hope that our responses have helped clarify some of the misunderstandings you had about the paper. However, we have not yet received your follow-up feedback based on these replies.** As the Author-Reviewer discussion period is nearing its end, we would be very grateful if you could kindly let us know if our responses have effectively addressed your concerns.
> > > > >
> > > > > Best regards,
> > > > > Authors

---

> ### Author Response · Authors · 2024-11-25
>
> Dear Reviewer eRke,
>
> We would like to express our sincere gratitude for the time and effort you have dedicated to reviewing our manuscript and providing thoughtful feedback. We understand that this is a particularly busy period, and we greatly appreciate the demands on your time.
>
> We have made every effort to carefully address your comments and to conduct additional experiments as per your suggestions. As the Author-Reviewer discussion period is drawing to a close, we would be truly grateful if you could kindly let us know whether our responses have adequately addressed your concerns.
>
> Once again, thank you for your invaluable contributions to our work.
>
> Best regards,
> Authors

---

> > ### Author Response · Authors · 2024-11-26
> >
> > Dear reviewer  eRke,
> >
> > We really appreciate your efforts to help improve this paper. We have carefully addressed all the mentioned concerns. We understand that you might be busy at this time, and we would deeply appreciate it if you could spare a little time to take a look at our rebuttal.
> >
> > Having further discussions really helps to achieve consensus and clarify misunderstandings, so we are eager to know if are there any remaining questions. We are more than happy to try our best to address them.
> >
> > Best regards,
> >
> > The Authors.

---

> ### Author Response · Authors · 2024-11-27
> **New updates of the ImageNet-1k experiment**
>
> Dear Reviewer eRke,
>
> We have made new progress regarding the ImageNet-1k experiment that we responded to a week ago. We have now obtained the results after running for 100 epochs, and the experimental results are as follows:
>
> | Imagenet-1k | Baseline |Removing   | Temperature Scaling   | Margin Tuning   | Combined Method   |
> |-------------|----------|-----|-----|-----|-----|
> |100 Epoch Accuracy  |  34.21 | 34.33| 35.02| 35.17| 35.68|
>
> From the experimental results, it is clear that our proposed method can achieve performance improvements on the real-world dataset, ImageNet-1k, similar to those observed on other datasets. Our experiments are still ongoing, and we will provide timely updates with new results as they become available.
>
> In addition, we have addressed each of your other concerns one by one. We eagerly look forward to your feedback on the rebuttal and further discussion, as we believe this is crucial for enhancing the quality of this paper.

---

> ### Author Response · Authors · 2024-11-29
> **Request for further specification of your concerns**
>
> Dear reviewer eRke,
>
> We have carefully addressed your concerns and we are still looking forward to your reply. As there is some time left, we sincerely ask you to specify your concern about the experiments, so that we could possibly provide additional results in the remaining limited time to address your concerns better. **As the description of your last concern is quite vague, would you please specify which part of the experiment you find "humble" and lacking "strong practical implication"?** This could really help in further improving our paper.
>
> In addition, **we sincerely ask you to rethink about the confidence score. There seems to be a severe misunderstanding of our paper, as especially evidenced by **Q3** and **Q5**.**
>
> Looking forward to your reply and further discussions.
>
> Best regards,
>
> The Authors.

---

> ### Comment · Reviewer_eRke · 2024-12-02
> **Response to the authors**
>
> Thanks the authors for your response.
>
> Although in the main paper, we do not mention explicitly how you use the adjacency matrix in Figure 3d. However, when I check your proof for Theorems 3.1 and 3.2, in Eq. (22) and Eq. (23), you use the discretized adjacency matrix $A$ in Figure 3d for solving $||\bar{A}-FF^{T}||_{F}^{2}$ where $\bar{A}$ is the normalized matrix of $A$. This is more evident for this use because in Line 1035 to 1039, you obtain very specific eigenvalues and eigenvectors for A which is impossible for a general similarity matrix A. This means that to develop theories, you use the discretized adjacency matrix in Figure 3d which does not convince me.  Similar observation for other theorems and corollaries.

---

> ### Author Response · Authors · 2024-12-02
> **Response to your remaining concern**
>
> Thank you for the timely response. We address your concern below.
>
> ---
>
> **Q.** To develop theories, you use the discretized adjacency matrix in Figure 3d which does not convince me. Similar observation for other theorems and corollaries.
>
> **A.** We would like to point out that in A4 of our previous response (and also in Appendix B.3), we discuss that **the discretized values in the adjacency matrix can be easily extended to a generalized one by adding element-wise random variables, without harming the theoretical insights**. To further elaborate, the effect of this randomization is to add an additional term to the upper bound of the eigenvalue. (**Please note that different from Theorem 3.1, in the proof of Theorem 3.2 (Lines 1148-1155), we use the same mathematical tool to derive an upper bound of the eigenvalues instead of the specific values.**) That means, the eigenvalues of the general similarity matrix are decomposed into two parts. The first part is the upper bound of the discretized adjacency matrix we derived in the proofs, and the second part is the eigenvalues of the mean-zero random matrix. (This could be roughly understood as a bias-variance decomposition, where the discretized matrix forms the bias term, and the randomized matrix forms the variance term.) As the "variance" term is the same regardless of the adjacency matrix, the relative quantity of the linear probing error bound relies only on the "bias" term, i.e. eigenvalues of the discretized matrix, and therefore our theoretical results still hold even under generalized modeling.
>
> Perhaps easier to understand, we summarize it in a non-mathematical way. **In the main paper we use the discretized adjacency matrix to keep the simplicity and clarity of the theorems and the theoretical insights. This framework "allows easier (tractable) analysis with the spectral contrastive loss" as commented by Reviewer aLaR. In Appendix B.3, we discuss how to extend the discretized adjacency matrix to a generalized one, and why our theoretical insights still hold under this extension.** However, if you strongly insist the general similarity matrix should be reflected in the main paper, we can replace Figure 3d and the proofs of theorems according to Appendix B.3 in our next version (considering that the pdf revision period is now over after several rounds of our discussions).
>
> ---
>
> We are still open to further discussions. If any of our previous responses have not fully addressed your questions, please do let us know.

---

> > ### Author Response · Authors · 2024-12-03
> >
> > Dear Reviewer eRke,
> >
> > We have already addressed your latest questions in our response yesterday. As the Author-Reviewer discussion period is within the final 12 hours, we kindly suggest that you revisit our previous responses and the paper. If there are any remaining questions or points you would like to discuss, we would be happy to continue the discussion.
> >
> > Best regards,
> > Authors

---

> ### Comment · Reviewer_aLaR · 2024-12-03
>
> Thank you for conducting more experiments and the detailed response.
>
> Indeed, as you have shown in the new experiment, there's a significant difference between the mean of considered $\beta$ and $\gamma$ measures for real-world datasets. I appreciate your effort and this result. That being said, even with the proposed relaxation, the theoretical framework is very interesting yet still a bit unconvincing for me, so that I'd like to keep my current score. The main reason is that, as also pointed out by reviewer eRke, it is still unconvincing that the discretized $A$ resembles the real $A$, since the distribution or overlap between $\beta$ and $\gamma$ might be critical to the behavior of the training and trained models. Although such a relaxation is possible, it is still unclear that 1) to what extent can we add a noisy $W$ to the ideal $A$ without hurting the results (i.e., any bounds etc.) and 2) If the only difference between the ideal $A$ and real $A$ in datasets is a GOE $W$. I believe that such limitation can better (only) be covered by a further research with intensive experiments and/or theoretical discussions to support the claims, which might be beyond the scope of current work. Therefore, although I lean towards to acceptance, I cannot raise my score to 8. Nevertheless, I really appreciate the fruitful discussion and experiments provided by the authors.

---

> ### Author Response · Authors · 2024-12-04
> **Further explanations on the relaxation of theoretical framework (1/2)**
>
> Thank you for the reply. We address your concerns with further relaxation of the assumptions (verified empirically) (**A1**) and with additional error bounds (**A2**). (For better explanations, we change the question order.)
>
> ---
>
> **Q1.** If the only difference between the ideal $A$ and real $A$ in datasets is a GOE $W$.
>
> **A1.** Unfortunately, this is not guaranteed. However, to derive the generalized theoretical bounds, the difference between the ideal $A$ and real $A$ does not have to be a GOE. Instead, **the generalized results only require the difference $\boldsymbol{W}=(\varepsilon_{ij})$ to be a Wigner matrix** so we can use Wigner’s Semicircle Law to get the eigenvalues of $\boldsymbol{W}$. That is, **the GOE assumption could be further relaxed to a Wigner matrix**, i.e. a symmetric random matrix where $\varepsilon_{ii}$ are i.i.d. distributed with $\mu(\varepsilon_{ii})$ and $\sigma(\varepsilon_{ii})$ bounded, and $\varepsilon_{ij}$ ($i<j$) are i.i.d. distributed with $\mu(\varepsilon_{ij})=0$ and $\sigma(\varepsilon_{ij})=1$.
>
> **We use the proxy augmentation graph to verify this generalized assumption.** We denote $\boldsymbol{W}'$ as the difference between the proxy matrix $\boldsymbol{A}'$ and the ideal matrix $\boldsymbol{A}$, i.e. $\boldsymbol{A}' = \boldsymbol{A} + \boldsymbol{W}'$. By calculations, $W'$ has mean zero and standard deviations 0.1138 (CIFAR-10), 0.0821 (CIFAR-100), and 0.1232 (Imagenet-1k). That is, we can normalize $\boldsymbol{W}'$ to be a Wigner matrix $\boldsymbol{W}$ with $\varepsilon_{ii}$ following the Dirac distribution ($\mu(\varepsilon_{ii})=\sigma(\varepsilon_{ii})=0$). (The diagonal elements of $\boldsymbol{A}'$ and $\boldsymbol{A}$ are both 1, so the diagonal elements of $\boldsymbol{W}'$ is 0.) Then we have $\boldsymbol{A}' = \boldsymbol{A} + \epsilon\boldsymbol{W}$, with $\epsilon=0.1138$ (CIFAR-10), $0.0821$ (CIFAR-100), or $0.1232$ (Imagenet-1k).
>
> ---
>
> **Q2.** To what extent can we add a noisy $W$ to the ideal $A$ without hurting the results (i.e., any bounds etc.)
>
> **A2.** Under the generalized assumption discussed in **A1** ($\boldsymbol{A}' = \boldsymbol{A} + \epsilon\boldsymbol{W}$), we derive additional error bounds to show the effect of $W$. **The range of $\epsilon$ is especially discussed in Remark 2**.
>
> **Theorem 3.1 (Generalized version)** *Denote $\mathcal{E}\_{\mathrm{w.o.}}$ as the linear probing erorr of a contrastive learning model trained on a dataset without difficult-to-learn examples. Under the generalized assumption that $\boldsymbol{A}' = \boldsymbol{A} + \epsilon\boldsymbol{W}$, where $\boldsymbol{W}$ is a Wigner matrix with $\varepsilon_{ii}$ following the Dirac distribution, then if $n(r+1)$ is large enough, we have*
> $$
> \mathcal{E}_{\mathrm{w.o.}} \leq \frac{4\delta}{1 - \frac{1-\alpha}{(1-\alpha)+n\alpha+nr\beta}-\frac{1}{(1-\alpha)+n(\alpha+r\beta)}x_0\cdot\epsilon}+8\delta,
> $$
> *where $x_0 \in (0,2)$ is the unique solution to the folowing Kepler's equation*
> $$
> \frac{1}{2}x_0\sqrt{4-x_0^2}+2\arg\sin(x_0/2)=\Big[1-\frac{2}{r+1}\frac{n_d}{n}\Big]\pi.
> $$
>
>
> **Theorem 3.2 (Generalized version)** *Denote $\mathcal{E}\_{\mathrm{w.d.}}$ as the linear probing erorr of a contrastive learning model trained on a dataset with $n_d$ difficult-to-learn examples per class. Under the generalized assumption that $\boldsymbol{A}' = \boldsymbol{A} + \epsilon\boldsymbol{W}$, where $\boldsymbol{W}$ is a Wigner matrix with $\varepsilon_{ii}$ following the Dirac distribution, if $n(r+1)$ is large enough and $r+1 \leq k < n_d+r+1$, we have*
> $$
> \mathcal{E}_{\mathrm{w.o.}} \leq \frac{4\delta}{1 - \frac{(1-\alpha)+r(\gamma-\beta)}{(1-\alpha)+n\alpha+nr\beta+n_dr(\gamma-\beta)}-\frac{1}{(1-\alpha)+n(\alpha+r\beta)}x_0\cdot\epsilon}+8\delta.
> $$
>
> **Remark 1. (Value of $x_0$)** We derive the value of $x_0$ through numerical methods for multiple datasets, by using the empirical values of $\alpha$, $\beta$, and $\gamma$ calculated on the proxy augmentation graph. We have $x_0=1.894$ for CIFAR-10, $x_0=1.976$ for CIFAR-100, and $x_0=1.995$ for Imagenet-1k. Intuitively, according to Wigner's Semicircle Law, because $n_d \ll n(r+1)$, the value of $x_0$ is near $2$.

---

> ### Author Response · Authors · 2024-12-04
> **Further explanations on the relaxation of theoretical framework (2/2)**
>
> **A2. (continued)**
>
> **Remark 2. (Range of $\epsilon$)**  The bounds are valid if $\frac{1-\alpha}{(1-\alpha)+n\alpha+nr\beta}+\frac{1}{(1-\alpha)+n(\alpha+r\beta)}x_0\cdot\epsilon <1$ and $\frac{(1-\alpha)+r(\gamma-\beta)}{(1-\alpha)+n\alpha+nr\beta+n_dr(\gamma-\beta)}+\frac{1}{(1-\alpha)+n(\alpha+r\beta)}x_0\cdot\epsilon < 1$. As $\frac{(1-\alpha)+r(\gamma-\beta)}{(1-\alpha)+n\alpha+nr\beta+n_dr(\gamma-\beta)} > \frac{1-\alpha}{(1-\alpha)+n\alpha+nr\beta}$, we require $\epsilon < \epsilon_{\mathrm{bound}} = \frac{1-\frac{(1-\alpha)+r(\gamma-\beta)}{(1-\alpha)+n\alpha+nr\beta+n_dr(\gamma-\beta)}}{\frac{1}{(1-\alpha)+n(\alpha+r\beta)}x_0}$. According to the empirical values of $\alpha$, $\beta$, $\gamma$, and $x_0$, we calculate the upper bounds of $\epsilon$ as $\epsilon_{\mathrm{bound}} = 6288$ for CIFAR-10, $\epsilon_{\mathrm{bound}} = 91297$ for CIFAR-100, and $\epsilon_{\mathrm{bound}} = 998839$ for Imagenet-1k. Besides, according to the discussions in **A1**, **we have empirically $\epsilon \ll \epsilon_{\mathrm{bound}}$ for all three datasets**.
>
> **Remark 3. (Existing conclusions still hold)** The effect of $W$ is to add an additional term to the eigenvalue, and the effect is the same for both situations (with and without difficult-to-learn examples). That is, even under the generalized assumptions, **we still have the conclusion that "the presence of difficult-to-learn examples leads to a strictly worse linear probing error bound"**. Moreover, as Corrollary 4.1 can be directly derived by Theorem 3.1, the generalized version becomes $\mathcal{E}\_\mathrm{R} \leq \frac{4\delta}{1 - \frac{1-\alpha}{(1-\alpha)+(n-n_d)\alpha+(n-n_d)r\beta}-\frac{1}{(1-\alpha)+(n-n_d)(\alpha+r\beta)}x_0\cdot\epsilon}+8\delta$. Similarly, as the bounds in Theorems 4.3 and 4.5 are based on a modified similarity matrix, we have $\mathcal{E}\_\mathrm{M} = \mathcal{E}\_{\mathrm{w.o.}}$ (Theorem 4.3) and $\mathcal{E}\_{\mathrm{T}} \leq \frac{4[1-(n_d/n)^2+(\gamma/\beta)^2(n_d/n)^2] \delta}{1-\frac{1-\alpha}{(1-\alpha)+n\alpha+nr\beta}-\frac{1}{(1-\alpha)+n(\alpha+r\beta)}x_0\cdot\epsilon} + 8\delta$ (Theorem 4.5), where the theoretical insights of these two theorems remain unchanged. That is, even under the generalized assumptions, **we still have the conclusion that sample removal, margin tuning, and temperature scaling improve the error bound under the existence of difficut-to-learn examples.**
>
> **Proof sketch.**
> Because $\mathbb{E}\boldsymbol{W}=\boldsymbol{0}$, when $n(r+1)$ is large enough, after normalization, we have $\bar{\boldsymbol{A}}' = \bar{\boldsymbol{A}} + \frac{1}{(1-\alpha)+n(\alpha+r\beta)}\cdot\epsilon\boldsymbol{W}$. By Equation 13 in Fulton (2000), when $r+1 \leq k < n_d+r+1$, we have the $k+1$-th largest eigenvalue of $\bar{\boldsymbol{A}}'$ satisfying
> $$
> \lambda'\_{k+1} \leq \min_{i+j=k+2} \lambda_i + \frac{1}{(1-\alpha)+n(\alpha+r\beta)}\cdot\epsilon \nu_j \leq \lambda_{r+2} + \frac{1}{(1-\alpha)+n(\alpha+r\beta)}\cdot\epsilon\nu_{n_d},
> $$
> where $\lambda_i$ is the $i$-th largest eigenvalue of $\bar{\boldsymbol{A}}$ and $\nu_j$ is the $j$-th largest eigenvalue of $\boldsymbol{W}$.
>
> On the one hand, according to the proofs of Theorems 3.1 and 3.2 in our paper, we have $\lambda_{r+2} = \frac{1-\alpha}{(1-\alpha)+n(\alpha+r\beta)}$ (Theorem 3.1) and $\lambda_{r+2} \leq \frac{(1-\alpha)+r(\gamma-\beta)}{(1-\alpha)+n(\alpha+r\beta)+n_dr(\gamma-\beta)}$ (Theorem 3.2).
>
> On the other hand, Because $\boldsymbol{W}$ is a Wigner matrix, we have its empirical spectral measure $\nu=\frac{1}{n(r+1)}\sum_{i=1}^{n(r+1)} \delta_{\nu_i}$ converging weakly almost surely to the quarter-circle distribution on $[0, 2]$, with density $f(\nu)=\frac{1}{2\pi}\sqrt{4-x^2}\boldsymbol{1}[|x|\leq 2]$. When $j \leq n(r+1)/2$ and $n(r+1)$ large enough, by symmetry of $f(\nu)$, we have
> $$
> \frac{1}{2}\Big[1-\frac{2j}{n(r+1)}\Big] = \int_{x=0}^{\nu_j} f(\nu) \,d\nu = \frac{1}{2\pi} \Big[\frac{1}{2}\nu_j\sqrt{4-\nu_j^2} + 2\arg\sin(\nu_j/2)\Big]. \text{    (*)}
> $$
>
> Then combine the above calculations, we have $\lambda'\_{k+1} \leq \frac{1-\alpha}{(1-\alpha)+n(\alpha+r\beta)} + \nu_j$ for the generalized Theorem 3.1 and $\lambda'\_{k+1} \leq \frac{(1-\alpha)+r(\gamma-\beta)}{(1-\alpha)+n(\alpha+r\beta)+n_dr(\gamma-\beta)} + \frac{1}{(1-\alpha)+n(\alpha+r\beta)}\cdot\epsilon \nu_j$, where $\nu_j$ is the solution to (*). Then we complete the proof by deriving the error bounds using the upper bounds of $\lambda'_{k+1}$.
>
> ---
>
> We appreciate that your reply always lead to potential improvements of our paper. Hope our responses can address your remaining concerns.

---

### Author Response · Authors · 2024-12-04
**Summary of additional theorems and experiments during rebuttal**

Dear Program Chairs, Senior Area Chairs, and Area Chairs, and Reviewers,

We would like to express our sincere gratitude to the Program Chairs, Senior Area Chairs, and Area Chairs for their contributions to the conference. We also thank the reviewers for their valuable suggestions on our paper.

Through our discussions with the reviewers, we have gained valuable insights and have made additional theoretical and experimental contributions according to the reviewers' suggestions.

The key points are summarized below:

**Additional Theoretical Contributions**:
1. We propose a potential generalization of the theoretical modeling, i.e. we assume the adjacency matrix to be $\boldsymbol{A}' = \boldsymbol{A} + \epsilon\boldsymbol{W}$, where $\boldsymbol{W}$ is a Wigner matrix. Based on this relaxation, we improve all theorems in the paper to a generalized version, without harming the original theoretical insights.

2. We discuss how our method can be possibly generalized to the InfoNCE loss.

**Additional Experimental Contributions**:

1. We conducted experiments on the larger, more realistic ImageNet1k dataset, and the results confirm the effectiveness of our method on large datasets.
2. To better understand our proposed selection mechanism, we visualized the distribution of the selected sample pairs from different classes in the CIFAR-10 dataset. This indicates that our theoretical modeling of difficult-to-learn examples is realistic.
3. We calculate the numerical values of $\alpha$, $\beta$, and $\gamma$ through a proxy augmentation graph on multiple real datasets. We also performed statistically significant tests on $\beta$ and $\gamma$ using the ImageNet1k dataset, proving that the difference between $\beta$ and $\gamma$ is highly statistically significant.
4. We empirically verify the variation trend of our bounds using the mixed-CIFAR datasets. We show that a larger $\gamma - \beta$ leads to a worse error bound.
5. We empirically verify the additional assumptions used in the generalized versions of our theorems.
6. We vary the batch size in the experiment. The results show that our sample removal improves performance across various batch sizes.

These experimental results further demonstrate the robustness of our theory and the potential effectiveness of the proposed method.

Finally, we would like to once again express our sincere appreciation for the time and effort you dedicated to reviewing and engaging with our work.

Best regards,
The Authors

---

### Meta-Review · Area_Chair_mQPS · 2024-12-20

**Metareview:**

Based on the reviews, I conclude that the paper cannot be accepted for publication in its current form. Of the four reviews, two recommend borderline acceptance, one recommends borderline rejection, and one recommends outright rejection.
The reviewer recommending rejection expressed high confidence in their assessment, citing several critical issues. The paper does not present strong empirical evidence or demonstrate significant practical implications for the research community. Moreover, concerns were raised about the novelty of the theoretical approach, suggesting that the contributions may not be sufficiently original or impactful.

**Additional Comments On Reviewer Discussion:**

The main points raised by the reviewers focused on the paper’s theoretical rigor, novelty, and empirical evaluation:

- **Theoretical Rigor and Novelty**: Reviewer eRke expressed concern about the confusing terminology, lack of rigorous definitions, and limited novelty. Despite additional experiments and responses from the authors, the reviewer remained unconvinced, stating that the theoretical contributions lacked sufficient originality and practical implications. This point carried significant weight due to the reviewer’s high confidence.

- **Empirical Evaluation**: all reviewers pointed out weaknesses in the empirical validation, with some suggesting that the experiments were insufficient or lacked scale. The authors provided additional experiments, including results on Imagenet-1k, but the improvements were not enough to fully address concerns.

- **Practical Implications**: Reviewer eRke and others raised concerns about the practical relevance of the work. Despite responses from the authors, eRke maintained that the work failed to demonstrate significant real-world or broader applicability.

While some reviewers were satisfied with the additional responses and experiments (KG9m, aLaR), the key criticisms around theoretical novelty, methodological clarity, and empirical validation remained unresolved. These concerns resulted in the final decision to recommend rejection.

---

### Decision · Program_Chairs · 2025-01-22

Reject